# Generalization Properties of NAS under Activation and Skip Connection Search

**Zhenyu Zhu,    Fanghui Liu,    Grigorios G Chrysos,    Volkan Cevher**

EPFL, Switzerland
{[first name].[surname]}@epfl.ch

## Abstract

Neural Architecture Search (NAS) has fostered the automatic discovery of state-of-the-art neural architectures. Despite the progress achieved with NAS, so far there is little attention to theoretical guarantees on NAS. In this work, we study the generalization properties of NAS under a unifying framework enabling (deep) layer skip connection search and activation function search. To this end, we derive the lower (and upper) bounds of the minimum eigenvalue of the Neural Tangent Kernel (NTK) under the (in)finite-width regime using a certain search space including mixed activation functions, fully connected, and residual neural networks. We use the minimum eigenvalue to establish generalization error bounds of NAS in the stochastic gradient descent training. Importantly, we theoretically and experimentally show how the derived results can guide NAS to select the top-performing architectures, even in the case without training, leading to a train-free algorithm based on our theory. Accordingly, our numerical validation shed light on the design of computationally efficient methods for NAS. Our analysis is non-trivial due to the coupling of various architectures and activation functions under the unifying framework and has its own interest in providing the lower bound of the minimum eigenvalue of NTK in deep learning theory.

## 1   Introduction

Neural Architecture Search (NAS) [Zoph and Le, 2017] is a powerful technique that enables the automatic design of neural architectures. NAS defines a set of operations (referred to as the *search space*), that include various activation functions and layer types, or potential connections among layers [Elsken et al., 2019, Ren et al., 2021]. Optimization over the search space returns the optimal architecture as a subset of the possible combinations of operations. NAS[1] obtains state-of-the-art results in image recognition [Liu et al., 2019a, Ding et al., 2020, Zhang et al., 2019, Chen et al., 2019] or can be used to further improve architectures defined by a human expert [Tan and Le, 2019]. The spectacular results obtained by NAS have led to a significant interest in the community to further improve the NAS algorithms, the search space etc. However, to date little focus has been provided in the following question: *Can NAS[1] achieve generalization guarantees similar to a typical neural network?*

Neural tangent kernel (NTK)-based analysis [Jacot et al., 2018] is a powerful method for analyzing the optimization and the generalization of deep networks [Allen-Zhu et al., 2019, Cao and Gu, 2019, Chen et al., 2020a, Arora et al., 2019a]. The minimum eigenvalue of NTK has been used in previous work to demonstrate the global convergence of gradient descent, such as two-layer networks [Du et al., 2019b], and deep networks with polynomially wide layers [Allen-Zhu et al., 2019]. Besides, the minimum eigenvalue of NTK is also used to prove generalization bounds [Arora et al., 2019a] and

---

[1] In the sequel, we interchangeably refer to NAS as the "architecture obtained from NAS" or the framework to design the neural architecture.

36th Conference on Neural Information Processing Systems (NeurIPS 2022).

memorization [Montanari and Zhong, 2020]. However, previous work mainly focuses on a limited set of architectures, e.g., fully-connected (FC) neural networks [Allen-Zhu et al., 2018, Bartlett et al., 2017] or residual neural networks [He et al., 2016, Huang et al., 2020], in which a single activation function is used throughout the network. These off-the-shelf theoretical results cannot be directly applied to analyze the rich search space (of NAS) that is covering various/mixed architectures and parameters. That makes the non-trivial analysis on NAS worth of study on its own right.

The recent work of Oymak et al. [2021] is the first work to provide generalization guarantees on a related problem, i.e., activation functions search. The study provides generalization results on two-layer networks relying on the minimum eigenvalue with a strictly larger than zero assumption, i.e., $\lambda_{\min}(\boldsymbol{K}) > 0$ for the NTK matrix $\boldsymbol{K}$.

In this work, we introduce the first theoretical guarantees for multilayer NAS where the search space includes activation functions and skip connections. We study the upper/lower bound of the minimum eigenvalue of NTK (in the (in)finite regime) under mixed activation functions and architectures which evade the minimum eigenvalue assumption of Oymak et al. [2021]. Then, we provide optimization and generalization guarantees of deep neural networks (DNNs) equipped with NAS. Our results indicate that the minimum eigenvalue estimation can act as a powerful metric for NAS. This method, called Eigen-NAS, is train-free, but still effective with experimental validation when compared to recent promising algorithms [Xu et al., 2021, Chen et al., 2021, Mellor et al., 2021]. Formally, our main contribution and findings are summarized below:

i) We build a general theoretical framework based on NTK for NAS with search on popular activation functions in each layer, fully-connected, and skip connections. We derive the NTK formula of these architectures in the (in)finite-width regime under the unifying framework.

ii) We derive the upper and lower bounds of the minimum eigenvalue of the NTK under the (in)finite-width regime for the considered architectures. We introduce a new technique to ensure the probability of concentration inequality remains positive. Our analysis highlights how the upper and lower bounds differs under activation function search and skip connection search and can guide NAS.

iii) We establish a connection between the minimum eigenvalue and generalization of the searched DNN trained by stochastic gradient descent (SGD). Our theoretical results show that the generalization performance largely depends on the minimum eigenvalue of NTK for NAS, which provides theoretical guarantees for the searched architecture.

iv) Our theoretical results are supported by thorough experimental validations with the following findings: 1) our upper and lower bounds on the minimum eigenvalue largely depend on the activation function in the first layer rather than the activation functions in deeper layers. 2) The applied NAS algorithm always picks up ReLU (Rectified Linear Unit) and LeakyReLU in the optimal architecture, which coincides with our theory that predicts ReLU and LeakyReLU achieve the largest minimum eigenvalues. 3) The skip connections are required in each layer under our not very large DNNs. Furthermore, our experimental evidence on Eigen-NAS indicates that the minimum eigenvalue is a promising metric to guide NAS (without training) as suggested by our theory.

**Technical challenges.** The technical challenges of this paper mainly focus on how to analyze activation functions with different properties and skip connections under a unifying framework. This work is non-trivial; previous works mainly focus on the ReLU activation function [Nguyen et al., 2021, Cao and Gu, 2019, Allen-Zhu et al., 2019] in optimization and generalization of a single fully-connected neural network. Their proofs heavily depend on the properties of $\mathrm{ReLU}$, e.g., homogeneity and $\mathrm{ReLU}(x) = x\mathrm{ReLU}'(x)$ which are invalid when other commonly-used activation functions, e.g., Tanh, Sigmoid, and Swish, are used. This problem becomes harder when mixed activation functions and residual connections are considered. To tackle these technical challenges, we develop the following techniques: a) to handle the non-homogeneous property of Tanh, Sigmoid, and Swish, we develop a new integral estimation approach for the minimal eigenvalue estimation. b) To establish the connection between the minimum eigenvalues of NTK and generalization errors, we use the Lipschitz continuity to avoid the special property of ReLU. More importantly, we introduce a new technique [Yaskov, 2014] to replace Gershgorin circle theorem for minimum eigenvalue estimation, which avoids concentration inequalities with negative probability in some certain cases [Nguyen et al., 2021].

## 2 Related work

**Network architecture search (NAS):** The idea of NAS stems from Zoph and Le [2017], while the idea of cell search, i.e., searching core building blocks and composing them together, emerged in Zoph et al. [2018]. The earlier literature used discrete optimization techniques for obtaining the architecture. DARTS [Liu et al., 2019b] considers NAS as a continuous bi-level optimization task. Recent variants of DARTS [Xu et al., 2019, Wu et al., 2019] and several train-free methods [Mellor et al., 2021, Chen et al., 2021, Xu et al., 2021] have demonstrated success in reducing the search time or improving the search algorithm. However, the aforementioned works have not provided generalization guarantees for the optimal architecture.

**Optimization and generalization of DNNs via NTK:** In the NTK framework [Jacot et al., 2018, Du et al., 2019a, Chen et al., 2020b], the training dynamics of (in)finite-width networks can be exactly characterized by kernel tools. Leveraging NTK facilitates studies on the global convergence of GD Allen-Zhu et al. [2019], Du et al. [2019a], Nguyen [2021] in DNNs via the minimum eigenvalue of NTK. In fact, it also controls the generalization performance of DNNs [Du et al., 2019b, Cao and Gu, 2019, Allen-Zhu et al., 2018], which is further studied in Bietti and Bach [2021].

## 3 Problem Settings

In this section we introduce the problem setting of our NAS framework based on the search space and algorithm (search strategy) for our paper.

Let $X \subseteq \mathbb{R}^d$ be a compact metric space and $Y \subseteq \mathbb{R}$. We assume that the training set $\mathcal{D}_{tr} = \{(\boldsymbol{x}_i, y_i)\}_{i=1}^N$ is drawn from a probability measure $\mathcal{D}$ on $X \times Y$, with its marginal data distribution denoted by $\mathcal{D}_X$. The goal of a supervised learning task is to find a hypothesis (i.e., a neural network used in this work) $f : X \to Y$ such that $f(\boldsymbol{x}; \boldsymbol{W})$ parameterized by $\boldsymbol{W}$ is a good approximation of the label $y \in Y$ corresponding to a new sample $\boldsymbol{x} \in X$. In this paper, we consider the classification task, evaluated by minimizing the expected risk

$$\min_{\boldsymbol{W}} \ \ell_{\mathcal{D}}(\boldsymbol{W}) := \mathbb{E}_{(\boldsymbol{x},y)\sim\mathcal{D}} \ \ell[yf(\boldsymbol{x}; \boldsymbol{W})] \,,$$

where $\ell[yf(\boldsymbol{x}; \boldsymbol{W})]$ is the classification loss $\ell(\cdot)$ as a surrogate of the expected 0-1 loss $\ell_{\mathcal{D}}^{0-1}(\boldsymbol{W}) := \mathbb{E}_{(\boldsymbol{x},y)\sim\mathcal{D}}[\mathbb{1}\{yf(\boldsymbol{x}; \boldsymbol{W}) < 0\}]$. In this paper, we employ the cross-entropy loss, which is defined as $\ell(z) = \log[1 + \exp(-z)]$.

*Notation:* For an integer $L$, we use the shorthand $[L] = \{1, 2, \ldots, L\}$. The multivariate standard Gaussian distribution is $\mathcal{N}(\boldsymbol{0}, \mathbb{I}_d)$ with the zero-mean vector $\boldsymbol{0}$ and the identity-variance matrix $\mathbb{I}_d$. We denote the direct sum by $\oplus$. We follow the standard Bachmann–Landau notation in complexity theory e.g., $\mathcal{O}, o, \Omega$, and $\Theta$ for order notation.

### 3.1 Neural Networks and Search Space

In this work, we consider a particular parametrization of $f$ as a deep neural network (DNN) with depth $L$ $(L \geq 3)$[2] which includes the fully-connected (FC) neural networks setting and the residual neural networks setting, and various activation functions in each layer. This enables a quite general NAS setting. Formally, we define a single-output DNN with the output $\boldsymbol{f}_l(\boldsymbol{x})$ in each layer

$$\boldsymbol{f}_l(\boldsymbol{x}) = \begin{cases} \boldsymbol{x} & l = 0 \,, \\ \sigma_1(\boldsymbol{W}_1 \boldsymbol{x}) & l = 1 \,, \\ \sigma_l(\langle \boldsymbol{W}_l, \boldsymbol{f}_{l-1}(\boldsymbol{x}) \rangle) + \alpha_{l-1} \boldsymbol{f}_{l-1}(\boldsymbol{x}) & 2 \leq l \leq L-1 \,, \\ \langle \boldsymbol{W}_L, \boldsymbol{f}_{L-1}(\boldsymbol{x}) \rangle & l = L \,, \end{cases} \tag{1}$$

where the weights of the neural networks are $\boldsymbol{W}_1 \in \mathbb{R}^{m \times d}$, $\boldsymbol{W}_l \in \mathbb{R}^{m \times m}$, $l = 2, \ldots, L-1$ and $\boldsymbol{W}_L \in \mathbb{R}^m$. The binary parameter $\alpha_l$ is for layer search, and the activation function is $\sigma_l(\cdot)$. The neural network output is $f(\boldsymbol{x}; \boldsymbol{W}) = f_L(\boldsymbol{x})$.

**Architecture search:** A binary vector $\boldsymbol{\alpha} = [\alpha_1, \alpha_2, \cdots, \alpha_{L-2}]^\top$ represents the skip connections, where the $\alpha_l \in \{0, 1\}$ in Equation (1) indicates whether there is a skip connection in the $l$-th layer. Notice that we unify FC and residual neural networks under the same framework.

---

[2]Our results hold for the $L = 2$ setting corresponding to one-hidden layer neural network with slight modifications on notation, so we focus on $L \geq 3$ for simplicity.

Table 1: Formula of different activation functions, definitions of relevant constants and some intermediate results.

| $\sigma_l$ | ReLU | LeakyReLU | Sigmoid[1] | Tanh[2] | Swish |
|---|---|---|---|---|---|
| Formula | $\max(0, x)$ | $\max(\eta x, x),\ \eta \in (0,1)$ | $\frac{1}{1+e^{-x}} - \frac{1}{2}$ | $\frac{e^x - e^{-x}}{e^x + e^{-x}}$ | $\frac{x}{1+e^{-x}}$ |
| $\beta_1(\sigma_l)$ | 1 | $1 + \eta^2$ | $1/8$ | 2 | 1 |
| $\beta_2(\sigma_l)$ | 1 | $1 + \eta^2$ | $1/8$ | 2 | 1.22 |
| $\beta_3(\sigma_l)$ | 1 | $1 + \eta^2$ | $f_S(t)$ | $f_T(t)$ | $1/2$ |

[1] We consider the integral $f_S(y) = \int_{-\infty}^{\infty} \frac{2}{\sqrt{2\pi y}} e^{-\frac{x^2}{2y}} f'_{\mathrm{Sigmoid}}(x)^2 \mathrm{d}x$. We add $-1/2$ in Sigmoid to ensure $f_{\mathrm{Sigmoid}}(0) = 0$ facilitates our theoretical analysis. The parameter is $t := 3(1 + \eta^2)(2 + \eta^2)^{L-3}$.

[2] The definition of $f_T$ is similar to $f_S$ by using the Tanh function.

**Activation function search:** We select five representative activation functions defined by $\mathcal{F}_\sigma = \{\mathrm{ReLU}, \mathrm{LeakyReLU}, \mathrm{Sigmoid}, \mathrm{Tanh}, \mathrm{Swish}\}$ used in Equation (1), that can be bounded, unbounded, smooth, non-smooth, monotonic, or non-monotonic, as reported in Table 1. We define $\boldsymbol{\sigma} = [\sigma_1, \sigma_2, \cdots, \sigma_{L-1}]^\top$ with $\sigma_l \in \mathcal{F}_\sigma$ for any $l \in [L-1]$ as the indicator to show which activation function is selected in each layer. Our NAS framework allows for a different activation function in each layer, which enlarges the search space.

In our setting, we conduct the architecture search and the skip connection search independently, and accordingly, our search space is defined as the direct sum of them

$$\mathcal{W} := \mathbb{R}^{L-2} \oplus \mathcal{F}_\sigma^{L-1} \oplus \left\{ \mathbb{R}^{m \times d} \times (\mathbb{R}^{m \times m})^{L-2} \times \mathbb{R}^m \right\}, \tag{2}$$

where $\boldsymbol{W} := (\boldsymbol{\alpha}, \boldsymbol{\sigma}, \boldsymbol{W}_1, \ldots, \boldsymbol{W}_L) \in \mathcal{W}$ represents the collection of weight matrices and indicator for skips and selected activation functions for all layers.

### 3.2 Algorithm (Search Strategy)

The search strategy is the core part in NAS to pick up the optimal architecture from the search space. Here we build a general Algorithm 1 combining the search strategy for NAS (the first part) and the subsequent neural network training by SGD (the second part).

We firstly utilize a typical NAS algorithm, e.g., random search WS [Li and Talwalkar, 2020] or DARTS[3], to search skip connections and activation functions independently, which results in the optimal architecture $\{(\sigma_i^*)_{i=1}^{L-1}, (\alpha_i^*)_{i=1}^{L-2}\}$ with the max probability, see sec. 5.1 for details. In particular, Algorithm 1 also allows for the guidance of NAS in a train-free strategy via some specific metrics, e.g., the minimum eigenvalue of NTK (and its variant), see our Eigen-NAS method in sec. 5.2.

Then, we conduct neural network training on the selected architecture by SGD. For ease of theoretical analysis, we employ the constant step-size SGD with one epoch and randomly choose the weight parameters during all the iterations, which is commonly used in deep learning theory [Cao and Gu, 2019, Zou et al., 2019].

## 4 Main result

In this section, we state the main theoretical results. We present the assumptions used in our proof in sec. 4.1. Then in sec. 4.2 we provide the recursive form of NTK for DNNs defined by Equation (1) with mixed activation functions and skip connections. The upper and lower bounds of the minimum eigenvalue of NTK in the infinite and finite-width setting is given in sec. 4.3 and 4.4, respectively. Finally, in sec. 4.5, we connect the minimum eigenvalue of NTK and the generalization error bound of DNNs under these search schemes. The proofs of our theoretical results presented in this section are deferred to Appendix B, C, and D, respectively.

### 4.1 Assumptions

We make the following assumptions on data and activation functions. Our assumptions are frequently employed in the literature as we highlight below.

**Assumption 1.** We assume that the data satisfy $\|\boldsymbol{x}\|_2 = 1$.

---

[3] This algorithm directly outputs the final optimal architecture and optimal parameters.

**Algorithm 1:** SGD for training DNNs by NAS

---

**Input:** search space $\mathcal{S}$, data $\mathcal{D}_{tr} = \{(\boldsymbol{x}_i, y_i)_{i=1}^N\}$, step size $\gamma$ and $\texttt{Flag}_{\text{method}} \in \{\texttt{EigenNAS}, \texttt{DARTS}, \cdots\}$.
// conduct NAS algorithms
**if** $\texttt{Flag}_{\text{GuideNAS}} = \texttt{EigenNAS}$ **then**
    Guide NAS from $\mathcal{S}$ by our Eigen-NAS algorithm.
**else if** $\texttt{Flag}_{\text{GuideNAS}} = \texttt{DARTS}$ **then**
    Search neural network architectures from $\mathcal{S}$ using the DARTS algorithm.
**end if**
Output the optimal architecture $\{(\sigma_i^*)_{i=1}^{L-1}, (\alpha_i^*)_{i=1}^{L-2}\} \in \mathcal{S}$ with max probability.
// do neural network training via SGD
Gaussian initialization: $\boldsymbol{W}_l^{(1)} \sim \mathcal{N}(0, 1/m), l \in [L]$
Construct the neural network $f(\boldsymbol{x}; \boldsymbol{W}_l^{(1)})$ based on $\{(\sigma_i^*)_{i=1}^{L-1}, (\alpha_i^*)_{i=1}^{L-2}\}$
**for** $i = 1$ **to** $N$ **do**
    $\boldsymbol{W}^{(i+1)} = \boldsymbol{W}^{(i)} - \gamma \cdot \nabla_{\boldsymbol{W}} \ell\big(f(\boldsymbol{x}_i; \boldsymbol{W}^{(i)}) y_i\big)$.
**end for**
**Output** Randomly choose $\hat{\boldsymbol{W}}$ uniformly from $\big\{\boldsymbol{W}^{(1)}, \ldots, \boldsymbol{W}^{(N)}\big\}$.

---

**Assumption 2.** The activation function $\sigma : \mathbb{R} \to \mathbb{R}$ satisfies $\sigma \in L^2(\mathbb{R}, e^{-x^2/2}/\sqrt{2\pi})$, where $L^2(\mathbb{R}, g)$ denotes the square integrable function.

**Assumption 3.** We further assume that $\boldsymbol{x}$ is isotropic. i.e. $\mathbb{E}[\boldsymbol{x}\boldsymbol{x}^\top] = \mathbb{I}_d/d$, where the coefficient $1/d$ is to satisfy Assumption 1 at the same time.

**Remark:** The first assumption on normalized data is commonly used in practice and theory on over-parameterized neural networks [Du et al., 2019b,a, Allen-Zhu et al., 2019, Oymak and Soltanolkotabi, 2020, Malach et al., 2020]. The second assumption is general as the studied activation functions in Table 1 satisfy it. The third assumption is standard in statistics and machine learning [Vershynin, 2018, Hastie et al., 2022, Klimovsky, 2012, Yaskov, 2014]. It covers Gaussian data, and data uniformly spread on the sphere, commonly used in deep learning theory [Mei et al., 2021, Ghosh et al., 2022].

### 4.2 Recursive NTK for DNNs defined by Equation (1)

Recall that NTK [Jacot et al., 2018] under the infinite-width setting ($m \to \infty$) is:

$$K^{(L)}(\boldsymbol{x}, \widetilde{\boldsymbol{x}}) := \mathbb{E}_{\boldsymbol{W}} \left\langle \frac{\partial f(\boldsymbol{x}; \boldsymbol{W})}{\partial \boldsymbol{W}}, \frac{\partial f(\widetilde{\boldsymbol{x}}; \boldsymbol{W})}{\partial \boldsymbol{W}} \right\rangle,$$

where the NTK matrix for residual networks is derived by the following regular chain rule.

**Lemma 1.** *For any $l \in [3, L]$ and $s \in [2, L]$, denote*

$$\boldsymbol{G}^{(1)} = \boldsymbol{X}\boldsymbol{X}^\top, \quad \boldsymbol{A}^{(2)} = \boldsymbol{G}^{(2)} = 2\mathbb{E}_{\boldsymbol{w} \sim \mathcal{N}(\boldsymbol{0}, \mathbb{I}_d)}[\sigma_1(\boldsymbol{X}\boldsymbol{w})\sigma_1(\boldsymbol{X}\boldsymbol{w})^\top],$$

$$\boldsymbol{G}^{(l)} = 2\mathbb{E}_{\boldsymbol{w} \sim \mathcal{N}(\boldsymbol{0}, \mathbb{I}_N)}[\sigma_{l-1}(\sqrt{\boldsymbol{A}^{(l-1)}}\boldsymbol{w})\sigma_{l-1}(\sqrt{\boldsymbol{A}^{(l-1)}}\boldsymbol{w})^\top], \quad \boldsymbol{A}^{(l)} = \boldsymbol{G}^{(l)} + \alpha_{l-2}\boldsymbol{A}^{(l-1)},$$

$$\dot{\boldsymbol{G}}^{(s)} = 2\mathbb{E}_{\boldsymbol{w} \sim \mathcal{N}(\boldsymbol{0}, \mathbb{I}_N)}[\sigma_{s-1}'(\sqrt{\boldsymbol{A}^{(s-1)}}\boldsymbol{w})\sigma_{s-1}'(\sqrt{\boldsymbol{A}^{(s-1)}}\boldsymbol{w})^\top].$$

*Then the NTK for residual networks defined in Equation* (1) *can be written as*

$$\boldsymbol{K}^{(L)} = \boldsymbol{G}^{(L)} + \sum_{l=1}^{L-1} \boldsymbol{G}^{(l)} \circ \dot{\boldsymbol{G}}^{(l+1)} \circ (\dot{\boldsymbol{G}}^{(l+2)} + \alpha_l \mathbf{1}_{N \times N}) \circ \cdots \circ (\dot{\boldsymbol{G}}^{(L)} + \alpha_{L-2}\mathbf{1}_{N \times N}).$$

**Remark:** ($i$) Our NTK formula of ResNet differs from the one of Tirer et al. [2022], Huang et al. [2020], Belfer et al. [2021] in two critical ways: 1) each skip-layer in our model skips one fully-connected layer and one activation function, as opposed to the two-layer skip of previous works, 2) our formulation does not require every layer to have a parallel skip connection, which increases the flexibility of the network. Those differences also result in a different NTK matrix.
($ii$) Our NTK formulation covers different activation functions, and we adopt the same initialization (coefficient) on them to ensure fair/equal search in our NAS framework .

Lemma 1 covers both FC and residual neural networks, which facilitates the analysis of minimum eigenvalue of NTK under the unifying framework. If $\alpha_l = 0$ for $l \in [L-1]$, our NTK formulation for residual neural networks degenerates to that of a fully-connected neural network, and $\boldsymbol{A}^l$ and $\boldsymbol{G}^l$ become equal.

## 4.3 Minimum Eigenvalue of NTK for infinite-width

We are now ready to state the main result on the infinite-width neural network. We provide the upper and lower bounds of minimum eigenvalue of NTK for infinite-width neural network mixed with five different activation functions. The main differences between different activation functions are illustrated in Table 1.

**Theorem 1.** *For a DNN defined by Equation* (1) *and a not very large L, let $\boldsymbol{K}^{(L)}$ be the limiting NTK recursively defined in Lemma 1. Then, under Assumptions 1 and 3, when $N \geq \Omega(d^4)$, with probability at least $1 - e^{-d}$, we have*

$$\lambda_{\min}(\boldsymbol{K}^{(L)}) \geq 2\mu_1(\sigma_1)^2 \Theta(N/d) \prod_{p=3}^{L} \left( \beta_3(\sigma_{p-1}) + \alpha_{p-2} \right),$$

$$\lambda_{\min}(\boldsymbol{K}^{(L)}) \leq \frac{N}{d} \sum_{l=1}^{L} \left( \beta_1(\sigma_{l-1}) \prod_{p=2}^{l-1} \left( \beta_1(\sigma_{p-1}) + \alpha_{p-2} \right) \prod_{p=l+1}^{L} \left( \beta_2(\sigma_{p-1}) + \alpha_{p-2} \right) \right),$$

*where $\mu_1(\sigma_1)$ is the 1-st Hermite coefficient of the first layer activation function, and $\beta_1, \beta_2, \beta_3$ are three constants on various activation functions defined in Table 1.*

**Remark:** A not very large depth, e.g., $L \leq 10$, is often sufficient for the search phase in practical implementations [Liu et al., 2018, Dong et al., 2021]. In addition, existing NAS algorithms such as DARTS tend to have architectures with wide and shallow cell structure as suggested by Shu et al. [2020]. Theorem 1 shows the upper and lower bounds of the minimum eigenvalue of NTK under the mix of activation functions and skip connections. The following conclusions can be drawn from our result:

1. The bounds of the minimum eigenvalue depend significantly on the depth of the network $L$, the skip connections via $\alpha_p$, that makes the minimum eigenvalue increasing fast as $L$ and the number of skip connections increase. Besides, the minimum eigenvalue is also effected by activation functions via $\beta_1, \beta_2, \beta_3$. Nevertheless, the lower bound is independent of $\beta_1$ and $\beta_2$.
2. Different activation functions lead to different tendency (increase or decrease) on $\lambda_{\min}(\boldsymbol{K}^{(L)})$. As the depth increases, the lower bound $\lambda_{\min}(\boldsymbol{K}^{(L)})$ under ReLU remains unchanged, increases under LeakyReLU, and decreases when Sigmoid, Tanh or Swish applied, which brings in new findings when compared to the ReLU-network analysis of Nguyen et al. [2021]. For the upper bound for $\lambda_{\min}(\boldsymbol{K}^{(L)})$, we can see our results are positively correlated with the depth $L$.
3. One can see that $\mu_1(\sigma_1)$ is only related to the activation function of the first layer, which implies that the activation function in the first layer is very important as $\lambda_{\min}(\boldsymbol{K}^{(L)})$ largely depends on it.

## 4.4 Minimum Eigenvalue of NTK for finite-width

To study the finite-width, we firstly introduce the Jacobian of the network. Let $\boldsymbol{F} = [f(\boldsymbol{x}_1), \ldots, f(\boldsymbol{x}_N)]^T$. Then, the Jacobian $\boldsymbol{J}$ of $\boldsymbol{F}$ with respect to $\boldsymbol{W}$ is $\boldsymbol{J} = \left[ \frac{\partial \boldsymbol{F}}{\partial \text{vec}(\boldsymbol{W}_1)}, \ldots, \frac{\partial \boldsymbol{F}}{\partial \text{vec}(\boldsymbol{W}_L)} \right]$, where $\boldsymbol{J}$ have dimension $\mathbb{R}^{(((L-2) \times m + d + 1) \times m \times N)}$. The empirical Neural Tangent Kernel (NTK) matrix can be defined as $\bar{\boldsymbol{K}}^{(L)} = \boldsymbol{J}\boldsymbol{J}^\top = \sum_{l=1}^{L} \left[ \frac{\partial \boldsymbol{F}}{\partial \text{vec}(\boldsymbol{W}_l)} \right] \left[ \frac{\partial \boldsymbol{F}}{\partial \text{vec}(\boldsymbol{W}_l)} \right]^\top$.

Accordingly, we generalize Theorem 1 from the infinite-width to finite-width setting below.

**Theorem 2.** *For an L-layer network defined by Equation* (1)*, let $\boldsymbol{K}^{(L)} = \boldsymbol{J}\boldsymbol{J}^\top$ be the NTK matrix, and the weights of the network be initialized as $[\boldsymbol{W}_l]_{i,j} \sim \mathcal{N}(0, 1/m)$, for all $l \in [L]$. Under Assumptions 1 and 3, when $N \geq \Omega(d^4)$, then $\lambda_{\min}(\boldsymbol{J}\boldsymbol{J}^\top)$ can be bounded by*

$$\Theta\left( \frac{N}{d} \prod_{i=2}^{L-1} (\beta_3(\sigma_i) + \alpha_{i-1}) \right) \leq \lambda_{\min}(\boldsymbol{J}\boldsymbol{J}^\top) \leq \frac{N}{d} \sum_{k=0}^{L-1} \Theta\left( \prod_{i=k+2}^{L-1} (\beta_2(\sigma_i) + \alpha_{i-1}) \right),$$

*where the first inequality (lower bound) holds with probability at least $1 - e^{-d} - \sum_{l=1}^{L-1} \exp(-\Omega(m)) - \exp(-\Omega(1))$ and the second inequality (upper bound) holds with probability at least $1 - \sum_{l=1}^{L-1} \exp(-\Omega(m)) - \exp(-\Omega(1))$. The definitions of $\beta_2$, and $\beta_3$ are the same as those in Theorem 1.*

**Remark:** Theorem 2 achieves a similar result as Theorem 1 if the width $m$ is large.

## 4.5 Connection to Generalization Error Bound

Based on the aforementioned upper and lower bounds of the minimum eigenvalue of NTK under different settings, here we establish its relationship with the generalization error of DNNs. We provide a bound on the expected 0-1 error obtained by Algorithm 1.

**Theorem 3.** *Given a DNN defined by Equation* (1) *with $\boldsymbol{y} = (y_1, \ldots, y_N)^\top$ determined by Algorithm 1 with the step size of SGD $\gamma = \kappa C_1 \cdot \sqrt{\boldsymbol{y}^\top (\boldsymbol{K}^{(L)})^{-1} \boldsymbol{y}}/(m\sqrt{N})$ for some small enough absolute constant $\kappa$. Under Assumptions 1, 2 and 3, for any $\delta \in (0, e^{-1}]$ and a not very large L, if the width $m \geq \hat{m}$, where $\hat{m}$ depends on $\lambda_{\min}(\boldsymbol{K}^{(L)}), \delta, N$, and L, then with probability at least $1 - \delta$ over the randomness of $\boldsymbol{W}^{(1)}$, we obtain the following high probability bound:*

$$\mathbb{E}[\ell_{\mathcal{D}}^{0-1}(\hat{\boldsymbol{W}})] \leq \tilde{\mathcal{O}}\left(C_2 \sqrt{\frac{\boldsymbol{y}^\top \boldsymbol{y}}{\lambda_{\min}(\boldsymbol{K}^{(L)})N}}\right) + \mathcal{O}\left(\sqrt{\frac{\log(1/\delta)}{N}}\right),$$

*where $C_1 = \sqrt{L}/(3\mathrm{Lip}_{\max} + 1)^{L-1}$ and $C_2 = \sqrt{L}(3\mathrm{Lip}_{\max} + 1)^{L-1}$ are two constants depending only on L and $\mathrm{Lip}_{\max}$ is the maximum value of the Lipschitz constants of the all activation functions.*

**Remark:** Theorem 3 gives an algorithm-dependent generalization error bound of DNNs defined by Equation (1) trained with SGD with different activation functions and skip connections. If $m$ is large enough, the learning rate is infinitesimal, which means the generalization error bound mainly depends on the NTK matrix, similarly to Cao and Gu [2019], Du et al. [2019a]. Admittedly, our result is in an exponential increasing order of the depth. However, in practice, the depth $L$ during the search phrase is smaller than 20, or even 10 [Liu et al., 2018, Dong et al., 2021]. As we detail in Appendix E, our results extend previously known results.

According to Theorem 3, the generalization performance of DNNs is controlled by the minimum eigenvalue of the NTK matrix, which is in turn affected by different activation functions and skip connections, as discussed in Theorem 1. Apart from the NTK matrix itself, the condition $m \geq \hat{m}$ is also effected by different activation functions, which implies that the required minimum width is different in these cases.

## 4.6 Proof sketch

Our work extends the proofs of Nguyen and Mondelli [2020], Cao and Gu [2019] beyond ReLU, which is critical for enabling search across activations. The extension to other activation functions and skip connections is non-trivial due to non-linearity, inhomogeneity and nonmonotonicity.

To derive the upper and lower bounds on the minimum eigenvalue, we start from Lemma 1 on the NTK formula under the mixed activation functions and skip connections, and we transform the minimum eigenvalue estimation to the computation (estimation) of the bound $\boldsymbol{G}, \dot{\boldsymbol{G}}$ ($\lambda_{\min}(\boldsymbol{G})$). The infinite-width and finite-width are included in Appendix B and C respectively. For the upper bound, we estimate the diagonal elements of $\boldsymbol{G}$ and use the property that the minimum eigenvalue is less than the mean of the diagonal elements of a matrix to proof. For the lower bound, we use Hermite expansion and [Yaskov, 2014, Corollary 3.1]. Combining these results concludes the proof.

To derive the generalization error bounds, we need a series of lemmas (see Appendix D). If the input weights are close, the output of each neuron with any activation function does not change too much (see Lemma 7). If the initilizations are close, the neural network output $f(\boldsymbol{x}; \boldsymbol{W})$ is almost linear in $\boldsymbol{W}$ (see Lemma 8), and the loss function $\ell[y_i f(\boldsymbol{x}_i; \boldsymbol{W})]$ is almost a convex function of $\boldsymbol{W}$ for any $i \in [N]$ (see Lemma 9). Accordingly, the gradient and loss of the neural network can be upper bounded by Lemmas 10 and 11, respectively, which concludes the proof when combined with some relevant results [Cao and Gu, 2019, Allen-Zhu et al., 2019]. Further discussion on the differences is deferred to Appendix E.

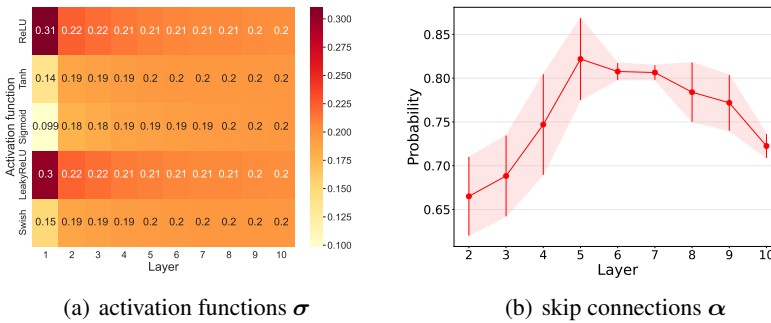

(a) activation functions $\boldsymbol{\sigma}$          (b) skip connections $\boldsymbol{\alpha}$

Figure 1: Architecture search results on activation functions indicated by the probability of $\boldsymbol{\sigma}$ in (a) and skip connections indicated by $\boldsymbol{\alpha}$ in (b). We notice that for each layer, ReLU and LeakyReLU are selected a the higher probability.

## 5 Numerical Validation

To validate our theoretical results, we conduct a series of experiments on NAS. Firstly, we simulate the NTK matrices under different depths in Appendix F.4 to verify the relationship between the minimum eigenvalue of NTK and the network depth $L$ in Theorem 1. In sec. 5.1 we use the DARTS algorithm [Liu et al., 2019b] to conduct experiments on activation function search and skip connection search under the search space of Equation (1). Finally, we use the minimum eigenvalue of NTK to guide the training of NAS on the benchmark NAS-Bench-201 [Dong and Yang, 2020], with a comparison of recent NAS algorithms. Additional experiments on NAS-Bench-101 [Ying et al., 2019] and transfer learning are deferred to Appendix F.5 and F.6.

### 5.1 DARTS experiment

In this section we employ a typical NAS algorithm, DARTS [Liu et al., 2019b], to assess our theoretical results on activation functions and skip connections. We select Fashion-MNIST [Xiao et al., 2017] as a standard benchmark. Details about Fashion-MNIST are shared in Appendix F.1.

**Search space and search strategy:** Our search space is defined by Equation (2) on skip connections, activation functions, and weight parameters. We follow the search strategy of Liu et al. [2019b] in a two-level scheme, one level is for weight parameter search $\boldsymbol{W}$ and the other level is for architecture search $\{\boldsymbol{\alpha}, \boldsymbol{\sigma}\}$, which results in the final optimal architecture $\{\boldsymbol{\alpha}^*, \boldsymbol{\sigma}^*, \boldsymbol{W}^*\}$. Different from Liu et al. [2019b], the activation function search and the skip connection search in our setting is independent. To obtain $\boldsymbol{\sigma}^*$, we use the softmax function to normalize the weights and choose the specific activation function with the highest probability in each layer. To obtain $\boldsymbol{\alpha}^*$, we initialize each entry $\alpha_l = 1/2$ ($l \in [L-2]$), constrain it to $[0,1]$ during training, and retain the skip connection when $\alpha_l^* > 1/2$.

**NAS Results:** We conduct the experiment via DARTS on a feedforward neural network with $L = 10$ and $m = 1024$, with 5 runs. After training, the probability of these activation functions and skip connections in each layer is reported in Figure 1(a) and 1(b), respectively. We have the following findings: Firstly, after the search process, LeakyReLU and ReLU are selected as the activations with the highest probability in each layer. This coincides with our theoretical results in Theorem 1. One minor difference is that the probability of LeakyReLU is slightly inferior to ReLU in practice. The reason behind this could be the sparsity of ReLU [de Dios and Bruna, 2020]. Secondly, in the first layer, we observe the largest difference on the probability of various activation functions. As the network becomes deeper, the differences decrease with the last layers having no difference between different activation functions. This phenomenon matches our theory well. To be specific, in Theorem 1, our result on the minimum eigenvalue largely depend on the first layer and its Hermite coefficient. Besides, this result also provides a justification on omitting the high-order terms while retaining the first layer activation terms. Thirdly, for the skip search result, we find that the skip connections are required in each layer when $L \leq 10$, as suggested by our theoretical results in Theorem 1. It also verifies the results of Zhou et al. [2020]. We expect that the skip connections might not be required in each layer for deep neural networks, since their capacity can already be enough [He et al., 2016]; but we defer the related study to a future work.

Table 2: Results on CIFAR-10, CIFAR-100 and ImageNet-16 as part of NAS-Bench-201. The best performance is highlighted by **bold**. The results of NASWOT, TE-NAS and KNAS are reported from the corresponding papers. The results of ResNet, NAS-RL and DARTS are reported in [Xu et al., 2021]. The results illustrate that Eigen-NAS outperforms the prior art in CIFAR-100 and Imagenet-16. In particular, Eigen-NAS outperforms KNAS in all three cases when the same number of top-$k$ architectures are selected, i.e., $k = 20$, and still achieves promising performance when smaller $k = 5$ used, which we attribute to the more precise minimum eigenvalue estimation.

| Type | Model/Algorithm | CIFAR-10 (%) | CIFAR-100 (%) | ImageNet-16 (%) |
|---|---|---|---|---|
| w/o Search | ResNet [He et al., 2016] | **93.97** | 70.86 | 42.63 |
| Search | NAS-RL [Zoph and Le, 2017] | 92.83 | 70.71 | 44.10 |
| Gradient | DARTS [Liu et al., 2019b] | 88.32 | 67.34 | 33.04 |
| Train-free | NASWOT [Mellor et al., 2021] | 92.96 | 70.03 | 44.43 |
| Train-free | TE-NAS [Chen et al., 2021] | 93.90 | 71.24 | 42.38 |
| Train-free | KNAS [Xu et al., 2021] ($k = 20$) | 93.38 | 70.78 | 44.63 |
| Train-free | NASI (T) [Shu et al., 2022] | $93.08 \pm 0.24$ | $69.51 \pm 0.59$ | $40.87 \pm 0.85$ |
| Train-free | NASI (4T) [Shu et al., 2022] | $93.55 \pm 0.10$ | $71.20 \pm 0.14$ | $44.84 \pm 1.41$ |
| Train-free | **Eigen-NAS** ($k = 20$) | $93.46 \pm 0.01$ | $\mathbf{71.42 \pm 0.63}$ | $\mathbf{45.54 \pm 0.04}$ |
| Train-free | **Eigen-NAS** ($k = 5$) | $93.43 \pm 0.08$ | $69.92 \pm 1.82$ | $45.53 \pm 0.06$ |

Interestingly, the search strategy favors the activation functions and the skip connections with larger minimum eigenvalue of NTK, which enjoy better generalization performance. This result also motivates us to study the following question: *can the minimum eigenvalue of NTK guide the search process in NAS?* We provide an affirmative answer in the next section with experimental validations.

## 5.2 NAS-Bench-201 Experiment

In this experiment, we use the minimum eigenvalue to guide NAS on NAS-Bench-201 [Dong and Yang, 2020]. Each experiment is repeated 5 times, while it can run on a single GPU in a few hours.

**Benchmark and baselines:** NAS-Bench-201 [Dong and Yang, 2020] is a commonly used benchmark for NAS algorithm evaluation, which includes three datasets: a) CIFAR-10 [Krizhevsky et al., 2014], b) CIFAR-100 [Krizhevsky et al., 2014] and c) ImageNet-16 [Chrabaszcz et al., 2017] for image classification. Details on the datasets exist in Appendix F.1. Apart from that, we evaluate the proposed approach with some baselines including ResNet, DARTS, RL based algorithm and some train-free algorithms.

**Algorithm procedure:** Our algorithm, called Eigen-NAS, also belongs in the train-free category. Eigen-NAS follows KNAS, which leverages the minimum eigenvalue of NTK to guide NAS. However, due to the $\mathcal{O}(N^3)$ time complexity of computing these eigenvalues, KNAS instead computes $\|\boldsymbol{K}\|_{\mathrm{F}}$. However, from the expression $\lambda_{\min}(\boldsymbol{K}) \leq \frac{1}{d} \sum_{i=1}^N K_{ii} \leq \|\boldsymbol{K}\|_{\mathrm{F}}$ we utilize the first inequality in Eigen-NAS to obtain a tighter (and more computationally efficient) bound to $\lambda_{\min}$. The computation cost of our method is $\mathcal{O}(N)$, which is less than computing the Frobenius norm ($\mathcal{O}(N^2)$). Sequentially, the top-$k$ best candidates architectures are chosen in KNAS and our Eigen-NAS, and then the best architecture is chosen by the validation error. Please refer to the results in Table 2. Due to the page limit, the algorithm is located in Appendix F.

**Results:** The experimental results in Table 2 verify that Eigen-NAS guided by the proposed metric above achieves the best performance on both the CIFAR-100 and ImageNet-16 datasets, and competitive performance on CIFAR-10, outperforming KNAS in all three cases when $k = 20$ for both methods. Even when we consider a smaller $k = 5$, Eigen-NAS can outperform KNAS, which we attribute to the more precise minimum eigenvalue estimation.

## 6 Conclusion

In this work, we explore the relationship between the minimum eigenvalue of NTK and neural architecture search. We derive upper and lower bounds on the minimum eigenvalues of NTK for (in)finite residual networks under different mixtures of activation functions, and establish a connection between the minimum eigenvalues and the generalization properties of the special search space: activation function and skip connection search of NAS. Our theoretical results on various activation functions and mixed activation cases can also be a tool for deep learning theory researchers to prove generic results rather than studying a single architecture, e.g., ReLU networks. In addition, we use

the minimum eigenvalue as a guide for the training of NAS in a train-free method, which greatly exceeds the efficiency of the classic NAS algorithm. When compared with existing train-free methods, our algorithm, called Eigen-NAS, achieves a higher accuracy. We posit that this will be useful for studying computationally efficient methods on NAS.

A core limitation is whether our proof framework can cover more general structures in NAS, such as the most commonly used convolutional neural networks (CNNs). Even though this seems possible, this is non-trivial due to the tensors that emerge. To be specific, it requires the element-recursive form of NTK matrices in Arora et al. [2019b] to be transformed into a global-recursive form (similar to Lemma 1), then analyze its minimum eigenvalue. Besides, the contraction operation of tensors, the locality and boundary effects of convolutional layer in CNNs make the analysis difficult. Therefore, we believe this is a topic on its own right. Another limitation of our work is that it does not analyze the various algorithms proposed for searching through the search space. We believe that a deeper understanding of such algorithms, such as DARTS can provide further insights into how to design improved search spaces. In addition, the upper and lower bounds of the minimum eigenvalues of the NTK matrices for different activation functions given by Theorem 1 have some overlaps, which means that our suggestions on activation functions selection based on these bound appear a bit vacuous in theory but still coincide with our experimental validations. Maybe, a tighter bound without overlap for different activation functions is needed to address this theoretical issue.

## Acknowledgements

We are also thankful to the reviewers for providing constructive feedback. Research was sponsored by the Army Research Office and was accomplished under Grant Number W911NF-19-1-0404. This work was supported by Hasler Foundation Program: Hasler Responsible AI (project number 21043). This work was supported by SNF project – Deep Optimisation of the Swiss National Science Foundation (SNSF) under grant number 200021_205011. This work was supported by Zeiss. This project has received funding from the European Research Council (ERC) under the European Union's Horizon 2020 research and innovation programme (grant agreement n° 725594 - time-data). Corresponding authors: Fanghui Liu and Zhenyu Zhu.

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
