## Appendix introduction

The Appendix is organized as follows:

- In Appendix A, we state the introductory notations and definitions.
- We prove Theorem 1 in Appendix B. We also provide the result when the residual network only has the same activation function.
- In Appendix C, we extend the results of infinitely width to finite-width and provide the proof for them.
- In Appendix D, we prove Theorem 3.
- In Appendix E, we discussion some key points of the proof and the motivation of the analysis.
- In Appendix F, we detail our experimental settings, our Eigen-NAS algorithm as used in sec. 5.2. We conduct additional numerical validations.
- Finally, in Appendix G, we discuss the societal impact of this work.

## A  Background

### A.1  Symbols and Notation

In the paper, vectors are indicated with bold small letters, matrices with bold capital letters. To facilitate the understanding of our work, we include the some core symbols and notation in Table 3.

Table 3: Core symbols and notations used in this project.

| Symbol | Dimension(s) | Definition |
|---|---|---|
| $\mathcal{N}(\mu, \sigma)$ | - | Gaussian distribution of mean $\mu$ and variance $\sigma$ |
| $1\{A\}$ | - | Indicator function for event $A$ |
| $[L]$ | - | Shorthand of $\{1, 2, \ldots, L\}$ |
| $\oplus$ | - | Direct sum |
| $\mathcal{O}, o, \Omega$ and $\Theta$ | - | Standard Bachmann–Landau order notation |
| $\circ$ | - | Element-wise hadamard product |
| $\|\boldsymbol{v}\|_2$ | - | Euclidean norms of vectors $\boldsymbol{v}$ |
| $\|\boldsymbol{M}\|_2$ | - | Spectral norms of matrices $\boldsymbol{M}$ |
| $\|\boldsymbol{M}\|_{\mathrm{F}}$ | - | Frobenius norms of matrices $\boldsymbol{M}$ |
| $\|\boldsymbol{M}\|_*$ | - | Nuclear norms of matrices $\boldsymbol{M}$ |
| $\lambda(\boldsymbol{M})$ | - | Eigenvalues of matrices $\boldsymbol{M}$ |
| $\boldsymbol{M}^{[l]}$ | - | $l$-th row of matrices $\boldsymbol{M}$ |
| $\boldsymbol{M}_{i,j}$ | - | $(i, j)$-th element of matrices $\boldsymbol{M}$ |
| $N$ | - | Size of the dataset |
| $d$ | - | Input size of the network |
| $L$ | - | Depth of the network |
| $m$ | - | Width of intermediate layer |
| $\alpha_l$ | $\mathbb{R}$ | A binary variable measures whether there is a skip connection in the $l$-th layer |
| $\sigma_l$ | - | The activation function of $l$-th layer |
| $\beta_1, \beta_2, \beta_3$ | $\mathbb{R}, \mathbb{R}, \mathbb{R}$ | Three constants defined in Table 1 |
| $\mu_i(\sigma)$ | $\mathbb{R}$ | The $i$-th Hermite coefficient of the activation function $\sigma$ |
| $\boldsymbol{x}_i$ | $\mathbb{R}^d$ | The $i$-th data point |
| $y_i$ | $\mathbb{R}$ | The $i$-th target vector |
| $\boldsymbol{W}_1$ | $\mathbb{R}^{m \times d}$ | Weight matrix for the input layer |
| $\boldsymbol{W}_l$ | $\mathbb{R}^{m \times m}$ | Weight matrix for the $l$-th hidden layer |
| $\boldsymbol{W}_L$ | $\mathbb{R}^{1 \times m}$ | Weight matrix for the output layer |

### A.1.1  Feature map

Here we define the core notation about feature maps that are required in the proof. Firstly, we define $\omega$-neighborhood to describe the difference between two matrices.

For any $\boldsymbol{W} \in \mathcal{W}$, we define its $\omega$-neighborhood as follows:

**Definition 1** ($\omega$-neighborhood)**.**

$$\mathcal{B}(\boldsymbol{W}, \omega) := \{\boldsymbol{W}' \in \mathcal{W} : \|\boldsymbol{W}_l' - \boldsymbol{W}_l\|_{\mathrm{F}} \leq \omega, \boldsymbol{\alpha}' = \boldsymbol{\alpha}, \boldsymbol{\sigma}' = \boldsymbol{\sigma}, l \in [L]\} \ .$$

Then we define $(\boldsymbol{D}_l)_{k,k} = \sigma_l'((\boldsymbol{W}_l \boldsymbol{f}_{l-1})_k)$ as the back-propagation matrix of the activation function. We use the notation $\widetilde{\boldsymbol{W}} \in \mathcal{B}(\boldsymbol{W}, \omega)$ to describe the relationship of the two matrices have $\omega$-neighborhood relationship.

In addition, we define the feature map of network and its perturbing matrix as follows:

**Definition 2.**

$$
\begin{aligned}
\widetilde{\boldsymbol{g}}_{i,1} &= \widetilde{\boldsymbol{W}}_1 \boldsymbol{x}_i\,, & \boldsymbol{g}_{i,1} &= \boldsymbol{W}_1 \boldsymbol{x}_i\,, & for\ i \in [N], \\
\widetilde{\boldsymbol{f}}_{i,1} &= \sigma_1(\widetilde{\boldsymbol{W}}_1 \boldsymbol{x}_i)\,, & \boldsymbol{f}_{i,1} &= \sigma_1(\boldsymbol{W}_1 \boldsymbol{x}_i)\,, & for\ i \in [N], \\
\widetilde{\boldsymbol{g}}_{i,l} &= \widetilde{\boldsymbol{W}}_l \widetilde{\boldsymbol{f}}_{i,l-1}\,, & \boldsymbol{g}_{i,l} &= \boldsymbol{W}_l \boldsymbol{f}_{i,l-1}\,, & for\ i \in [N]\ \text{and}\ l = 2, \ldots, L-1, \\
\widetilde{\boldsymbol{f}}_{i,l} &= \sigma_l(\widetilde{\boldsymbol{W}}_l \widetilde{\boldsymbol{f}}_{i,l-1}) + \alpha_{l-1} \widetilde{\boldsymbol{f}}_{i,l-1}\,, & \boldsymbol{f}_{i,l} &= \sigma_l(\boldsymbol{W}_l \boldsymbol{f}_{i,l-1}) + \alpha_{l-1} \boldsymbol{f}_{i,l-1}\,, & for\ i \in [N]\ \text{and}\ l = 2, \ldots, L-1\,.
\end{aligned}
$$

Let us define diagonal matrices $\widetilde{\boldsymbol{D}}_{i,l} \in \mathbb{R}^{m \times m}$ and $\boldsymbol{D}_{i,l} \in \mathbb{R}^{m \times m}$ by letting $(\widetilde{\boldsymbol{D}}_{i,l})_{k,k} = \sigma_l'((\widetilde{\boldsymbol{g}}_{i,l})_k)$ and $(\boldsymbol{D}_{i,l})_{k,k} = \sigma_l'((\boldsymbol{g}_{i,l})_k)$, $\forall k \in [m]$. Accordingly, we let $\hat{\boldsymbol{g}}_{i,l} = \widetilde{\boldsymbol{g}}_{i,l} - \boldsymbol{g}_{i,l}$, $\hat{\boldsymbol{f}}_{i,l} = \widetilde{\boldsymbol{f}}_{i,l} - \boldsymbol{f}_{i,l}$ and diagonal matrix $\hat{\boldsymbol{D}}_{i,l} = \widetilde{\boldsymbol{D}}_{i,l} - \boldsymbol{D}_{i,l}$.

### A.1.2 Other notations

For the Hadamard product of the matrices $\boldsymbol{X}_1, \boldsymbol{X}_2, \cdots, \boldsymbol{X}_r$ that share the same dimensions, we use the following abbreviation:

$$\bigcirc_{i=1}^{r}(\boldsymbol{X}_i) = \boldsymbol{X}_1 \circ \boldsymbol{X}_2 \circ \cdots \circ \boldsymbol{X}_r\,.$$

## B The bound of the minimum eigenvalues of NTK for infinite-width

We present the details of our results on sec. 4.3 in this section. Firstly, we provide the proof of Theorem 1 in Appendix B.2. Then in Appendix B.3 we provide the result when several activation functions exist alone.

### B.1 Proof of Lemma 1

Our proof mainly follows the results of Huang et al. [2020], but due to the different network structures, the proof process and results are slightly different. Moreover, we provide a matrix version results, which Huang et al. [2020] does not contain. For self-completeness, we include the proof here.

*Proof.* By Huang et al. [2020, Proposition 3], written as matrix form, we have:

$$
\begin{aligned}
\boldsymbol{A}^{(1)} &= \boldsymbol{X}\boldsymbol{X}^\top\,, \\
\boldsymbol{A}^{(2)} &= 2\mathbb{E}_{\boldsymbol{w} \sim \mathcal{N}(\boldsymbol{0}, \mathbb{I}_d)}[\sigma_1(\boldsymbol{X}\boldsymbol{w})\sigma_1(\boldsymbol{X}\boldsymbol{w})^\top]\,, \\
\boldsymbol{A}^{(l)} &= 2\mathbb{E}_{\boldsymbol{w} \sim \mathcal{N}(\boldsymbol{0}, \mathbb{I}_N)}[\sigma_{l-1}(\sqrt{\boldsymbol{A}^{(l-1)}}\boldsymbol{w})\sigma_{l-1}(\sqrt{\boldsymbol{A}^{(l-1)}}\boldsymbol{w})^\top] + \alpha_{l-2}\boldsymbol{A}^{(l-1)}\,.
\end{aligned}
$$

Note that, it is slightly different from the original result because the network structure is slightly different.

Let $\boldsymbol{G}^{(1)} = \boldsymbol{A}^{(1)}$, $\boldsymbol{G}^{(2)} = \boldsymbol{A}^{(2)}$ and $\boldsymbol{G}^{(l)} = 2\mathbb{E}_{\boldsymbol{w} \sim \mathcal{N}(\boldsymbol{0}, \mathbb{I}_N)}[\sigma_{l-1}(\sqrt{\boldsymbol{A}^{(l-1)}}\boldsymbol{w})\sigma_{l-1}(\sqrt{\boldsymbol{A}^{(l-1)}}\boldsymbol{w})^\top]$, then we have:

$$
\begin{aligned}
\boldsymbol{A}^{(1)} &= \boldsymbol{G}^{(1)} = \boldsymbol{X}\boldsymbol{X}^\top\,, \\
\boldsymbol{A}^{(2)} &= \boldsymbol{G}^{(2)} = 2\mathbb{E}_{\boldsymbol{w} \sim \mathcal{N}(\boldsymbol{0}, \mathbb{I}_d)}[\sigma_1(\boldsymbol{X}\boldsymbol{w})\sigma_1(\boldsymbol{X}\boldsymbol{w})^\top]\,, \\
\boldsymbol{G}^{(l)} &= 2\mathbb{E}_{\boldsymbol{w} \sim \mathcal{N}(\boldsymbol{0}, \mathbb{I}_N)}[\sigma_{l-1}(\sqrt{\boldsymbol{A}^{(l-1)}}\boldsymbol{w})\sigma_{l-1}(\sqrt{\boldsymbol{A}^{(l-1)}}\boldsymbol{w})^\top]\,, \\
\boldsymbol{A}^{(l)} &= \boldsymbol{G}^{(l)} + \alpha_{l-2}\boldsymbol{A}^{(l-1)}\,.
\end{aligned}
$$

According to Huang et al. [2020, Proposition 4], written as matrix form, we have:

$$\boldsymbol{K}^{(L)} = \sum_{l=1}^{L} \boldsymbol{G}^{(l)} \circ \dot{\boldsymbol{G}}^{(l+1)} \circ (\dot{\boldsymbol{G}}^{(l+2)} + \alpha_l \mathbf{1}_{N \times N}) \circ \cdots \circ (\dot{\boldsymbol{G}}^{(L)} + \alpha_{L-2} \mathbf{1}_{N \times N}),$$

where the $\dot{\boldsymbol{G}}^{(s)}$ satisfy that $\dot{\boldsymbol{G}}^{(s)} = 2\mathbb{E}_{\boldsymbol{w} \sim \mathcal{N}(\mathbf{0}, \mathbb{I}_N)}[\sigma'_{s-1}(\sqrt{\boldsymbol{A}^{(s-1)}}\boldsymbol{w})\sigma'_{s-1}(\sqrt{\boldsymbol{A}^{(s-1)}}\boldsymbol{w})^\top]$. Combining the above results, we finish the proof.

$\square$

## B.2 Proof of Theorem 1

In this part, we present the proof of Theorem 1. Differently from Oymak and Soltanolkotabi [2020], our result allows for activation functions search in each layer.

Before we prove Theorem 1, we provide some propositions that are helpful to our proof. To facilitate the writing of the proof, let $\alpha_0 := 0$.

**Proposition 1.** *When $\sigma_1$ is Tanh, the remaining layers are with LeakyReLU and for $l \in [L-2]$, $\alpha_l = 1$, the quantity $G_{ii}^{(l)}$ has the largest upper bound:*

$$G_{ii}^{(l)} \leq \begin{cases} 1 & \text{if } l = 1 \\ 2(2+\eta^2)^{l-2} & \text{if } l \geq 2 \,. \end{cases} \tag{3}$$

*We set $G_{\max} = 2(2+\eta^2)^{L-2}$ as the upper bound of $G_{ii}^{(L)}$.*

*Proof.* To prove our result, we need bound $\boldsymbol{G}^{(l)}$ under different activation functions. We summarize them as below.

When $\sigma_{l-1}$ is ReLU:

$$
\begin{aligned}
G_{ii}^{(l)} = 2\mathbb{E}_{w \sim \mathcal{N}(0, A_{ii}^{(l-1)})}[\sigma_{l-1}(w)^2] &= \int_{-\infty}^{\infty} \frac{2}{\sqrt{2\pi A_{ii}^{(l-1)}}} e^{-\frac{x^2}{2A_{ii}^{(l-1)}}} \max(0, x)^2 \mathrm{d}x \\
&= \int_0^{\infty} \frac{2}{\sqrt{2\pi A_{ii}^{(l-1)}}} e^{-\frac{x^2}{2A_{ii}^{(l-1)}}} x^2 \mathrm{d}x \\
&= A_{ii}^{(l-1)} \,.
\end{aligned} \tag{4}
$$

When $\sigma_{l-1}$ is LeakyReLU:

$$
\begin{aligned}
G_{ii}^{(l)} = 2\mathbb{E}_{w \sim \mathcal{N}(0, A_{ii}^{(l-1)})}[\sigma_{l-1}(w)^2] &= \int_{-\infty}^{\infty} \frac{2}{\sqrt{2\pi A_{ii}^{(l-1)}}} e^{-\frac{x^2}{2A_{ii}^{(l-1)}}} \max(\eta x, x)^2 \mathrm{d}x \\
&= \int_0^{\infty} \frac{2}{\sqrt{2\pi A_{ii}^{(l-1)}}} e^{-\frac{x^2}{2A_{ii}^{(l-1)}}} x^2 \mathrm{d}x + \int_{-\infty}^0 \frac{2}{\sqrt{2\pi A_{ii}^{(l-1)}}} e^{-\frac{x^2}{2A_{ii}^{(l-1)}}} \eta^2 x^2 \mathrm{d}x \\
&= (1+\eta^2) A_{ii}^{(l-1)} \,.
\end{aligned} \tag{5}
$$

When $\sigma_{l-1}$ is Sigmoid:

$$
\begin{aligned}
G_{ii}^{(l)} = 2\mathbb{E}_{w \sim \mathcal{N}(0, A_{ii}^{(l-1)})}[\sigma_{l-1}(w)^2] &= \int_{-\infty}^{\infty} \frac{2}{\sqrt{2\pi A_{ii}^{(l-1)}}} e^{-\frac{x^2}{2A_{ii}^{(l-1)}}} f_{\text{Sigmoid}}(x)^2 \mathrm{d}x \\
&\leq \int_{-\infty}^{\infty} \frac{2}{\sqrt{2\pi A_{ii}^{(l-1)}}} e^{-\frac{x^2}{2A_{ii}^{(l-1)}}} (\frac{1}{2})^2 \mathrm{d}x \\
&= \frac{1}{2} \,.
\end{aligned} \tag{6}
$$

| $\sigma_{l-1}$ | ReLU | LeakyReLU | Sigmoid | Tanh | Swish |
|---|---|---|---|---|---|
| Upper bound | 1 | $1+\eta^2$ | $\frac{1}{8}$ | 2 | 1 |
| Lower bound | 1 | $1+\eta^2$ | $\left(\frac{1}{2}-\frac{1}{2\sqrt{1+\frac{G_{\max}}{4}}}\right)\frac{1}{G_{\max}}$ | $\left(2-\frac{2}{\sqrt{1+G_{\max}}}\right)\frac{1}{G_{\max}}$ | $\frac{1}{2}$ |

Table 4: Upper and lower bounds for $A_{ii}^{(l)}/A_{ii}^{(l-1)} - \alpha_{l-2}$ for different activation functions $\sigma_{l-1}$ and the binary variable $\alpha_{l-2} \in \{0,1\}$ indicates whether $(l-1)$-th layer has a skip connection or not.

When $\sigma_{l-1}$ is Tanh:

$$
\begin{aligned}
G_{ii}^{(l)} = 2\mathbb{E}_{w\sim\mathcal{N}(0,A_{ii}^{(l-1)})}[\sigma_{l-1}(w)^2] &= \int_{-\infty}^{\infty} \frac{2}{\sqrt{2\pi A_{ii}^{(l-1)}}} e^{-\frac{x^2}{2A_{ii}^{(l-1)}}} f_{\text{Tanh}}(x)^2 \mathrm{d}x \\
&\leq \int_{-\infty}^{\infty} \frac{2}{\sqrt{2\pi A_{ii}^{(l-1)}}} e^{-\frac{x^2}{2A_{ii}^{(l-1)}}} \mathrm{d}x \\
&= 2\,.
\end{aligned}
\tag{7}
$$

When $\sigma_{l-1}$ is Swish:

$$
\begin{aligned}
G_{ii}^{(l)} = 2\mathbb{E}_{w\sim\mathcal{N}(0,A_{ii}^{(l-1)})}[\sigma_{l-1}(w)^2] &= \int_{-\infty}^{\infty} \frac{2}{\sqrt{2\pi A_{ii}^{(l-1)}}} e^{-\frac{x^2}{2A_{ii}^{(l-1)}}} f_{\text{Swish}}(x)^2 \mathrm{d}x \\
&= \int_{-\infty}^{\infty} \frac{2}{\sqrt{2\pi A_{ii}^{(l-1)}}} e^{-\frac{x^2}{2A_{ii}^{(l-1)}}} \frac{x^2}{(1+e^{-x})^2} \mathrm{d}x \\
&= \int_{0}^{\infty} \frac{2}{\sqrt{2\pi A_{ii}^{(l-1)}}} e^{-\frac{x^2}{2A_{ii}^{(l-1)}}} x^2 \times \left(\frac{1}{(1+e^{-x})^2} + \frac{1}{(1+e^{x})^2}\right) \mathrm{d}x \\
&\leq \int_{0}^{\infty} \frac{2}{\sqrt{2\pi A_{ii}^{(l-1)}}} e^{-\frac{x^2}{2A_{ii}^{(l-1)}}} x^2 \mathrm{d}x \\
&= A_{ii}^{(l-1)}\,.
\end{aligned}
\tag{8}
$$

Combining Equations (4) to (8) with Lemma 1 we draw the conclusion and finish the proof. $\qquad\square$

**Proposition 2.** *The relationship between $A_{ii}^{(l)}$ and $A_{ii}^{(l-1)}$ for different activation functions can be summarized as Table 4 according to the difference of $\sigma_{l-1}$.*

*Proof.* To prove our result, we need to bound the ratio $A_{ii}^{(l)}/A_{ii}^{(l-1)}$ for different activation functions. We illustrate how this is achieved in different cases below:

For $l \geq 2$:

When $\sigma_{l-1}$ is ReLU by Equation (4) we have:

$$
A_{ii}^{(l)} = G_{ii}^{(l)} + \alpha_{l-2}A_{ii}^{(l-1)} = (1+\alpha_{l-2})A_{ii}^{(l-1)}\,.
\tag{9}
$$

When $\sigma_{l-1}$ is LeakyReLU by Equation (5) we have:

$$
A_{ii}^{(l)} = G_{ii}^{(l)} + \alpha_{l-2}A_{ii}^{(l-1)} = (1+\alpha_{l-2}+\eta^2)A_{ii}^{(l-1)}\,.
\tag{10}
$$

When $\sigma_{l-1}$ is Swish, $G_{ii}^{(l)}$ can be upper by Equation (8) and lower bounded by:

$$G_{ii}^{(l)} = 2\mathbb{E}_{w \sim \mathcal{N}(0, A_{ii}^{(l-1)})}[\sigma_{l-1}(w)^2] = \int_{-\infty}^{\infty} \frac{2}{\sqrt{2\pi A_{ii}^{(l-1)}}} e^{-\frac{x^2}{2A_{ii}^{(l-1)}}} f_{\text{Swish}}(x)^2 dx$$

$$= \int_{-\infty}^{\infty} \frac{2}{\sqrt{2\pi A_{ii}^{(l-1)}}} e^{-\frac{x^2}{2A_{ii}^{(l-1)}}} \frac{x^2}{(1+e^{-x})^2} dx$$

$$= \int_{0}^{\infty} \frac{2}{\sqrt{2\pi A_{ii}^{(l-1)}}} e^{-\frac{x^2}{2A_{ii}^{(l-1)}}} x^2 \times \left( \frac{1}{(1+e^{-x})^2} + \frac{1}{(1+e^x)^2} \right) dx \qquad (11)$$

$$\geq \int_{0}^{\infty} \frac{2}{\sqrt{2\pi A_{ii}^{(l-1)}}} e^{-\frac{x^2}{2A_{ii}^{(l-1)}}} x^2 \times \frac{1}{2} dx$$

$$= \frac{1}{2} A_{ii}^{(l-1)},$$

which implies:

$$\left( \frac{1}{2} + \alpha_{l-2} \right) A_{ii}^{(l-1)} \leq A_{ii}^{(l)} \leq (1 + \alpha_{l-2}) A_{ii}^{(l-1)}. \qquad (12)$$

When $\sigma_{l-1}$ is Sigmoid, $G_{ii}^{(l)}$ can be upper by:

$$G_{ii}^{(l)} = 2\mathbb{E}_{w \sim \mathcal{N}(0, A_{ii}^{(l-1)})}[\sigma_{l-1}(w)^2] = \int_{-\infty}^{\infty} \frac{2}{\sqrt{2\pi A_{ii}^{(l-1)}}} e^{-\frac{x^2}{2A_{ii}^{(l-1)}}} f_{\text{Sigmoid}}(x)^2 dx$$

$$\leq \int_{-\infty}^{\infty} \frac{2}{\sqrt{2\pi A_{ii}^{(l-1)}}} e^{-\frac{x^2}{2A_{ii}^{(l-1)}}} \left( \frac{1}{4} - e^{-\frac{x^2}{4}} \right) dx = \frac{1}{2} - \frac{1}{2\sqrt{1 + \frac{A_{ii}^{(l-1)}}{2}}} \qquad (13)$$

$$\leq \frac{A_{ii}^{(l-1)}}{8}, \qquad \text{holds for } x \geq 0.$$

Then $G_{ii}^{(l)}$ can be lower bounded by:

$$G_{ii}^{(l)} = 2\mathbb{E}_{w \sim \mathcal{N}(0, A_{ii}^{(l-1)})}[\sigma_{l-1}(w)^2] = \int_{-\infty}^{\infty} \frac{2}{\sqrt{2\pi A_{ii}^{(l-1)}}} e^{-\frac{x^2}{2A_{ii}^{(l-1)}}} f_{\text{Sigmoid}}(x)^2 dx$$

$$\geq \int_{-\infty}^{\infty} \frac{2}{\sqrt{2\pi A_{ii}^{(l-1)}}} e^{-\frac{x^2}{2A_{ii}^{(l-1)}}} \left( \frac{1}{4} - e^{-\frac{x^2}{8}} \right) dx$$

$$= \frac{1}{2} - \frac{1}{2\sqrt{1 + \frac{A_{ii}^{(l-1)}}{4}}} \qquad (14)$$

$$\geq \left( \frac{1}{2} - \frac{1}{2\sqrt{1 + \frac{G_{\max}}{4}}} \right) \frac{A_{ii}^{(l-1)}}{G_{\max}},$$

where we use the fact that the penultimate line is a concave function with respect to $A_{ii}^{(l-1)}$. When $A_{ii}^{(l-1)} = 0$, the function value is 0. That means $G_{ii}^{(l)}/A_{ii}^{(l-1)}$ obtains the minimum value at $G_{ii}^{(l)} = G_{\max}$. Combined with Equation (3), we get the last inequality.

Then, we have:

$$\left( \left[ \frac{1}{2} - \frac{1}{2\sqrt{1 + \frac{G_{\max}}{4}}} \right] \frac{1}{G_{\max}} + \alpha_{l-2} \right) A_{ii}^{(l-1)} \leq A_{ii}^{(l)} \leq \left( \frac{1}{8} + \alpha_{l-2} \right) A_{ii}^{(l-1)}. \qquad (15)$$

| $\sigma_{l-1}$ | ReLU | LeakyReLU | Sigmoid | Tanh | Swish |
|---|---|---|---|---|---|
| Upper bound for $\dot{G}_{ii}^{(l)}$ | 1 | $1+\eta^2$ | $1/8$ | 2 | 1.22 |
| Lower bound for $\dot{G}_{ii}^{(l)}$ | 1 | $1+\eta^2$ | $f_{\mathrm{S}}(G_{\max})$ | $f_{\mathrm{T}}(G_{\max})$ | $1/2$ |

Table 5: Upper and lower bounds for $\dot{G}_{ii}^{(l)}$ for different activation function $\sigma_{l-1}$.

When $\sigma_{l-1}$ is Tanh, $G_{ii}^{(l)}$ can be upper bounded by:

$$
\begin{aligned}
G_{ii}^{(l)} = 2\mathbb{E}_{w\sim\mathcal{N}(0,A_{ii}^{(l-1)})}[\sigma_{l-1}(w)^2] &= \int_{-\infty}^{\infty} \frac{2}{\sqrt{2\pi A_{ii}^{(l-1)}}} e^{-\frac{x^2}{2A_{ii}^{(l-1)}}} f_{\mathrm{Tanh}}(x)^2 \mathrm{d}x \\
&\leq \int_{-\infty}^{\infty} \frac{2}{\sqrt{2\pi A_{ii}^{(l-1)}}} e^{-\frac{x^2}{2A_{ii}^{(l-1)}}} (1-e^{-x^2})\mathrm{d}x = 2 - \frac{2}{\sqrt{1+2A_{ii}^{(l-1)}}} \\
&\leq 2A_{ii}^{(l-1)}, \qquad \text{holds for } x \geq 0.
\end{aligned}
\tag{16}
$$

The lower bound is:

$$
\begin{aligned}
G_{ii}^{(l)} = 2\mathbb{E}_{w\sim\mathcal{N}(0,A_{ii}^{(l-1)})}[\sigma_{l-1}(w)^2] &= \int_{-\infty}^{\infty} \frac{2}{\sqrt{2\pi A_{ii}^{(l-1)}}} e^{-\frac{x^2}{2A_{ii}^{(l-1)}}} f_{\mathrm{Tanh}}(x)^2 \mathrm{d}x \\
&\geq \int_{-\infty}^{\infty} \frac{2}{\sqrt{2\pi A_{ii}^{(l-1)}}} e^{-\frac{x^2}{2A_{ii}^{(l-1)}}} (1-e^{-\frac{x^2}{2}})\mathrm{d}x \\
&= 2 - \frac{2}{\sqrt{1+A_{ii}^{(l-1)}}} \\
&\geq \left(2 - \frac{2}{\sqrt{1+G_{\max}}}\right)\frac{A_{ii}^{(l-1)}}{G_{\max}}.
\end{aligned}
\tag{17}
$$

Similar to the Sigmoid, the penultimate line is an concave function with respect to $A_{ii}^{(l-1)}$. When $A_{ii}^{(l-1)} = 0$, the function value is 0. That means $G_{ii}^{(l)}/A_{ii}^{(l-1)}$ obtains the minimum value at $G_{ii}^{(l)} = G_{\max}$. Combined with Equation (3), we get the last inequality.

Then, we have:

$$
\left(\left[2 - \frac{2}{\sqrt{1+G_{\max}}}\right]\frac{1}{G_{\max}} + \alpha_{l-2}\right)A_{ii}^{(l-1)} \leq A_{ii}^{(l)} \leq (2+\alpha_{l-2})A_{ii}^{(l-1)}.
\tag{18}
$$

According to Equations (9), (10), (12), (15) and (18), we can summarized the results about bound of $A_{ii}^{(l)}/A_{ii}^{(l-1)}$ in Table 4.

$\qquad\qquad\qquad\qquad\qquad\qquad\qquad\qquad\qquad\qquad\qquad\qquad\qquad\qquad\qquad\qquad\qquad$ $\square$

**Proposition 3.** *The bound of $\dot{G}_{ii}^{(l)}$ with respect to different activation function $\sigma_{l-1}$ can be summarized in Table 5.*

*Proof.* To prove our result, we need to bound $\dot{G}_{ii}^{(l)}$ with respect to different activation function $\sigma_{l-1}$ as follows.

When $\sigma_{l-1}$ is ReLU:

$$
\begin{aligned}
\dot{G}_{ii}^{(l)} = 2\mathbb{E}_{w\sim\mathcal{N}(0,A_{ii}^{(l-1)})}[\sigma_{l-1}'(w)^2] &= \int_{0}^{\infty} \frac{2}{\sqrt{2\pi A_{ii}^{(l-1)}}} e^{-\frac{x^2}{2A_{ii}^{(l-1)}}} \mathrm{d}x \\
&= 1.
\end{aligned}
\tag{19}
$$

When $\sigma_{l-1}$ is LeakyReLU:

$$\dot{G}_{ii}^{(l)} = 2\mathbb{E}_{w\sim\mathcal{N}(0,A_{ii}^{(l-1)})}[\sigma'_{l-1}(w)^2]$$

$$= \int_0^\infty \frac{2}{\sqrt{2\pi A_{ii}^{(l-1)}}} e^{-\frac{x^2}{2A_{ii}^{(l-1)}}}\,\mathrm{d}x + \int_{-\infty}^0 \frac{2}{\sqrt{2\pi A_{ii}^{(l-1)}}} e^{-\frac{x^2}{2A_{ii}^{(l-1)}}}\eta^2\mathrm{d}x \qquad (20)$$

$$= 1 + \eta^2\,.$$

When $\sigma_{l-1}$ is Sigmoid, according to the monotonicity of the $f_{\mathrm{S}}$, we have:

$$f_{\mathrm{S}}(G_{\max}) \leq \dot{G}_{ii}^{(l)} = 2\mathbb{E}_{w\sim\mathcal{N}(0,A_{ii}^{(l-1)})}[\sigma'_{l-1}(w)^2] = f_{\mathrm{S}}(A_{ii}^{(l-1)}) \leq f_{\mathrm{S}}(0) \leq \frac{1}{8}\,. \qquad (21)$$

When $\sigma_{l-1}$ is Tanh, according to the monotonicity of the $f_{\mathrm{T}}$, we have:

$$f_{\mathrm{T}}(G_{\max}) \leq \dot{G}_{ii}^{(l)} = 2\mathbb{E}_{w\sim\mathcal{N}(0,A_{ii}^{(l-1)})}[\sigma'_{l-1}(w)^2] = f_{\mathrm{T}}(A_{ii}^{(l-1)}) \leq f_{\mathrm{T}}(0) \leq 2\,. \qquad (22)$$

When $\sigma_{l-1}$ is Swish, The quantity $\dot{G}_{ii}^{(l)}$ can be upper bounded by:

$$\dot{G}_{ii}^{(l)} = 2\mathbb{E}_{w\sim\mathcal{N}(0,A_{ii}^{(l-1)})}[\sigma'_{l-1}(w)^2] = \int_{-\infty}^\infty \frac{2}{\sqrt{2\pi A_{ii}^{(l-1)}}} e^{-\frac{x^2}{2A_{ii}^{(l-1)}}} f'_{\mathrm{Swish}}(x)^2\mathrm{d}x$$

$$\leq \int_0^\infty \frac{2}{\sqrt{2\pi A_{ii}^{(l-1)}}} e^{-\frac{x^2}{2A_{ii}^{(l-1)}}} \left(\sup_x f'_{\mathrm{Swish}}(x)\right)^2\mathrm{d}x$$

$$+ \int_{-\infty}^0 \frac{2}{\sqrt{2\pi A_{ii}^{(l-1)}}} e^{-\frac{x^2}{2A_{ii}^{(l-1)}}} \left(\inf_x f'_{\mathrm{Swish}}(x)\right)^2\mathrm{d}x \qquad (23)$$

$$= \left(\inf_x f'_{\mathrm{Swish}}(x)\right)^2 + \left(\sup_x f'_{\mathrm{Swish}}(x)\right)^2$$

$$\leq 1.22\,,$$

where the last inequality holds by $1.099 < \sup_x f'_{\mathrm{Swish}}(x) < 1.1$ and $-0.1 < \inf_x f'_{\mathrm{Swish}}(x) < -0.099$.

Then the quantity $\dot{G}_{ii}^{(l)}$ can be lower bounded by:

$$\dot{G}_{ii}^{(l)} = 2\mathbb{E}_{w\sim\mathcal{N}(0,A_{ii}^{(l-1)})}[\sigma'_{l-1}(w)^2] = \int_{-\infty}^\infty \frac{2}{\sqrt{2\pi A_{ii}^{(l-1)}}} e^{-\frac{x^2}{2A_{ii}^{(l-1)}}} f'_{\mathrm{Swish}}(x)^2\mathrm{d}x$$

$$= \int_0^\infty \frac{2}{\sqrt{2\pi A_{ii}^{(l-1)}}} e^{-\frac{x^2}{2A_{ii}^{(l-1)}}} \left(f'_{\mathrm{Swish}}(x)^2 + f'_{\mathrm{Swish}}(-x)^2\right)\mathrm{d}x \qquad (24)$$

$$\geq \int_0^\infty \frac{2}{\sqrt{2\pi A_{ii}^{(l-1)}}} e^{-\frac{x^2}{2A_{ii}^{(l-1)}}} \frac{1}{2}\mathrm{d}x$$

$$= \frac{1}{2}\,.$$

Combining Equations (19) to (24), we can summarize the results about bound of $\dot{G}_{ii}^{(l)}$ in Table 5.

$\square$

**Proposition 4.** *Let $\boldsymbol{X} \in \mathbb{R}^{N\times d}$ be an matrix that every row $\boldsymbol{x}_i$ sampled from data distribution $\mathcal{D}_X$. when $N \geq \Omega(d^4)$, with probability at least $1 - e^{-d}$, we have*

$$\lambda_{\min}(\boldsymbol{X}^\top\boldsymbol{X}) \geq \frac{N}{d} - 9N^{2/3}d^{1/3} = \Theta(N/d)\,.$$

*Proof.* Firstly, we can compute that, for any $\alpha \in (0, 2]$,

$$\sup_{\|\boldsymbol{v}\|_2=1} \mathbb{E}\left|\left\langle \sqrt{d}\boldsymbol{x}_i, \boldsymbol{v}\right\rangle\right|^{2+\alpha} = \left\|\sqrt{d}\boldsymbol{x}_i\right\|_2^{2+\alpha} = d^{(2+\alpha)/2} < \infty .$$

Then from Assumption 3, we have $\sqrt{d}\boldsymbol{x}_i$ is an isotropic random vector in $\mathbb{R}^d$.

According to Yaskov [2014, Corollary 3.1], if we choose $\alpha = 1$, then with probability at least $1 - e^{-d}$ we have:

$$\begin{aligned}
\lambda_{\min}(\frac{d}{N}\boldsymbol{X}^\top \boldsymbol{X}) &\geq 1 - C_\alpha(\frac{d}{N})^{1/3} \\
&= 1 - 9(L_p(\alpha))^{2/(2+\alpha)}(\frac{d}{N})^{1/3} \\
&= 1 - 9(d^{(2+\alpha)/2})^{2/(2+\alpha)}(\frac{d}{N})^{1/3} \\
&= 1 - \frac{9d^{4/3}}{N^{1/3}} .
\end{aligned}$$

That is, with probability at least $1 - e^{-d}$ we have:

$$\lambda_{\min}(\boldsymbol{X}^\top \boldsymbol{X}) \geq \frac{N}{d} - 9N^{2/3}d^{1/3} = \Theta(N/d) .$$

$\square$

Now we are ready to prove Theorem 1.

*Proof of Theorem 1.* Now we are ready to present the estimation on $\lambda_{\min}(\boldsymbol{K}^{(L)})$ as below.

From Lemma 1, we have the NTK formulation for residual neural networks:

$$\boldsymbol{K}^{(L)} = \boldsymbol{G}^{(L)} + \sum_{l=1}^{L-1} \boldsymbol{G}^{(l)} \circ (\dot{\boldsymbol{G}}^{(l+1)} + \alpha_l \mathbf{1}_{N\times N}) \circ \cdots \circ (\dot{\boldsymbol{G}}^{(L)} + \alpha_{L-1}\mathbf{1}_{N\times N}) .$$

$$\boldsymbol{K}^{(L)} = \boldsymbol{G}^{(L)} + \sum_{l=1}^{L-1} \boldsymbol{G}^{(l)} \circ \dot{\boldsymbol{G}}^{(l+1)} \circ (\dot{\boldsymbol{G}}^{(l+2)} + \alpha_l \mathbf{1}_{N\times N}) \circ \cdots \circ (\dot{\boldsymbol{G}}^{(L)} + \alpha_{L-2}\mathbf{1}_{N\times N}) .$$

It is clear that all the matrices $\boldsymbol{G}^{(l)}$, $\dot{\boldsymbol{G}}^{(l)}$ are positive semi-definite (PSD), then $(\dot{\boldsymbol{G}}^{(l+1)} + \alpha_l \mathbf{1}_{N\times N})$ are also PSD. For two PSD matrices $\boldsymbol{P}, \boldsymbol{Q} \in \mathbb{R}^{N\times N}$, it holds $\lambda_{\min}(\boldsymbol{P} \circ \boldsymbol{Q}) \geq \lambda_{\min}(\boldsymbol{P}) \min_{i\in[N]} Q_{ii}$ [Schur, 1911]. Accordingly, we have:

$$\lambda_{\min}(\boldsymbol{K}^{(L)}) \geq \sum_{l=1}^{L} \lambda_{\min}(\boldsymbol{G}^{(l)}) \min_{i\in[N]} \prod_{p=l+1}^{L} \left(\dot{G}_{ii}^{(p)} + \alpha_{p-2}\right) .$$

Then we bound $\lambda_{\min}(\boldsymbol{G}^{(2)})$:

$$\begin{aligned}
\lambda_{\min}(\boldsymbol{G}^{(2)}) &= \lambda_{\min}\left( 2\mathbb{E}_{\boldsymbol{w}\sim\mathcal{N}(0,\mathbb{I}_d)}[\sigma_1(\boldsymbol{X}\boldsymbol{w})\sigma_1(\boldsymbol{X}\boldsymbol{w})^\top] \right) \\
&= 2\lambda_{\min}\left( \sum_{s=0}^{\infty} \mu_s(\sigma_1)^2 \bigcirc_{i=1}^{s} (\boldsymbol{X}\boldsymbol{X}^\top) \right) \quad \text{[Nguyen et al., 2021, Lemma D.3]} \\
&\geq 2\mu_1(\sigma_1)^2 \lambda_{\min}(\boldsymbol{X}\boldsymbol{X}^\top) .
\end{aligned}$$

As a reminder, the symbol $\bigcirc$ denotes the Hadamard product, which is defined in Appendix A.1.2.

We know that when $N > d$, $\boldsymbol{X}\boldsymbol{X}^\top$ and $\boldsymbol{X}^\top\boldsymbol{X}$ have the same non-zero eigenvalues. Then according to Proposition 4, with probability at least $1 - e^{-d}$ we have:

$$\lambda_{\min}(\boldsymbol{G}^{(2)}) \geq 2\mu_1(\sigma_1)^2(\frac{N}{d} - 9N^{2/3}d^{1/3}) = 2\mu_1(\sigma_1)^2\Theta(N/d)\,.$$

That means, with probability at least $1 - e^{-d}$ we have the lower bound of $\lambda_{\min}(\boldsymbol{K}^{(L)})$:

$$\lambda_{\min}(\boldsymbol{K}^{(L)}) \geq \sum_{l=1}^{L} \lambda_{\min}(\boldsymbol{G}^{(l)}) \min_{i \in [N]} \prod_{p=l+1}^{L} (\dot{G}_{ii}^{(p)} + \alpha_{p-2}) \geq \lambda_{\min}(\boldsymbol{G}^{(2)}) \min_{i \in [N]} \prod_{p=3}^{L}(\dot{G}_{ii}^{(p)} + \alpha_{p-2})$$

$$\geq 2\mu_1(\sigma_1)^2\Theta(N/d) \min_{i \in [N]} \prod_{p=3}^{L}(\dot{G}_{ii}^{(p)} + \alpha_{p-2})\,.$$

$$(25)$$

According to Table 1 and Table 5, we have:

$$\prod_{p=l+1}^{L} (\dot{G}_{ii}^{(p)} + \alpha_{p-2}) \leq \prod_{p=l+1}^{L} \left( \beta_2(\sigma_{p-1}) + \alpha_{p-2} \right), \tag{26}$$

$$\prod_{p=3}^{L}(\dot{G}_{ii}^{(p)} + \alpha_{p-2}) \geq \prod_{p=3}^{L} \left( \beta_3(\sigma_{p-1}) + \alpha_{p-2} \right). \tag{27}$$

According to the previous analysis, we know that $\boldsymbol{K}^{(L)}$ has $d$ non-zero eigenvalues. Their sum is equal to the trace of $\boldsymbol{K}^{(L)}$. The upper bound of $\lambda_{\min}(\boldsymbol{K}^{(L)})$ is directly given by:

$$\lambda_{\min}(\boldsymbol{K}^{(L)}) \leq \frac{1}{d} \sum_{i=1}^{N} \sum_{l=1}^{L}(G_{ii}^{(l)}) \prod_{p=l+1}^{L} (\dot{G}_{ii}^{(p)} + \alpha_{p-2})\,. \tag{28}$$

**Final result - upper bound**

For $G_{ii}^{(l)}$ we have the following bound:

$$G_{ii}^{(l)} = \beta_1(\sigma_{l-1})A_{ii}^{(l-1)} \leq \beta_1(\sigma_{l-1}) \prod_{p=3}^{l-1} \left( \beta_1(\sigma_{p-1}) + \alpha_{p-2} \right)A_{ii}^{(2)} \leq \beta_1(\sigma_{l-1}) \prod_{p=2}^{l-1} \left( \beta_1(\sigma_{p-1}) + \alpha_{p-2} \right).$$

$$(29)$$

By Equations (26), (28) and (29), we have:

$$\lambda_{\min}(\boldsymbol{K}^{(L)}) \leq \frac{N}{d} \sum_{l=1}^{L} \left( \beta_1(\sigma_{l-1}) \prod_{p=2}^{l-1} \left[ \beta_1(\sigma_{p-1}) + \alpha_{p-2} \right] \prod_{p=l+1}^{L} \left[ \beta_2(\sigma_{p-1}) + \alpha_{p-2} \right] \right). \tag{30}$$

**Final result - lower bound**

By Equations (25) and (27), with probability at least $1 - e^{-d}$, we have:

$$\lambda_{\min}(\boldsymbol{K}^{(L)}) \geq 2\mu_1(\sigma_1)^2\Theta(N/d) \prod_{p=3}^{L} \left( \beta_3(\sigma_{p-1}) + \alpha_{p-2} \right).$$

$\square$

### B.3 Special cases

To provide further insights into our proofs of mixed activation functions as provided in the previous sections, we now consider the special case of a single activation function in each layer.

**Corollary 4.** *Under Assumption 1, 3, for a deep fully-connected ResNet with the same activation functions in every layer and for a not very large $L$, let $\boldsymbol{K}^{(L)}$ be the limiting NTK recursively defined in Lemma 1. Then, with probability at least $1 - e^{-d}$, we have:*

*For ReLU:*

$$2\mu_1(\sigma_1)^2\Theta(N/d)\prod_{p=3}^{L}(1+\alpha_{p-2}) \le \lambda_{\min}(\boldsymbol{K}^{(L)}) \le \frac{N}{d}\sum_{l=1}^{L}\left(\frac{\prod_{p=2}^{L}(1+\alpha_{p-2})}{1+\alpha_{l-2}}\right).$$

*For LeakyReLU:*

$$2\mu_1(\sigma_1)^2\Theta(N/d)\prod_{p=3}^{L}(1+\eta^2+\alpha_{p-2}) \le \lambda_{\min}(\boldsymbol{K}^{(L)}) \le \frac{N}{d}(1+\eta^2)\sum_{l=1}^{L}\left(\frac{\prod_{p=2}^{L}(1+\eta^2+\alpha_{p-2})}{1+\eta^2+\alpha_{l-2}}\right).$$

*For Sigmoid:*

$$2\mu_1(\sigma_1)^2\Theta(N/d)\prod_{p=3}^{L}(f_S(\tfrac{1}{2})+\alpha_{p-2}) \le \lambda_{\min}(\boldsymbol{K}^{(L)}) \le \frac{N}{8d}\sum_{l=1}^{L}\left(\frac{\prod_{p=2}^{L}(\tfrac{1}{8}+\alpha_{p-2})}{\tfrac{1}{8}+\alpha_{l-2}}\right). \quad (31)$$

*For Tanh:*

$$2\mu_1(\sigma_1)^2\Theta(N/d)\prod_{p=3}^{L}(f_T(2)+\alpha_{p-2}) \le \lambda_{\min}(\boldsymbol{K}^{(L)}) \le \frac{2N}{d}\sum_{l=1}^{L}\left(\frac{\prod_{p=2}^{L}(2+\alpha_{p-2})}{2+\alpha_{l-2}}\right). \quad (32)$$

*For Swish:*

$$2\mu_1(\sigma_1)^2\Theta(N/d)\prod_{p=3}^{L}(\tfrac{1}{2}+\alpha_{p-2}) \le \lambda_{\min}(\boldsymbol{K}^{(L)}) \le \frac{N}{d}\sum_{l=1}^{L}\left(\prod_{p=2}^{l-1}(1+\alpha_{p-2})\prod_{p=l+1}^{L}(1.22+\alpha_{p-2})\right).$$

*The $\mu_1(\sigma_1)$ is 1-st Hermite coefficient of the activation function.*

*Proof.* By, Table 1 and Equations (25) and (30). we can have this result.

It should be noted that for Sigmoid network (all of activation functions are Sigmoid) and Tanh (all of activation functions are Tanh) network , the upper bound of $G_{\max}$ will change. By Equations (6) and (7) we have for Sigmoid $G_{\max} = \frac{1}{2}$, For Tanh $G_{\max} = 2$. That means $f_S(G_{\max})$ in the Theorem 1 is replaced by $f_S(\frac{1}{2})$ in Equation (31) and $f_T(G_{\max})$ in the Theorem 1 is replaced by $f_T(2)$ in Equation (32).

$\square$

## C   The bound of the minimum eigenvalues of NTK for finite-width

We present the details of our results on sec. 4.4 in this section. Firstly, we introduce the specific expression form for NTK of finite-width network in Appendix C.1. Then, we introduce some lemmas in Appendix C.2 to facilitate the proof of theorems, after that we provide the results of multiple activation functions are mixed in one network in Appendix C.3 directly, finally we discuss the results.

### C.1   Neural Tangent Kernel for finite-width

$$\bar{\boldsymbol{K}}^{(L)} = \boldsymbol{J}\boldsymbol{J}^{\top} = \sum_{l=1}^{L}\left[\frac{\partial \boldsymbol{F}}{\partial \text{vec}(\boldsymbol{W}_l)}\right]\left[\frac{\partial \boldsymbol{F}}{\partial \text{vec}(\boldsymbol{W}_l)}\right]^{\top}.$$

Let $\boldsymbol{F}_k = [\boldsymbol{f}_k(\boldsymbol{x}_1), \dots, \boldsymbol{f}_k(\boldsymbol{x}_N)]^T$, by chain rule and some standard calculation, we have,

$$\boldsymbol{J}\boldsymbol{J}^\top = \sum_{k=0}^{L-1} \boldsymbol{F}_k \boldsymbol{F}_k^\top \circ \boldsymbol{B}_{k+1} \boldsymbol{B}_{k+1}^\top\,,$$

where $\boldsymbol{B}_k \in \mathbb{R}^{N \times m}$ is a matrix of which the $i$-th row is given by

$$(\boldsymbol{B}_k)_{i:} = \begin{cases} \boldsymbol{D}_{i,k} \prod_{l=k+1}^{L-1} (\boldsymbol{W}_l \boldsymbol{D}_{i,l} + \alpha_{l-1} \boldsymbol{I}_{m\times m}) \boldsymbol{W}_L, & k \in [L-2]\,, \\ \boldsymbol{D}_{i,L-1} \boldsymbol{W}_L, & k = L - 1\,, \\ 1, & k = L\,. \end{cases}$$

## C.2 Relevant Lemmas

**Lemma 2.** *Fix any $k \in [0, L-1]$ and $\boldsymbol{x} \sim P_X$, then for ReLU, LeakyReLU, Sigmoid, Tanh and Swish we have*

$$\|\boldsymbol{f}_k(\boldsymbol{x})\|_2^2 = \Theta(1)\,,$$

*with probability at least $1 - \sum_{l=1}^k \exp(-\Omega(m))$ over $(\boldsymbol{W}_l)_{l=1}^k$ and $\boldsymbol{x}$. Moreover,*

$$\mathbb{E}_{\boldsymbol{x}} \|\boldsymbol{f}_k(\boldsymbol{x})\|_2^2 = \Theta(1)\,,$$

*with probability at least $1 - \sum_{l=1}^{k-1} \exp(-\Omega(m))$ over $(\boldsymbol{W}_l)_{l=1}^k$.*

*Proof.* We prove this by induction.

The result holds for $k = 0$ due to Assumption 1 and Assumption 3.

Assume that the lemma holds for some $k - 1$, i.e.

$$\|\boldsymbol{f}_{k-1}(\boldsymbol{x})\|_2^2 = \Theta(1)\,,$$

with probability at least $1 - \sum_{l=1}^{k-1} \exp(-\Omega(m))$ over $(\boldsymbol{W}_l)_{l=1}^k$ and $\boldsymbol{x}$.

Let us condition on this event of $(\boldsymbol{W}_l)_{l=1}^{k-1}$ and study probability bounds over $\boldsymbol{W}_k$: Let $\boldsymbol{W}_k = [\boldsymbol{w}_1, \cdots, \boldsymbol{w}_m]^\top$ where $\boldsymbol{w}_j \sim \mathcal{N}(0, \mathbb{I}_m/m)$ and $f_k^{[j]}$ represents the $j$-th element of $\boldsymbol{f}_k$. Note that:

$$\|\boldsymbol{f}_k(\boldsymbol{x})\|_2^2 = \sum_{j=1}^m f_k^{[j]}(\boldsymbol{x})^2\,. \tag{33}$$

Then we have:

$$
\begin{aligned}
\mathbb{E}_{\boldsymbol{W}_k} \|\boldsymbol{f}_k(\boldsymbol{x})\|_2^2 &= \sum_{j=1}^m \mathbb{E}_{\boldsymbol{w}_j}[f_k^{[j]}(\boldsymbol{x})^2] \\
&= \sum_{j=1}^m \mathbb{E}_{\boldsymbol{w}_j}\left(\left[\sigma_k\left(\langle \boldsymbol{w}_j, \boldsymbol{f}_{k-1}(\boldsymbol{x})\rangle\right) + \alpha_{k-1} f_{k-1}^{[j]}(\boldsymbol{x})\right]^2\right) \qquad Equation~(1) \\
&= \sum_{j=1}^m \left(\mathbb{E}_{\boldsymbol{w}_j}\left[\left(\sigma_k(\langle \boldsymbol{w}_j, \boldsymbol{f}_{k-1}(\boldsymbol{x})\rangle)\right)^2\right] + \mathbb{E}_{\boldsymbol{w}_j}\left(\alpha_{k-1}^2 f_{k-1}^{[j]}(\boldsymbol{x})^2\right)\right. \\
&\quad \left. + \mathbb{E}_{\boldsymbol{w}_j}\left[2\sigma_k\left(\langle \boldsymbol{w}_j, \boldsymbol{f}_{k-1}(\boldsymbol{x})\rangle\right)\alpha_{k-1} f_{k-1}^{[j]}(\boldsymbol{x})\right]\right) \\
&= m\mathbb{E}_{w\sim\mathcal{N}(0,\|\boldsymbol{f}_{k-1}(\boldsymbol{x})\|_2^2/m)}(\sigma_k(w)^2) + \sum_{j=1}^m \alpha_{k-1}^2 \mathbb{E}_{\boldsymbol{w}_j}\left(f_{k-1}^{[j]}(\boldsymbol{x})^2\right) \\
&\quad + 2\sum_{j=1}^m \alpha_{k-1}\mathbb{E}_{\boldsymbol{w}_j}\left(\sigma_k\left[\langle \boldsymbol{w}_j, \boldsymbol{f}_{k-1}(\boldsymbol{x})\rangle\right]\right)\mathbb{E}_{\boldsymbol{w}_j}\left(f_{k-1}^{[j]}(\boldsymbol{x})\right) \\
&= m\mathbb{E}_{w\sim\mathcal{N}(0,\|\boldsymbol{f}_{k-1}(\boldsymbol{x})\|_2^2/m)}(\sigma_k(w)^2) + \alpha_{k-1}^2 \|\boldsymbol{f}_{k-1}(\boldsymbol{x})\|_2^2 \\
&\quad + 2\alpha_{k-1}\mathbb{E}_{w\sim\mathcal{N}(0,\|\boldsymbol{f}_{k-1}(\boldsymbol{x})\|_2^2/m)}(\sigma_k(w))\sum_{j=1}^m f_{k-1}^{[j]}(\boldsymbol{x}) .
\end{aligned}
$$

$$(34)$$

According to Equations (4), (5), (8), (11), (13), (14), (16) and (17), we know that when $\sigma_{k-1}$ are in ReLU, LeakyReLU, Sigmoid, Tanh and Swish we have:

$$
m\mathbb{E}_{w\sim\mathcal{N}(0,\|\boldsymbol{f}_{k-1}(\boldsymbol{x})\|_2^2/m)}(\sigma_k(w)^2) = m\Theta\left(\frac{\|\boldsymbol{f}_{k-1}(\boldsymbol{x})\|_2^2}{m}\right) = \Theta(\|\boldsymbol{f}_{k-1}(\boldsymbol{x})\|_2^2) . \qquad (35)
$$

When $\sigma_{k-1}$ is ReLU, LeakyReLU or Swish, Equation (35) can be written as:

$$
\frac{1}{2}\|\boldsymbol{f}_{k-1}(\boldsymbol{x})\|_2^2 \leq m\mathbb{E}_{w\sim\mathcal{N}(0,\|\boldsymbol{f}_{k-1}(\boldsymbol{x})\|_2^2/m)}(\sigma_k(w)^2) \leq (1+\eta^2)\|\boldsymbol{f}_{k-1}(\boldsymbol{x})\|_2^2 ,
$$

$$
0 < \mathbb{E}_{w\sim\mathcal{N}(0,\|\boldsymbol{f}_{k-1}(\boldsymbol{x})\|_2^2/m)}(\sigma_k(w)) \leq \mathbb{E}_{w\sim\mathcal{N}(0,\|\boldsymbol{f}_{k-1}(\boldsymbol{x})\|_2^2/m)}(f_{\mathrm{ReLU}}(w)) = \frac{2\|\boldsymbol{f}_{k-1}(\boldsymbol{x})\|_2}{5\sqrt{m}} .
$$

According to the relationship between the vectors 1-norm and 2-norm, we have:

$$
-\sqrt{m}\|\boldsymbol{f}_{k-1}(\boldsymbol{x})\|_2 \leq \sum_{j=1}^m f_{k-1}^{[j]}(\boldsymbol{x}) \leq \sqrt{m}\|\boldsymbol{f}_{k-1}(\boldsymbol{x})\|_2 .
$$

Then:

$$
-\frac{2\|\boldsymbol{f}_{k-1}(\boldsymbol{x})\|_2^2}{5} \leq 2\alpha_{k-1}\mathbb{E}_{w\sim\mathcal{N}(0,\|\boldsymbol{f}_{k-1}(\boldsymbol{x})\|_2^2/m)}(\sigma_k(w))\sum_{j=1}^m f_{k-1}^{[j]}(\boldsymbol{x}) \leq \frac{2\|\boldsymbol{f}_{k-1}(\boldsymbol{x})\|_2^2}{5} .
$$

If we substitute into Equation (34), we have upper bound and lower bound for $\mathbb{E}_{\boldsymbol{W}_k} \|\boldsymbol{f}_k(\boldsymbol{x})\|_2^2$:

$$
\begin{aligned}
\mathbb{E}_{\boldsymbol{W}_k} \|\boldsymbol{f}_k(\boldsymbol{x})\|_2^2 &= m\mathbb{E}_{w\sim\mathcal{N}(0,\|\boldsymbol{f}_{k-1}(\boldsymbol{x})\|_2^2/m)}(\sigma_k(w)^2) + \alpha_{k-1}^2 \|\boldsymbol{f}_{k-1}(\boldsymbol{x})\|_2^2 \\
&\quad + 2\alpha_{k-1}\mathbb{E}_{w\sim\mathcal{N}(0,\|\boldsymbol{f}_{k-1}(\boldsymbol{x})\|_2^2/m)}(\sigma_k(w)) \sum_{j=1}^{m} f_{k-1}^{[j]}(\boldsymbol{x}) \\
&\leq \left(1 + \eta^2 + \alpha_{k-1} + \frac{2}{5}\right) \|\boldsymbol{f}_{k-1}(\boldsymbol{x})\|_2^2 \\
&\leq \left(\eta^2 + \frac{12}{5}\right) \Theta(1) ,
\end{aligned}
$$

$$
\begin{aligned}
\mathbb{E}_{\boldsymbol{W}_k} \|\boldsymbol{f}_k(\boldsymbol{x})\|_2^2 &= m\mathbb{E}_{w\sim\mathcal{N}(0,\|\boldsymbol{f}_{k-1}(\boldsymbol{x})\|_2^2/m)}(\sigma_k(w)^2) + \alpha_{k-1}^2 \|\boldsymbol{f}_{k-1}(\boldsymbol{x})\|_2^2 \\
&\quad + 2\alpha_{k-1}\mathbb{E}_{w\sim\mathcal{N}(0,\|\boldsymbol{f}_{k-1}(\boldsymbol{x})\|_2^2/m)}(\sigma_k(w)) \sum_{j=1}^{m} f_{k-1}^{[j]}(\boldsymbol{x}) \\
&\geq \left(\frac{1}{2} + \alpha_{k-1} - \frac{2}{5}\right) \|\boldsymbol{f}_{k-1}(\boldsymbol{x})\|_2^2 \\
&\geq \frac{1}{10}\Theta(1) .
\end{aligned}
$$

That means, when $\sigma_{k-1}$ is ReLU, LeakyReLU or Swish we have:

$$
\mathbb{E}_{\boldsymbol{W}_k} \|\boldsymbol{f}_k(\boldsymbol{x})\|_2^2 = \Theta(1) . \tag{36}
$$

When $\sigma_{k-1}$ is Sigmoid or Tanh, according to symmetry we have:

$$
\mathbb{E}_{w\sim\mathcal{N}(0,\|\boldsymbol{f}_{k-1}(\boldsymbol{x})\|_2^2/m)}[\sigma_k(w)] = 0 .
$$

Then:

$$
\begin{aligned}
\mathbb{E}_{\boldsymbol{W}_k} \|f_k(\boldsymbol{x})\|_2^2 &= \Theta(\|\boldsymbol{f}_{k-1}(\boldsymbol{x})\|_2^2) + \alpha_{k-1}^2 \|\boldsymbol{f}_{k-1}(\boldsymbol{x})\|_2^2 \\
&= \Theta(\|\boldsymbol{f}_{k-1}(\boldsymbol{x})\|_2^2) .
\end{aligned} \tag{37}
$$

By Equations (36) and (37), when $\sigma_{k-1}$ is ReLU, LeakyReLU, Sigmoid, Tanh or Swish we have:

$$
\mathbb{E}_{\boldsymbol{W}_k} \|\boldsymbol{f}_k(\boldsymbol{x})\|_2^2 = \Theta(1).
$$

Thus, by applying Bernstein's inequality to the sum of i.i.d. random variables in Equation (33), we have:

$$
\frac{1}{2}\mathbb{E}_{\boldsymbol{W}_k} \|\boldsymbol{f}_k(\boldsymbol{x})\|_2^2 \leq \|\boldsymbol{f}_k(\boldsymbol{x})\|_2^2 \leq \frac{3}{2}\mathbb{E}_{\boldsymbol{W}_k} \|\boldsymbol{f}_k(\boldsymbol{x})\|_2^2 ,
$$

with probability at least $1 - \exp(-\Omega(m))$. i.e.:

$$
\|\boldsymbol{f}_k(\boldsymbol{x})\|_2^2 = \Theta(1) ,
$$

with probability at least $1 - \sum_{l=1}^{k} \exp(-\Omega(m))$.

The proof for $\mathbb{E}_{\boldsymbol{x}} \|\boldsymbol{f}_k(\boldsymbol{x})\|_2^2$ can be done by following similar passages and using that $\left\| \mathbb{E}_{\boldsymbol{x}}[f_k^{[j]}(\boldsymbol{x})^2] \right\|_{\psi_1} \leq \mathbb{E}_{\boldsymbol{x}} \left\| f_k^{[j]}(\boldsymbol{x})^2 \right\|_{\psi_1}$. $\qquad\square$

**Lemma 3.** *Fix any layer $k \in [L-1]$, and $\boldsymbol{x} \sim P_X$. Then, we have that $\|\boldsymbol{D}_k\|_{\mathrm{F}}^2 = \Theta(m)$ with probability at least $1 - \sum_{l=1}^{k} \exp(-\Omega(m))$ over $(\boldsymbol{W}_l)_{l=1}^k$ and $\boldsymbol{x}$.*

*Proof.* By Lemma 2, we have $f_{k-1}(x) \neq 0$ with probability at least $1 - \sum_{l=1}^{k} \exp(-\Omega(m))$ over $(W_l)_{l=1}^{k}$ and $x$. Let us condition on this event and derive probability bounds over $W_k$. Let $W_k = [w_1, \cdots, w_{n_k}]$. Then, $\|D_k\|_F^2 = \sum_{j=1}^{m} \sigma_k'^2(\langle f_{k-1}(x), w_j \rangle)$. Thus:

$$\mathbb{E}_{W_k} \|D_k\|_F^2 = m\mathbb{E}_{w_1}[\sigma_k'^2(\langle f_{k-1}(x), w_1 \rangle)] = m\mathbb{E}_{w \sim \mathcal{N}(0, \|f_{k-1}(x)\|_2^2/m)}[\sigma_k'^2(w)].$$

By Equations (19) to (24), we know that when $\sigma_k$ are in ReLU, LeakyReLU, Sigmoid, Tanh and Swish we have:

$$\mathbb{E}_{W_k} \|D_k\|_F^2 = m\Theta(1) = \Theta(m).$$

By Hoeffding's inequality on bounded random variables, we have:

$$\mathbb{P}\left(\left|\|D_k\|_F^2 - \mathbb{E}_{W_k} \|D_k\|_F^2\right| > t\right) \leq 2\exp\left(-\frac{2t^2}{m}\right).$$

Picking $t := 0.01m$ concludes the proof.

$\square$

**Lemma 4.** *For any $k \in [L-1]$, $k \leq p \leq L-1$ and $x \sim P_X$, we have that:*

$$\Theta\left(m \prod_{i=k+1}^{p} (\beta_3(\sigma_i) + \alpha_{i-1})\right) \leq \left\|D_k \prod_{l=k+1}^{p} \left(W_l D_l + \alpha_{l-1} I_{m \times m}\right)\right\|_F^2 \leq \Theta\left(m \prod_{i=k+1}^{p} (\beta_2(\sigma_i) + \alpha_{i-1})\right),$$

*with probability at least $1 - \sum_{l=k+1}^{p} \exp(-\Omega(m))$ over $(W_l)_{l=k+1}^{p}$ and $x$.*

*Proof.* We prove this by induction on $p$.

Lemma 3 implies that the statement holds for $p = k$.

Suppose it holds for some $p-1$. Let $S_p = D_k \prod_{l=k+1}^{p}(W_l D_l + \alpha_{l-1} I_{m \times m})$. Then, $S_p = S_{p-1}(W_p D_p + \alpha_{p-1} I_{m \times m}) = S_{p-1} W_p D_p + \alpha_{p-1} S_{p-1}$. Let $W_p = [w_1, \ldots, w_{n_p}]$. Then:

$$\|S_p\|_F^2 = \sum_{j=1}^{m} \|S_{p-1} w_j\|_2^2 \sigma_p'(\langle f_{p-1}(x), w_j \rangle)^2 + \alpha_{p-1} \|S_{p-1}\|_F^2.$$

Then we have:

$$\begin{aligned}
\mathbb{E}_{W_p} \|S_p\|_F^2 &= m\mathbb{E}_{w \sim \mathcal{N}(0, \mathbb{I}_m/m)} \|S_{p-1} w_j\|_2^2 \sigma_p'(\langle f_{p-1}(x), w_j \rangle)^2 + \alpha_{p-1} \|S_{p-1}\|_F^2 \\
&= m\mathbb{E}_{w \sim \mathcal{N}(0, \mathbb{I}_m/m)} \|S_{p-1} w_j\|_2^2 \, \mathbb{E}_{w \sim \mathcal{N}(0, \mathbb{I}_m/m)} \sigma_p'(\langle f_{p-1}(x), w_j \rangle)^2 + \alpha_{p-1} \|S_{p-1}\|_F^2 \\
&= \|S_{p-1}\|_F^2 \, \mathbb{E}_{w \sim \mathcal{N}(0, \|f_{p-1}(x)\|_2^2/m)} \sigma_p'(w)^2 + \alpha_{p-1} \|S_{p-1}\|_F^2.
\end{aligned}$$

From the previous result Equations (19) to (24), we have:

$$\beta_3(\sigma_p) \leq \mathbb{E}_{w \sim \mathcal{N}(0, \|f_{p-1}(x)\|_2^2/m)} \sigma_p'(w)^2 \leq \beta_2(\sigma_p).$$

That is:

$$(\beta_3(\sigma_p) + \alpha_{p-1}) \|S_{p-1}\|_F^2 \leq \mathbb{E}_{W_p} \|S_p\|_F^2 \leq (\beta_2(\sigma_p) + \alpha_{p-1}) \|S_{p-1}\|_F^2.$$

Moreover:

$$\left\| \|\boldsymbol{S}_{p-1}\boldsymbol{w}_j\|_2^2 \, \sigma_p'(\langle \boldsymbol{f}_{p-1}(\boldsymbol{x}), \boldsymbol{w}_j \rangle)^2 \right\|_{\psi_1} \leq \left\| \|\boldsymbol{S}_{p-1}\boldsymbol{w}_j\|_2 \right\|_{\psi_2}^2 \leq \frac{c}{m}\|\boldsymbol{S}_{p-1}\|_{\mathrm{F}}^2 \ .$$

By Bernstein's inequality [Vershynin, 2018], we have:

$$\frac{1}{2}\mathbb{E}_{\boldsymbol{W}_p}\|\boldsymbol{S}_p\|_{\mathrm{F}}^2 \leq \|\boldsymbol{S}_p\|_{\mathrm{F}}^2 \leq \frac{3}{2}\mathbb{E}_{\boldsymbol{W}_p}\|\boldsymbol{S}_p\|_{\mathrm{F}}^2 \ ,$$

with probability at least $1 - \exp(-\Omega(m))$. Finally, taking the intersection of all the events finishes the proof.

$\square$

**Lemma 5.** *For any layer $k \in [L-2]$ and $\boldsymbol{x} \sim P_X$, we have:*

$$\Theta\bigg(\prod_{i=k+1}^{L-1}(\beta_3(\sigma_i)+\alpha_{i-1})\bigg) \leq \bigg\|\boldsymbol{D}_k \prod_{l=k+1}^{L-1}\Big(\boldsymbol{W}_l\boldsymbol{D}_l + \alpha_{l-1}\boldsymbol{I}_{m\times m}\Big)\boldsymbol{W}_L\bigg\|_2^2 \leq \Theta\bigg(\prod_{i=k+1}^{L-1}(\beta_2(\sigma_i)+\alpha_{i-1})\bigg),$$

*with probability at least $1 - \sum_{l=k+1}^{L-1}\exp(-\Omega(m)) - \exp(-\Omega(1))$.*

*Proof.* Let $\boldsymbol{B} = \boldsymbol{D}_k \prod_{l=k+1}^{L-1}(\boldsymbol{W}_l\boldsymbol{D}_l + \alpha_{l-1}\boldsymbol{I}_{m\times m})$.

By Lemma 4, we have:

$$\Theta\bigg(m \prod_{i=k+1}^{L-1}(\beta_3(\sigma_i) + \alpha_{i-1})\bigg) \leq \|\boldsymbol{B}\|_{\mathrm{F}}^2 \leq \Theta\bigg(m \prod_{i=k+1}^{L-1}(\beta_2(\sigma_i) + \alpha_{i-1})\bigg), \tag{38}$$

with probability at least $1 - \sum_{l=k+1}^{L-1}\exp(-\Omega(m))$.

Then, by Hanson-Wright inequality [Vershynin, 2018], we have:

$$\frac{1}{2m}\|\boldsymbol{B}\|_{\mathrm{F}}^2 = \frac{1}{2}\mathbb{E}_{\boldsymbol{W}_L}\|\boldsymbol{B}\boldsymbol{W}_L\|_2^2 \leq \|\boldsymbol{B}\boldsymbol{W}_L\|_2^2 \leq \frac{3}{2}\mathbb{E}_{\boldsymbol{W}_L}\|\boldsymbol{B}\boldsymbol{W}_L\|_2^2 = \frac{3}{2m}\|\boldsymbol{B}\|_{\mathrm{F}}^2 \ , \tag{39}$$

with probability at least $1 - \exp(-\Omega(\|\boldsymbol{B}\|_{\mathrm{F}}^2 / \|\boldsymbol{B}\|_2^2)) \geq 1 - \exp(-\Omega(1))$ over $\boldsymbol{W}_L$.

According to Equations (38) and (39), we can get the result.

$\square$

## C.3 Results for mixed activation functions under the finite-width setting (Proof of Theorem 2)

*Proof.* We firstly present the lower bound of the minimal eigenvalue of $\boldsymbol{J}\boldsymbol{J}^\top$ and then derive its upper bound.

**Lower bound**

For PSD matrices $\boldsymbol{P}, \boldsymbol{Q} \in \mathbb{R}^{N \times N}$, it holds $\lambda_{\min}(\boldsymbol{P} \circ \boldsymbol{Q}) \geq \lambda_{\min}(\boldsymbol{P}) \min_{i \in [N]} Q_{ii}$. Then, by Theorem 1 and Theorem 5.1 of Nguyen et al. [2021]:

$$\begin{aligned}
\lambda_{\min}(\boldsymbol{J}\boldsymbol{J}^\top) &\geq \sum_{k=0}^{L-1} \lambda_{\min}(\boldsymbol{F}_k\boldsymbol{F}_k^\top) \min_{i \in [N]} \|(\boldsymbol{B}_{k+1})_{i:}\|_2^2 \\
&\geq \lambda_{\min}(\boldsymbol{F}_0\boldsymbol{F}_0^\top) \min_{i \in [N]} \|(\boldsymbol{B}_1)_{i:}\|_2^2 \\
&\geq \Theta\bigg(\frac{N}{d} \prod_{i=2}^{L-1}(\beta_3(\sigma_i) + \alpha_{i-1})\bigg),
\end{aligned}$$

with probability at least $1 - e^{-d} - \sum_{l=1}^{L-1} \exp(-\Omega(m)) - \exp(-\Omega(1))$. where the last inequality hold by Lemma 5 and Proposition 4.

**Upper bound**

For ReLU, LeakyReLU and Swish we have:

$$\lambda_{min}(\boldsymbol{J}\boldsymbol{J}^\top) \le \sum_{i=0}^{N}(\boldsymbol{J}\boldsymbol{J}^\top)_{ii}/d = \frac{1}{d}\sum_{i=0}^{N}\sum_{k=0}^{L-1}\|(\boldsymbol{F}_k)_{1:}\|_2^2\,\|(\boldsymbol{B}_{k+1})_{1:}\|_2^2$$

$$= \frac{1}{d}\sum_{i=0}^{N}\sum_{k=0}^{L-1}\|\boldsymbol{f}_k(\boldsymbol{x_1})\|_2^2\,\|(\boldsymbol{B}_{k+1})_{1:}\|_2^2\,.$$

By Lemma 2 and Lemma 5 we have:

$$\lambda_{min}(\boldsymbol{J}\boldsymbol{J}^\top) \le \frac{N}{d}\sum_{k=0}^{L-1}\Theta\left(\prod_{i=k+2}^{L-1}(\beta_2(\sigma_i)+\alpha_{i-1})\right),$$

with probability at least $1 - \sum_{l=1}^{L-1}\exp(-\Omega(m)) - \exp(-\Omega(1))$.

$\square$

# D  Generalization error via the minimum eigenvalue of NTK

In this section, firstly, we provide some useful lemmas in Appendix D.1,then present the proof of Theorem 3 in Appendix D.2.

## D.1  Relevant Lemmas

**Lemma 6.** *( Vershynin [2018, Theorem 4.4.5]) Let $\boldsymbol{A}$ be an $N \times n$ matrix whose entries are independent standard normal random variables. Then for every $t \ge 0$, with probability at least $1 - 2\exp(-t^2/2)$, one has:*

$$s(\boldsymbol{A})_{\max} \le \sqrt{N} + \sqrt{n} + t\,.$$

We need the following lemma to show that the output of each neuron with any activation function does not change too much if the input weights are close.

**Lemma 7.** *Let $\boldsymbol{W} \in \mathbb{R}^{m \times m}$ be the random Gaussian matrix with $W_{i,j} \sim \mathcal{N}(0, 1/m)$, $\mathrm{Lip}_{\max}$ be the maximum value of the Lipschitz constants of the all activation functions, with $\omega = \mathcal{O}((3\mathrm{Lip}_{\max} + 1)^{-(L-1)})$, assuming $\widetilde{\boldsymbol{W}} \in \mathcal{B}(\boldsymbol{W}, \omega)$, for any $l \in [L]$, it holds that $\left\|\hat{\boldsymbol{f}}_{i,l}\right\|_2 = \mathcal{O}(1)$ with probability at least $1 - 2l\exp(-m/2) - l\exp(-\Omega(m))$.*

*Proof.* We provide the estimation on $\hat{\boldsymbol{f}}_{i,1}$ and $\hat{\boldsymbol{f}}_{i,l}$ ($l = 2, 3, \cdots, L$) in Definition 2, respectively. Firstly, $\hat{\boldsymbol{f}}_{i,1}$ admits:

$$\left\|\hat{\boldsymbol{f}}_{i,1}\right\|_2 = \left\|\widetilde{\boldsymbol{f}}_{i,1} - \boldsymbol{f}_{i,1}\right\|_2 = \left\|\sigma_1(\widetilde{\boldsymbol{W}}_1\boldsymbol{x}_i) - \sigma_1(\boldsymbol{W}_1\boldsymbol{x}_i)\right\|_2$$

$$\le \mathrm{Lip}_{\sigma_1}\left\|\widetilde{\boldsymbol{W}}_1 - \boldsymbol{W}_1\right\|_2\|\boldsymbol{x}_i\|_2 \le \omega\mathrm{Lip}_{\sigma_1}$$

$$= \mathcal{O}(1)\,.$$

For $\hat{\boldsymbol{f}}_{i,l}$ with $l = 2, 3, \ldots, L$, we have:

$$
\begin{aligned}
\left\|\hat{\boldsymbol{f}}_{i,l}\right\|_2 &= \left\|\widetilde{\boldsymbol{f}}_{i,l} - \boldsymbol{f}_{i,l}\right\|_2 \\
&= \left\|\sigma_l(\widetilde{\boldsymbol{W}}_l\widetilde{\boldsymbol{f}}_{i,l-1}) + \alpha_{l-1}\widetilde{\boldsymbol{f}}_{i,l-1} - \sigma_l(\boldsymbol{W}_l\boldsymbol{f}_{i,l-1}) - \alpha_{l-1}\boldsymbol{f}_{i,l-1}\right\|_2 \\
&\leq \left\|\sigma_l(\widetilde{\boldsymbol{W}}_l\widetilde{\boldsymbol{f}}_{i,l-1}) - \sigma_l(\boldsymbol{W}_l\boldsymbol{f}_{i,l-1})\right\|_2 + \alpha_{l-1}\left\|\hat{\boldsymbol{f}}_{i,l-1}\right\|_2 \\
&\leq \mathrm{Lip}_{\sigma_l}\left\|\widetilde{\boldsymbol{W}}_l\widetilde{\boldsymbol{f}}_{i,l-1} - \boldsymbol{W}_l\boldsymbol{f}_{i,l-1}\right\|_2 + \left\|\hat{\boldsymbol{f}}_{i,l-1}\right\|_2 \qquad \text{[Lipschitz continuity of } \sigma_l\text{]} \\
&= \mathrm{Lip}_{\sigma_l}\left\|\boldsymbol{W}_l(\widetilde{\boldsymbol{f}}_{i,l-1} - \boldsymbol{f}_{i,l-1}) + (\widetilde{\boldsymbol{W}}_l - \boldsymbol{W}_l)\widetilde{\boldsymbol{f}}_{i,l-1}\right\|_2 + \left\|\hat{\boldsymbol{f}}_{i,l-1}\right\|_2 \qquad (40) \\
&\leq \mathrm{Lip}_{\sigma_l}\left\{\left\|\boldsymbol{W}_l(\widetilde{\boldsymbol{f}}_{i,l-1} - \boldsymbol{f}_{i,l-1})\right\|_2 + \left\|(\widetilde{\boldsymbol{W}}_l - \boldsymbol{W}_l)\widetilde{\boldsymbol{f}}_{i,l-1}\right\|_2\right\} + \left\|\hat{\boldsymbol{f}}_{i,l-1}\right\|_2 \\
&\leq \mathrm{Lip}_{\sigma_l}\left\{\left\|\boldsymbol{W}_l\right\|_2\left\|\widetilde{\boldsymbol{f}}_{i,l-1} - \boldsymbol{f}_{i,l-1}\right\|_2 + \left\|\widetilde{\boldsymbol{W}}_l - \boldsymbol{W}_l\right\|_2\left\|\widetilde{\boldsymbol{f}}_{i,l-1}\right\|_2\right\} + \left\|\hat{\boldsymbol{f}}_{i,l-1}\right\|_2 \\
&\leq (\mathrm{Lip}_{\sigma_l}\left\|\boldsymbol{W}_l\right\|_2 + 1)\left\|\hat{\boldsymbol{f}}_{i,l-1}\right\|_2 + \mathrm{Lip}_{\sigma_l}\omega\left(\left\|\widetilde{\boldsymbol{f}}_{i,l-1} - \boldsymbol{f}_{i,l-1}\right\|_2 + \|\boldsymbol{f}_{i,l-1}\|_2\right) \\
&= \left\{\mathrm{Lip}_{\sigma_l}(\left\|\boldsymbol{W}_l\right\|_2 + \omega) + 1\right\}\left\|\hat{\boldsymbol{f}}_{i,l-1}\right\|_2 + \mathrm{Lip}_{\sigma_l}\omega\|\boldsymbol{f}_{i,l-1}\|_2 \, .
\end{aligned}
$$

By Lemma 6, choosing $t = \sqrt{m}$, with probability at least $1 - 2\exp(-m/2)$, we have:

$$
\|\boldsymbol{W}_l\|_2 = s(\boldsymbol{W}_l)_{\max} \leq \frac{\sqrt{m} + \sqrt{m} + \sqrt{m}}{\sqrt{m}} = 3 \, .
$$

Then, $\|\hat{\boldsymbol{f}}_{i,l}\|_2$ in Equation (40) can be further upper bounded with probability at least $1 - 2l\exp(-m/2) - l\exp(-\Omega(m))$:

$$
\begin{aligned}
\left\|\hat{\boldsymbol{f}}_{i,l}\right\|_2 &\leq \left((3 + \omega)\mathrm{Lip}_{\max} + 1\right)\left\|\hat{\boldsymbol{f}}_{i,l-1}\right\|_2 + \mathrm{Lip}_{\max}\omega\|\boldsymbol{f}_{i,l-1}\|_2 \\
&\leq \left([(3 + \omega)\mathrm{Lip}_{\max} + 1]^{l-1} - 1\right)\left(\mathrm{Lip}_{\sigma_1}\omega + \frac{\mathrm{Lip}_{\max}\omega\|\boldsymbol{f}_{i,l-1}\|_2}{(3 + \omega)\mathrm{Lip}_{\max}}\right) + \mathrm{Lip}_{\sigma_1}\omega \\
&\leq (3\mathrm{Lip}_{\max} + 1)^{L-1}\Theta(1)\omega + \mathrm{Lip}_{\sigma_1}\omega \\
&= \mathcal{O}(1)\Theta(1) + \mathcal{O}(1) \\
&= \mathcal{O}(1) \, ,
\end{aligned}
$$

where the second inequality holds by the recursion which conclude the proof.

$\square$

We also need the following lemma, demonstrating that the neural network function is almost linear in terms of its weights if the initializations are close to each other.

**Lemma 8.** *Let $\boldsymbol{W}, \boldsymbol{W}' \in \mathcal{B}(\boldsymbol{W}^{(0)}, \omega)$ with $\omega = \mathcal{O}((3\mathrm{Lip}_{\max} + 1)^{-(L-1)})$, for any $i \in [N]$, with probability at least $1 - 2(L-1)\exp(-m/2) - L\exp(-\Omega(m)) - 2/m$, we have:*

$$
|f(\boldsymbol{x}_i; \boldsymbol{W}') - f(\boldsymbol{x}_i; \boldsymbol{W}) - \langle\nabla f(\boldsymbol{x}_i; \boldsymbol{W}), \boldsymbol{W}' - \boldsymbol{W}\rangle| = \mathcal{O}(1) \, .
$$

*Proof.* We have the following expression:

$$\left| f(\boldsymbol{x}_i; \boldsymbol{W}') - f(\boldsymbol{x}_i; \boldsymbol{W}) - \langle \nabla f(\boldsymbol{x}_i; \boldsymbol{W}), \boldsymbol{W}' - \boldsymbol{W} \rangle \right|$$

$$= \left| \sum_{l=1}^{L-1} \boldsymbol{W}_L \prod_{r=l+1}^{L-1} (\boldsymbol{D}_{i,r} \boldsymbol{W}_r + \alpha_{r-1} \boldsymbol{I}_{m \times m}) \boldsymbol{D}_{i,l} (\boldsymbol{W}_l' - \boldsymbol{W}_l) \boldsymbol{f}_{i,l-1} + \boldsymbol{W}_L' (\boldsymbol{f}_{i,L-1}' - \boldsymbol{f}_{i,L-1}) \right|$$

$$\leq \sum_{l=1}^{L-1} \left| \boldsymbol{W}_L \prod_{r=l+1}^{L-1} (\boldsymbol{D}_{i,r} \boldsymbol{W}_r + \alpha_{r-1} \boldsymbol{I}_{m \times m}) \boldsymbol{D}_{i,l} (\boldsymbol{W}_l' - \boldsymbol{W}_l) \boldsymbol{f}_{i,l-1} \right| + \left| \boldsymbol{W}_L' (\boldsymbol{f}_{i,L-1}' - \boldsymbol{f}_{i,L-1}) \right|$$

$$\leq \sum_{l=1}^{L-1} \|\boldsymbol{W}_L\|_2 \left\| \prod_{r=l+1}^{L-1} (\boldsymbol{D}_{i,r} \boldsymbol{W}_r + \alpha_{r-1} \boldsymbol{I}_{m \times m}) \boldsymbol{D}_{i,l} (\boldsymbol{W}_l' - \boldsymbol{W}_l) \boldsymbol{f}_{i,l-1} \right\|_2$$

$$+ \|\boldsymbol{W}_L'\|_2 \|\boldsymbol{f}_{i,L-1}' - \boldsymbol{f}_{i,L-1}\|_2$$

$$\leq \sum_{l=1}^{L-1} \|\boldsymbol{W}_L\|_2 \prod_{r=l+1}^{L-1} (\|\boldsymbol{D}_{i,r}\|_2 \|\boldsymbol{W}_r\|_2 + \alpha_{r-1}) \|\boldsymbol{D}_{i,l}\|_2 \|\boldsymbol{W}_l' - \boldsymbol{W}_l\|_2 \|\boldsymbol{f}_{i,l-1}\|_2 + \|\boldsymbol{W}_L'\|_2 \|\boldsymbol{f}_{i,L-1}' - \boldsymbol{f}_{i,L-1}\|_2 \,.$$

(41)

Here we require the derivative of the activation function $\sigma'$ is bound, i.e., $\|\boldsymbol{D}\|_2 \leq \text{Lip}_{\max}$. The considered activation functions in this paper satisfy this condition.

By Lemma 6, Lemma 7 and Lemma 2 with probability at least $1 - 2(L-1)\exp(-m/2) - L\exp(-\Omega(m))$, we have $\|\boldsymbol{f}_{i,L-1}' - \boldsymbol{f}_{i,L-1}\|_2 \leq \mathcal{O}(1)$, $\|\boldsymbol{f}_{i,l-1}\|_2 = \Theta(1)$ and $\|\boldsymbol{W}_r\|_2 \leq 3 \quad \forall r \in [L-1]$.

Moreover, $m\|\boldsymbol{W}_L\|_2^2$ is a random Variables obey chi-square distribution with $m$ degrees of freedom. That means $\mathbb{E}(m\|\boldsymbol{W}_L\|_2^2) = m$ and $\mathbb{V}(m\|\boldsymbol{W}_L\|_2^2) = 2m$. By Chebyshev's Inequality we have $P(|m\|\boldsymbol{W}_L\|_2^2 - m| \geq m) \leq 2m/m^2$. i.e.:

$$\|\boldsymbol{W}_L\|_2 \leq \sqrt{2}\,,$$

with probability at least $1 - 2/m$.

Accordingly, Equation (41) can be further upper bounded by:

$$|f(\boldsymbol{x}_i; \boldsymbol{W}') - f(\boldsymbol{x}_i; \boldsymbol{W}) - \langle \nabla f(\boldsymbol{x}_i; \boldsymbol{W}), \boldsymbol{W}' - \boldsymbol{W} \rangle|$$

$$\leq \sum_{l=1}^{L-1} (3\text{Lip}_{\max} + 1)^{L-l-1} \omega \sqrt{2} \text{Lip}_{\max} \Theta(1) + (\sqrt{2} + \omega)\mathcal{O}(1)$$

$$= \frac{(3\text{Lip}_{\max} + 1)^{L-1} - 1}{3\text{Lip}_{\max}} \omega \sqrt{2} \text{Lip}_{\max} \Theta(1) + (\sqrt{2} + \omega)\mathcal{O}(1)$$

$$= \mathcal{O}(1)\,.$$

$\square$

We define $L_i(\boldsymbol{W}) = \ell[y_i f(\boldsymbol{x}_i; \boldsymbol{W})]$, then the following lemma shows that, $L_i(\boldsymbol{W})$ is almost a convex function of $\boldsymbol{W}$ for any $i \in [N]$ if the initilizations are close.

**Lemma 9.** *Let* $\boldsymbol{W}, \boldsymbol{W}' \in \mathcal{B}(\boldsymbol{W}^{(0)}, \omega)$ *with* $\omega = \mathcal{O}((3\text{Lip}_{\max} + 1)^{-(L-1)})$, *for any* $i \in [N]$, *it holds that:*

$$L_i(\boldsymbol{W}') \geq L_i(\boldsymbol{W}) + \langle \nabla_{\boldsymbol{W}} L_i(\boldsymbol{W}), \boldsymbol{W}' - \boldsymbol{W} \rangle - \mathcal{O}(1)\,,$$

*with probability at least* $1 - 2(L-1)\exp(-m/2) - L\exp(-\Omega(m)) - 2/m$.

*Proof.* By the convexity of $\ell(z)$, we have:

$$L_i(\boldsymbol{W}') - L_i(\boldsymbol{W}) = \ell[y_i f(\boldsymbol{x}_i; \boldsymbol{W}')] - \ell[y_i f(\boldsymbol{x}_i; \boldsymbol{W})] \geq \ell'[y_i f(\boldsymbol{x}_i; \boldsymbol{W})] \cdot y_i \cdot [f(\boldsymbol{x}_i; \boldsymbol{W}') - f(\boldsymbol{x}_i; \boldsymbol{W})]\,.$$

Using the chain rule leads to:

$$\sum_{l=1}^{L} \langle \nabla_{\boldsymbol{W}_l} L_i(\boldsymbol{W}), \boldsymbol{W}_l' - \boldsymbol{W}_l \rangle = \ell'[y_i f(\boldsymbol{x}_i; \boldsymbol{W})] \cdot y_i \cdot \langle \nabla f(\boldsymbol{x}_i; \boldsymbol{W}), \boldsymbol{W}' - \boldsymbol{W} \rangle\,.$$

Combining the above two equations, by triangle inequality, we have:

$$\ell'[y_i f(\boldsymbol{x}_i; \boldsymbol{W})] \cdot y_i \cdot [f(\boldsymbol{x}_i; \boldsymbol{W}') - f(\boldsymbol{x}_i; \boldsymbol{W})] \geq \ell'[y_i f(\boldsymbol{x}_i; \boldsymbol{W})] \cdot y_i \cdot \langle \nabla f(\boldsymbol{x}_i; \boldsymbol{W}), \boldsymbol{W}' - \boldsymbol{W} \rangle - \varepsilon$$
$$= \textstyle\sum_{l=1}^{L} \langle \nabla_{\boldsymbol{W}_l} L_i(\boldsymbol{W}), \boldsymbol{W}_l' - \boldsymbol{W}_l \rangle - \varepsilon ,$$

where $\varepsilon := |\ell'[y_i f(\boldsymbol{x}_i; \boldsymbol{W})] \cdot y_i \cdot [f(\boldsymbol{x}_i; \boldsymbol{W}') - f(\boldsymbol{x}_i; \boldsymbol{W}) - \langle \nabla f(\boldsymbol{x}_i; \boldsymbol{W}), \boldsymbol{W}' - \boldsymbol{W} \rangle]|$. Then by upper-bounding $\varepsilon$ with Lemma 8 and the fact that $|\ell'[y_i f(\boldsymbol{x}_i; \boldsymbol{W})] \cdot y_i| \leq 1$, we have:

$$L_i(\boldsymbol{W}') - L_i(\boldsymbol{W}) \geq \sum_{l=1}^{L} \langle \nabla_{\boldsymbol{W}_l} L_i(\boldsymbol{W}), \boldsymbol{W}_l' - \boldsymbol{W}_l \rangle - \varepsilon$$

$$= \sum_{l=1}^{L} \langle \nabla_{\boldsymbol{W}_l} L_i(\boldsymbol{W}), \boldsymbol{W}_l' - \boldsymbol{W}_l \rangle - \mathcal{O}(1) .$$

$\square$

We need the following lemma to show that, the gradient of the neural network function can be upper bounded under near initialization.

**Lemma 10.** *Let* $\boldsymbol{W} \in \mathcal{B}(\boldsymbol{W}^{(0)}, \omega)$ *with* $\omega = \mathcal{O}((3\mathrm{Lip}_{\max} + 1)^{-(L-1)})$ *, for any* $i \in [N]$, *with probability at least* $1 - 2(L - l)\exp(-m/2) - l\exp(-\Omega(m)) - 2/m$, *it holds that:*

$$\|\nabla_{\boldsymbol{W}_l} f(\boldsymbol{x}_i; \boldsymbol{W})\|_2, \|\nabla_{\boldsymbol{W}_l} L_i(\boldsymbol{W})\|_2 \leq \Theta(3\mathrm{Lip}_{\max} + 1)^{L-l} .$$

*Proof.* According to the triangle inequality and definition of operator norm, we have:

$$\|\nabla_{\boldsymbol{W}_l} f(\boldsymbol{x}_i; \boldsymbol{W})\|_2 = \left\| \boldsymbol{f}_{i,l-1} \boldsymbol{W}_L \prod_{r=l+1}^{L-1} (\boldsymbol{D}_{i,r} \boldsymbol{W}_r + \alpha_{r-1} \boldsymbol{I}_{m \times m}) \boldsymbol{D}_{i,l} \right\|_2$$

$$\leq \|\boldsymbol{f}_{i,l-1}\|_2 \left\| \boldsymbol{W}_L \prod_{r=l+1}^{L-1} (\boldsymbol{D}_{i,r} \boldsymbol{W}_r + \alpha_{r-1} \boldsymbol{I}_{m \times m}) \boldsymbol{D}_{i,l} \right\|_2$$

$$\leq \|\boldsymbol{f}_{i,l-1}\|_2 \|\boldsymbol{W}_L\|_2 \prod_{r=l+1}^{L-1} (\|\boldsymbol{D}_{i,r}\|_2 \|\boldsymbol{W}_r\|_2 + \alpha_{r-1}) \|\boldsymbol{D}_{i,l}\|_2 .$$

By Lemma 2 and Lemma 6, with probability at least $1 - 2(L - l - 1)\exp(-m/2) - l\exp(-\Omega(m)) - 2/m$ we have $\|\boldsymbol{f}_{i,l-1}\|_2 = \Theta(1)$, $\left\|\boldsymbol{W}_i^{(0)}\right\|_2 \leq 3$ for $i = l + 1, \cdots, L - 1$, $\left\|\boldsymbol{W}_L^{(0)}\right\|_2 \leq \sqrt{2}$ and $\|\boldsymbol{D}\|_2 \leq \mathrm{Lip}_{\max}$ due to $\sigma'$ is bounded, then we have:

$$\|\nabla_{\boldsymbol{W}_l} f(\boldsymbol{x}_i; \boldsymbol{W})\|_2 \leq \Theta(1)(3\mathrm{Lip}_{\max} + 1)^{L-l-1} \sqrt{2}\mathrm{Lip}_{\max} = \Theta(3\mathrm{Lip}_{\max} + 1)^{L-l-1} ,$$

which implies:

$$\|\nabla_{\boldsymbol{W}_l} L_i(\boldsymbol{W})\|_2 \leq |\ell'[y_i \cdot f(\boldsymbol{x}_i; \boldsymbol{W})] \cdot y_i| \cdot \|\nabla_{\boldsymbol{W}_l} f(\boldsymbol{x}_i; \boldsymbol{W})\|_2 \leq \|\nabla_{\boldsymbol{W}_l} f(\boldsymbol{x}_i; \boldsymbol{W})\|_2 \leq \Theta(3\mathrm{Lip}_{\max} + 1)^{L-l-1} ,$$

where we use the fact that $|\ell'[y_i f(\boldsymbol{x}_i; \boldsymbol{W})] \cdot y_i| \leq 1$. $\square$

We need the following lemma to show that, the cumulative loss can be upper bounded under small changes on the parameters (i.e., weights).

**Lemma 11.** *For any* $\epsilon, \delta, R > 0$, *there exists:*

$$m^\star = \frac{(3\mathrm{Lip}_{\max} + 1)^{4L-4} L^2 R^4}{4\varepsilon^2} ,$$

*such that if* $m \geq m^*(\epsilon, \delta, R, L)$, *then with probability at least* $1 - \delta$ *over the randomness of* $\boldsymbol{W}^{(1)}$, *for any* $\boldsymbol{W}^* \in \mathcal{B}(\boldsymbol{W}^{(1)}, Rm^{-1/2})$, *Algorithm 1 with* $\gamma = \varepsilon/[m(3\mathrm{Lip}_{\max} + 1)^{2L-2}]$, $N = LR^2(3\mathrm{Lip}_{\max} + 1)^{2L-2}/(2\varepsilon^2)$, *the cumulative loss can be upper bounded by:*

$$\sum_{i=1}^{N} L_i(\boldsymbol{W}^{(i)}) \leq \sum_{i=1}^{N} L_i(\boldsymbol{W}^*) + 3N\epsilon .$$

**Remark:** Discussion on the required width $m$ refer to Appendix E.

*Proof.* Set $\omega = 1/(3\text{Lip}_{\max}+1)^{L-1}$ such that the conditions on $\omega$ given in Lemmas 7, 8, 9 and 10 hold. It is easy to see that as long as $m \geq R^2(3\text{Lip}_{\max}+1)^{2L-2}$, we have $\boldsymbol{W}^* \in \mathcal{B}(\boldsymbol{W}^{(1)}, \omega)$. We now show that under our parameter choice, $\boldsymbol{W}^{(1)}, \ldots, \boldsymbol{W}^{(N)}$ are inside $\mathcal{B}(\boldsymbol{W}^{(1)}, \omega)$ as well.

This result follows by simple induction. Clearly we have $\boldsymbol{W}^{(1)} \in \mathcal{B}(\boldsymbol{W}^{(1)}, \omega)$. Suppose that $\boldsymbol{W}^{(1)}, \ldots, \boldsymbol{W}^{(i)} \in \mathcal{B}(\boldsymbol{W}^{(1)}, \omega)$. Then by Lemma 10, for $l \in [L]$, we have $\|\nabla_{\boldsymbol{W}_l} L_i(\boldsymbol{W}^{(i)})\|_2 \leq \Theta(3\text{Lip}_{\max}+1)^{L-l-1}$.

Therefore:

$$\big\|\boldsymbol{W}_l^{(i+1)} - \boldsymbol{W}_l^{(1)}\big\|_2 \leq \sum_{j=1}^{i} \big\|\boldsymbol{W}_l^{(j+1)} - \boldsymbol{W}_l^{(j)}\big\|_2 \leq \Theta\big((3\text{Lip}_{\max}+1)^{L-l-1}\gamma N\big).$$

Plugging in our parameter choice $\gamma = \varepsilon/[m(3\text{Lip}_{\max}+1)^{2L-2}]$, $N = LR^2(3\text{Lip}_{\max}+1)^{2L-2}/(2\varepsilon^2)$ for some small enough absolute constant $\nu$ provides:

$$\big\|\boldsymbol{W}_l^{(i+1)} - \boldsymbol{W}_l^{(1)}\big\|_{\text{F}} \leq \Theta\bigg(\sqrt{m}(3\text{Lip}_{\max}+1)^{L-l-1}\frac{LR^2}{2m\varepsilon}\bigg) \leq \omega\,,$$

where the last inequality holds as long as $m \geq (3\text{Lip}_{\max}+1)^{4L-4}L^2R^4/(4\varepsilon^2)$. Therefore by induction we see that $\boldsymbol{W}^{(1)}, \ldots, \boldsymbol{W}^{(N)} \in \mathcal{B}(\boldsymbol{W}^{(1)}, \omega)$. As a result, the conditions of Lemmas. 7, 8, 9 and 10 are satisfied for $\boldsymbol{W}^*$ and $\boldsymbol{W}^{(1)}, \ldots, \boldsymbol{W}^{(N)}$.

In the following, we utilize the results of Lemmas 7, 8, 9 and 10 to prove the bound of cumulative loss. First of all, by Lemma 9, we have:

$$L_i(\boldsymbol{W}^{(i)}) - L_i(\boldsymbol{W}^*) \leq \Big\langle \nabla_{\boldsymbol{W}} L_i(\boldsymbol{W}^{(i)}), \boldsymbol{W}^{(i)} - \boldsymbol{W}^* \Big\rangle + \epsilon$$

$$= \sum_{l=1}^{L} \frac{\Big\langle \boldsymbol{W}_l^{(i)} - \boldsymbol{W}_l^{(i+1)}, \boldsymbol{W}_l^{(i)} - \boldsymbol{W}_l^* \Big\rangle}{\gamma} + \epsilon\,.$$

Note that for the matrix inner product we have the equality $2\langle \boldsymbol{A}, \boldsymbol{B}\rangle = \|\boldsymbol{A}\|_{\text{F}}^2 + \|\boldsymbol{B}\|_{\text{F}}^2 - \|\boldsymbol{A} - \boldsymbol{B}\|_{\text{F}}^2$. Applying this equality to the right hand side above provides:

$$L_i(\boldsymbol{W}^{(i)}) - L_i(\boldsymbol{W}^*) \leq \sum_{l=1}^{L} \frac{\|\boldsymbol{W}_l^{(i)} - \boldsymbol{W}_l^{(i+1)}\|_{\text{F}}^2 + \|\boldsymbol{W}_l^{(i)} - \boldsymbol{W}_l^*\|_{\text{F}}^2 - \|\boldsymbol{W}_l^{(i+1)} - \boldsymbol{W}_l^*\|_{\text{F}}^2}{2\gamma} + \epsilon\,.$$

By Lemma 10, for $l \in [L]$ we have $\|\boldsymbol{W}_l^{(i)} - \boldsymbol{W}_l^{(i+1)}\|_{\text{F}} \leq \gamma\sqrt{m}\|\nabla_{\boldsymbol{W}_l} L_i(\boldsymbol{W}^{(i)})\|_2 \leq \Theta(\gamma\sqrt{m}(3\text{Lip}_{\max}+1)^{L-l-1})$.

Therefore:

$$L_i(\boldsymbol{W}^{(i)}) - L_i(\boldsymbol{W}^*) \leq \sum_{l=1}^{L} \frac{\|\boldsymbol{W}_l^{(i)} - \boldsymbol{W}_l^*\|_{\text{F}}^2 - \|\boldsymbol{W}_l^{(i+1)} - \boldsymbol{W}_l^*\|_{\text{F}}^2}{2\gamma} + \Theta((3\text{Lip}_{\max}+1)^{2L-2}\gamma m) + \epsilon\,.$$

Telescoping over $i = 1, \ldots, N$, we obtain:

$$\frac{1}{N}\sum_{i=1}^{N} L_i(\boldsymbol{W}^{(i)}) \leq \frac{1}{N}\sum_{i=1}^{N} L_i(\boldsymbol{W}^*) + \sum_{l=1}^{L} \frac{\|\boldsymbol{W}_l^{(1)} - \boldsymbol{W}_l^*\|_{\text{F}}^2}{2N\gamma} + \Theta((3\text{Lip}_{\max}+1)^{2L-2}\gamma m) + \epsilon$$

$$\leq \frac{1}{N}\sum_{i=1}^{N} L_i(\boldsymbol{W}^*) + \frac{LR^2}{2\gamma mN} + \Theta((3\text{Lip}_{\max}+1)^{2L-2}\gamma m) + \epsilon\,,$$

where in the first inequality we simply remove the term $-\|\boldsymbol{W}_l^{(N+1)} - \boldsymbol{W}_l^*\|_{\text{F}}^2/(2\gamma)$ to obtain an upper bound, the second inequality follows by the assumption that $\boldsymbol{W}^* \in \mathcal{B}(\boldsymbol{W}^{(1)}, Rm^{-1/2})$. Plugging in the parameter choice $\gamma = \varepsilon/[m(3\text{Lip}_{\max}+1)^{2L-2}]$, $N = LR^2(3\text{Lip}_{\max}+1)^{2L-2}/(2\varepsilon^2)$, then:

$$\frac{1}{N}\sum_{i=1}^{N} L_i(\boldsymbol{W}^{(i)}) \leq \frac{1}{N}\sum_{i=1}^{N} L_i(\boldsymbol{W}^*) + 3\epsilon\,,$$

which finishes the proof. $\qquad\square$

## D.2 Proof of Theorem 3

*Proof.* By Lemmas 8, Lemma 11 and Theorem 3.3, Lemma 4.4, Corollary 3.10 in [Cao and Gu, 2019], let $C_1(L) = \sqrt{L}/(3\text{Lip}_{\max}+1)^{L-1}$ and $C_2(L) = \sqrt{L}(3\text{Lip}_{\max}+1)^{L-1}$, bring in our $\gamma$ and $N$ with a not very large $L$, we have:

$$\mathbb{E}[\ell_{\mathcal{D}}^{0-1}(\hat{\boldsymbol{W}})] \leq \tilde{\mathcal{O}}\left(C_2\sqrt{\frac{\boldsymbol{y}^\top(\boldsymbol{K}^{(L)})^{-1}\boldsymbol{y}}{N}}\right) + \mathcal{O}\left(\sqrt{\frac{\log(1/\delta)}{N}}\right).$$

According to the courant minimax principle [Golub and Van Loan, 1996]: $\frac{1}{\lambda_{\min}(\boldsymbol{K}^{(L)})} = \lambda_{\max}((\boldsymbol{K}^{(L)})^{-1}) = \max \frac{\boldsymbol{y}^\top(\boldsymbol{K}^{(L)})^{-1}\boldsymbol{y}}{\boldsymbol{y}^\top \boldsymbol{y}}$, that means $\boldsymbol{y}^\top((\boldsymbol{K}^{(L)})^{-1})\boldsymbol{y} \leq \frac{\boldsymbol{y}^\top \boldsymbol{y}}{\lambda_{\min}(\boldsymbol{K}^{(L)})}$, then we have the final bound:

$$\mathbb{E}[\ell_{\mathcal{D}}^{0-1}(\hat{\boldsymbol{W}})] \leq \tilde{\mathcal{O}}\left(C_2\sqrt{\frac{\boldsymbol{y}^\top \boldsymbol{y}}{\lambda_{\min}(\boldsymbol{K}^{(L)})N}}\right) + \mathcal{O}\left(\sqrt{\frac{\log(1/\delta)}{N}}\right).$$

$\square$

# E  Discussion on the key points and the motivation of the NTK analysis

In this section, we discuss the motivation and few key points in the proof of this paper and we also explain how the proof differs from previous results.

**The motivation for studying the minimum eigenvalue of NTK**:

To make this clearer, we provide an illustrative example on the significance of the minimum eigenvalue. Let us consider the square loss $\Phi(\theta) = \frac{1}{2}\sum_{i=1}^n \|f(x_i) - y_i\|^2$. A simple calculation shows that $\Phi(\theta) \leq \frac{[\nabla\Phi(\theta)]^2}{2\lambda_{\min}(K)}$. Thus if the minimum eigenvalue of NTK is strictly greater than 0, then minimizing the gradient on the LHS will drive the loss to zero. The larger the minimum eigenvalue, the smaller the loss.

Therefore, in this work, we are using the minimum eigenvalue to derive the generalization bound of NAS.

**Key points in the proof**:

- *Minimum eigenvalue*: Our proof framework is motivated by Nguyen et al. [2021] on minimal eigenvalue of NTK of ReLU neural networks. However, our proofs differ from them in two aspects. Firstly, as we discussed in sec. 1, extension to mixed activation functions is non-trivial due to the special properties of ReLU. More importantly, we remark that the lower bound of the minimal eigenvalue of NTK in [Nguyen et al., 2021, Theorem 3.2] holds with probability at least $1 - Ne^{-\Omega(d)} - N^2 e^{-\Omega\left(dN^{-\frac{2}{r-0.5}}\right)}$, where $r \geq 2$ is some constant. It can be found that, this concentration probability decreases as the number of training data increases. Thus, it could be negative for a large $N$. This is due to the use of Gershgorin circle theorem leading to a loose probability estimation. Instead, in this paper, we do not use this theorem, and we develop a tighter estimation based on Yaskov [2014] under the assumption of isotropic data distribution. Accordingly, we achieve the reasonable $1 - e^{-d}$ probability, *c.f.* Theorem 1.

- *Generalization*: Our proof framework is based on Cao and Gu [2019] for generalization guarantees of deep ReLU neural networks requiring $m = \Omega(L^{56})$. Their results cannot be directly extended to other activation functions as the nice homogeneity and the derivative property of ReLU are used in their proof. To make our result feasible to various activation functions, we employ Lipschitz continuous properties of all activation functions, and achieve the generalization guarantees with $m = \Omega(4^{4L})$, *c.f.* Theorem 3 and Lemma 11. Admittedly, our result is in an exponential increasing order of the depth. However, in practice, the depth of neural networks in NAS is usually smaller than 20, or even 10 [Liu et al., 2018, Dong

et al., 2021], which leads to $4^{4L} \ll L^{56}$ in this case when compared to their result. This result makes our theory reasonable and fair for NAS.

## F  Auxiliary numerical validations

### F.1  Dataset details and algorithm

We describe here the datasets that we have used for the numerical validation of our theory. Those are the following five datasets:

1. *Fashion-MNIST* [Xiao et al., 2017] includes grayscale images of clothing. The training set consists of $60,000$ examples and the test set of $10,000$ examples. The resolution of each image is $28 \times 28$, with each image belonging to one of the 10 classes.

2. *MNIST* [Lecun et al., 1998] includes handwritten digits images. MNIST has a training set of $60,000$ examples and a test set of $10,000$ examples. The resolution of each image is $28 \times 28$.

3. *CIFAR-10 and CIFAR-100 [Krizhevsky et al., 2014]* depicts images of natural scenes. CIFAR-100 has a training set of $50,000$ examples and a test set of $10,000$ examples. The resolution of each RGB image is $32 \times 32$.

4. *ImageNet-16* [Chrabaszcz et al., 2017] is the down-sampled version of ImageNet [Deng et al., 2009] with image size $16 \times 16$ on 120 classes.

Our Eigen-NAS algorithm used in sec. 5.2 is summarized as below.

---
**Algorithm 2:** Eigen-NAS Algorithm

---
**Require:** Search space $\mathcal{S}$, training data $\mathcal{D}_{tr} = \{(\boldsymbol{x}_i, y_i)_{i=1}^N\}$, validation data $\mathcal{D}_{val} = \{(\boldsymbol{x}_j, y_j)_{j=1}^{N_v}\}$.
  Initialize max_iteration $= M$
  Initialize candidate set $\mathcal{C} = []$
  **for** search_iteration in $1, 2, \ldots$,max_iteration **do**
    Randomly sample architecture $s$ from search space $\mathcal{S}$.
    Compute $Eigen :=$ minimum eigenvalue of NTK.
    $\mathcal{C}$.append($s, Eigen$)
    update $\mathcal{C}$ to kept top-K best architectures
  **end for**
  $s^\star = best_s(\mathcal{C}, \mathcal{D}_{tr}, \mathcal{D}_{val})$ # Choose the best architecture based on validation error after training 20 epochs.
  **Output** $s^\star$

---

### F.2  Compared algorithms

We provide a thorough comparison with the following baselines:

1. *Classical network:* ResNet [He et al., 2016], which is the default baseline used widely in image-related tasks.
2. *Reinforcement learning based algorithm:* NAS-RL [Zoph and Le, 2017] with the validation accuracy as a reward, which is an classical and representative NAS Algorithm.
3. *Differentiable algorithm:* DARTS [Liu et al., 2019b][4], which is the earliest and basic gradient-based NAS algorithm.
4. *Train-free algorithms using metrics to guide NAS:* A new type of NAS algorithm, they use some special metrics to pick models directly from candidate models. Common Train-free algorithms are: NASWOT [Mellor et al., 2021] using the output of ReLU; TE-NAS [Chen et al., 2021] leveraging the spectrum of NTK and linear partition of the input space; KNAS [Xu et al., 2021] employing the Frobenius norm of NTK. Our Eigen-NAS algorithm also belongs to this type.

---
[4]We directly use the results from  Xu et al. [2021].

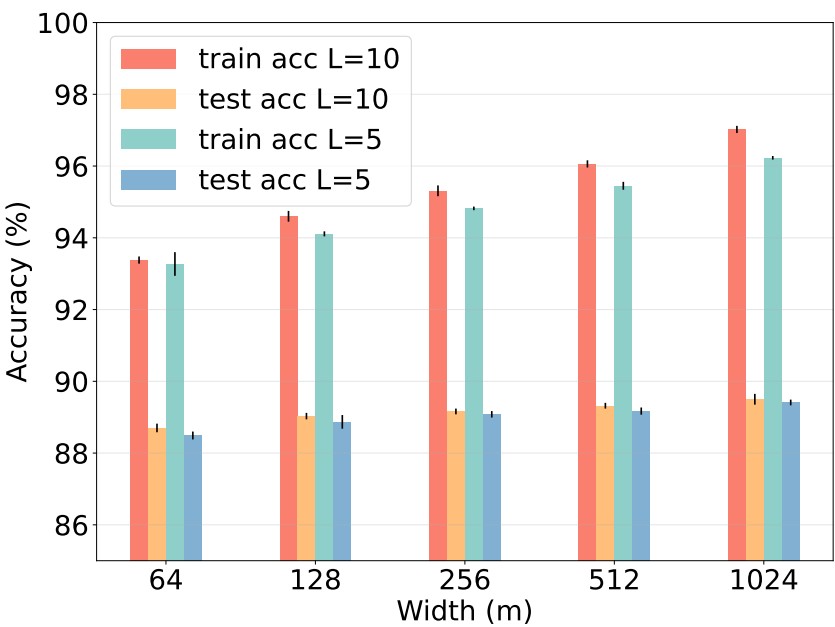

Figure 2: The accuracy of neural networks by NAS under different widths and depths.

### F.3 Training/test accuracy of DNNs by NAS

Here we evaluate the classification results with 5 runs of the obtained architecture by DARTS under varying widths $m \in \{64, 128, 256, 512, 1024\}$ and depths $L \in \{5, 10\}$ on Fashion-MNIST. Figure 2 shows that nearly 90% accuracy is achieved on the test set under different depth and width settings. The result is competitive on FC/residual networks within 10 layers and without training tricks, e.g., data augmentation, batch norm and drop out. We find that when compared to the depth, the network width also contributes on test accuracy. As suggested by Equation (2), the amount of parameters in the neural network is approximately proportional to the depth, but squared to the width.

### F.4 Simulation of minimum eigenvalues of NTK

We calculate the minimum eigenvalue of each NTK matrix under different architectures with activation functions, skip connections and depths, according to Lemma 1. We consider four special cases on skip connections: a) no skip connections with $\boldsymbol{\alpha} = \boldsymbol{0}$, i.e., fully connected neural network in Figure 3(a); b) skip connections between all consecutive layers with $\boldsymbol{\alpha} = \boldsymbol{1}$, in Figure 3(b); c) the alpha of the first half (of the network) is 1 and the alpha of the second half is 0 shown in Figure 4(a); d) The alpha of the first half is 0 and the alpha of the second half is 1. The results are shown in Figure 4(b).

Figure 3(a) indicates that as the network depth increases, the minimum eigenvalue of NTK will become larger when LeakyReLU, ReLU, Swish and Tanh employed, but Sigmoid leads to a decreasing minimum eigenvalue, which is consistent with the upper bound shown in Theorem 1. The LeakyReLU, ReLU and Swish generate the fastest increasing rate of depth, while Tanh and Sigmoid are slow, which coincides with the derived lower bound in Theorem 1 and previous work Bietti and Bach [2021]. Figure 3(b) shows that, under the skip connection, the tendency of the minimum eigenvalue of NTK is similar to that of FC neural networks when various activation functions are employed. However, the specific values and the growth rate are significantly larger than FC neural networks. This result is consistent with the conclusion we state in Theorem 1 about skip layers leading to the increase of minimum eigenvalue of NTK with respect to the depth. Moreover, Figure 4 show similar growth speed.

Then, we plot the comparison figure of NTK under above two settings and two settings in main paper for the same activation function in Figure 5. In addition to reconfirming the order between different activation functions, we can also see that the effect of adding an activation layer in the second half

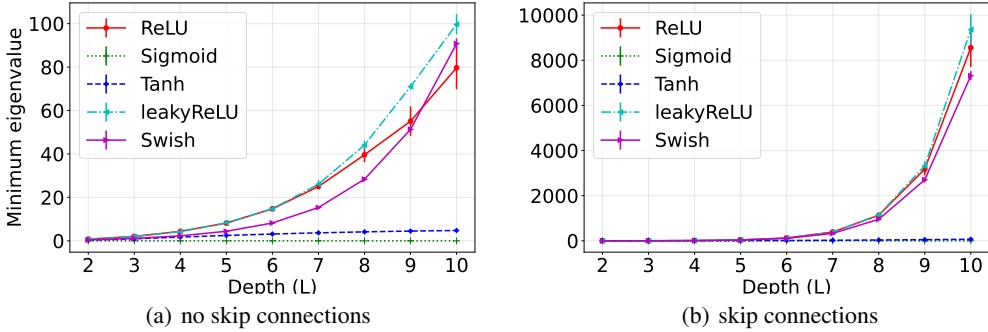

(a) no skip connections  (b) skip connections

Figure 3: Minimum eigenvalue of NTK *vs.* depth ($L$) under various activation functions with/without skip connections in each layer.

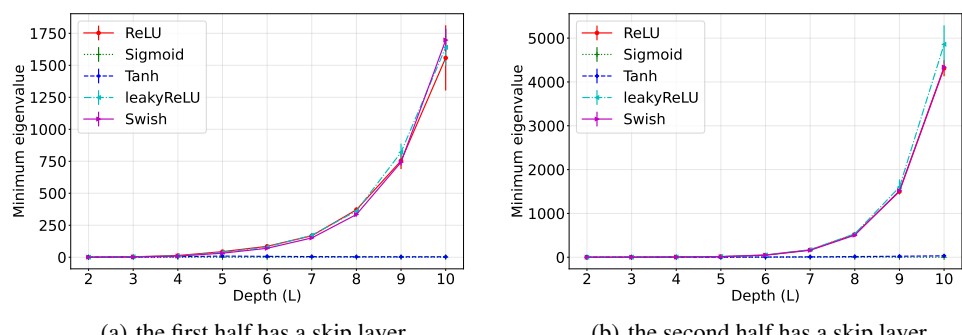

(a) the first half has a skip layer  (b) the second half has a skip layer

Figure 4: Minimum eigenvalue of NTK *vs.* depth ($L$) under various activation functions with/without skip connections in each layer.

of the 10-layer neural network is better than the first half of the neural network. This verifies the experimental results in Figure 1(b).

### F.5  Additional experiments on NAS-Bench-101 and ranking correlations

In this section, we conduct more experiments on two new benchmarks NAS-Bench-101 [Ying et al., 2019] and Network Design Spaces (NDS) [Radosavovic et al., 2019] using the same setting as sec. 5.2.

Table 6 provides a comparison of the accuracy of Eigen-NAS, KNAS and NASWOT on four new search spaces. For all of four search spaces, our method achieves the best results with $1\% - 2\%$ accuracy improvement.

Moreover, we conduct more detailed experiments using the CIFAR-10 dataset on NAS-Bench-101. Table 7 provides the running time and Kendall rank correlation coefficient between minimum eigenvalues and accuracy for the above three train-free NAS algorithm. We can see that our Eigen-NAS method can get the best rank correlation coefficient with the fastest speed among three methods. The scatter plot of the relationship between the minimum eigenvalue and the accuracy is shown in Figure 6.

Table 6: New results on NAS-Benchmark-101, NDS-DARTS and NDS-PNAS using CIFAR-10 and ImageNette2, a subset of ImageNet.

| Benchmark | NAS-Bench-101 | NDS-DARTS | NDS-PNAS | NDS-PNAS |
|---|---|---|---|---|
| Dataset | CIFAR-10 | CIFAR-10 | CIFAR-10 | ImageNette2 |
| **Eigen-NAS** ($k = 20$) | **92.7**% | **92.6**% | **93.8**% | **69.2**% |
| KNAS ($k = 20$) | 91.7% | 90.1% | 91.7% | 67.3% |
| NASWOT | 91.3% | 90.6% | 93.3% | 68.4% |

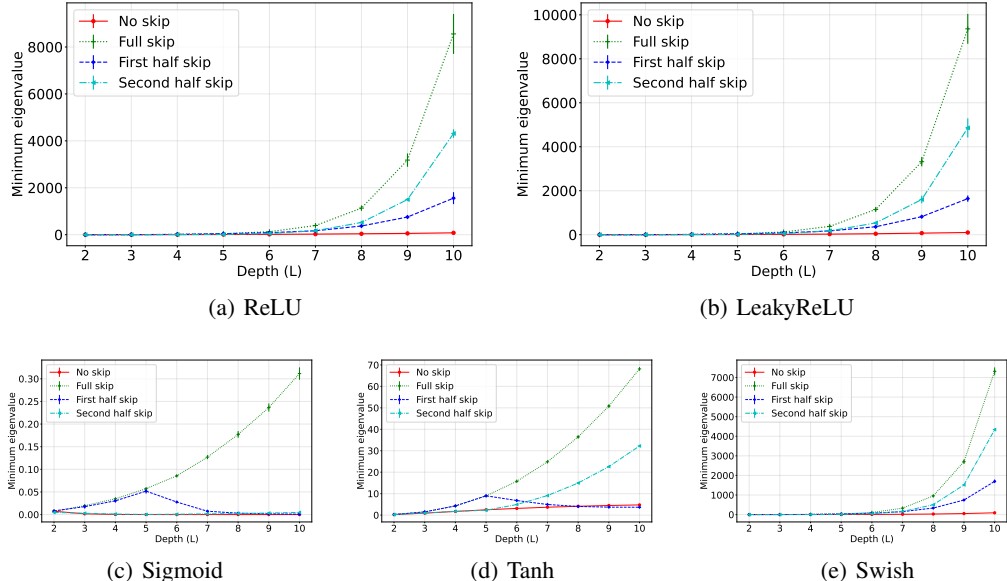

(a) ReLU

(b) LeakyReLU

(c) Sigmoid

(d) Tanh

(e) Swish

Figure 5: Minimum eigenvalue of NTK for different activation function. The red line have skip connections in each layer, green line does not contain any skip connections, the blue line represents the skip connections in the first half and the cyan line represents the skip connections in the second half.

Table 7: Running time (in Second) and the Kendall rank correlation coefficient on NAS-Bench-101, CIFAR-10 (the larger the absolute value of Rank correlation, the stronger the correlation between the guide used by the algorithm and the network accuracy).

| Method | Eigen-NAS ($k = 20$) | KNAS ($k = 20$) | NASWOT |
|---|---|---|---|
| Running time | **1136** | 1967 | 1468 |
| Rank correlation | **−0.355** | 0.309 | −0.313 |

## F.6 Transfer learning experiment

Here we evaluate the proposed NAS framework on transfer learning. The algorithm from sec. 5.1 is employed for this experiment, e.g., the same search space and search strategy. The experiment setting is the following: we train the model on FashionMNIST for 20 epochs, then we use the pretrained weights and fine-tune them for 5 epochs on MNIST, with repeated three times.

Table 8 show that, after the fine-tuning for just 2 epochs, the method obtains up to $95\%$ accuracy and after fine-tuning for 5 epochs it obtains up to $97\%$ accuracy. This verifies our intuition that the proposed NAS framework can obtain architectures that generalize well beyond the dataset they were optimized on.

## F.7 DARTS experiment on CNN

Our theory relies on fully-connected matrices and we have indeed verified experimentally the validity of our theoretical findings. To scrutinize our method even further, we attempt to extend our results to the popular convolutional neural networks. We believe this will provide some further insights

Table 8: Transfer learning result of our network for different width ($m$) which training in FashionMNIST (domain dataset) for 20 epochs and then training in MNIST (target dataset) for 2 or 5 epochs. (the accuracy in the table are displayed in percentages)

| Epochs | $m = 64$ | $m = 128$ | $m = 256$ | $m = 512$ | $m = 1024$ |
|---|---|---|---|---|---|
| 20 + 2 | $94.13 \pm 0.64$ | $95.18 \pm 0.25$ | $94.73 \pm 0.22$ | $94.40 \pm 0.80$ | $\mathbf{95.41 \pm 0.03}$ |
| 20 + 5 | $95.73 \pm 0.28$ | $96.12 \pm 0.32$ | $96.73 \pm 0.29$ | $96.73 \pm 0.11$ | $\mathbf{96.96 \pm 0.22}$ |

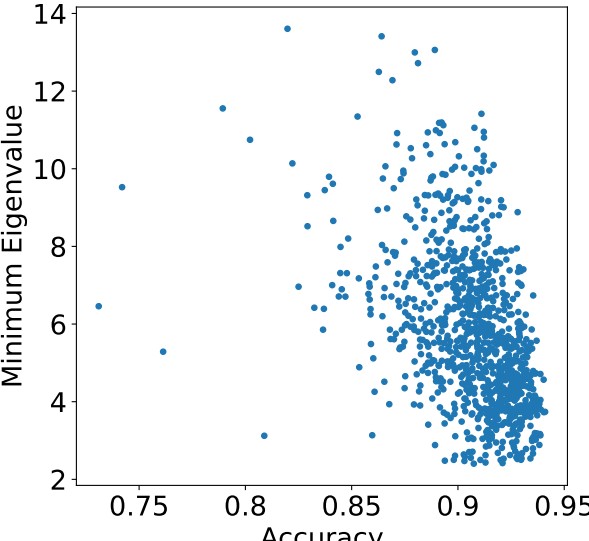

Figure 6: The standard scatter plot on the kendall rank correlation coefficient.

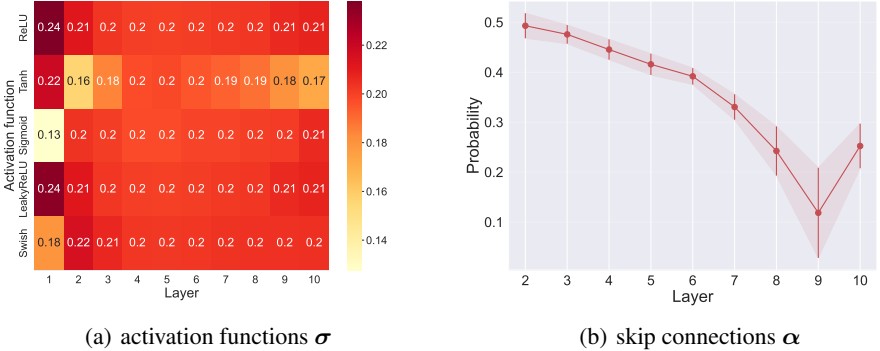

(a) activation functions $\boldsymbol{\sigma}$

(b) skip connections $\boldsymbol{\alpha}$

Figure 7: Architecture search results on activation functions indicated by the probability of $\boldsymbol{\sigma}$ in (a) and skip connections indicated by $\boldsymbol{\alpha}$ in (b). We notice that for each layer, ReLU and LeakyReLU are selected with the higher probability.

on future extensions of our theory. In particular, we use DARTS (similarly with the experiment in sec. 5.1) with convolutional layers. The standard dataset of CIFAR-10 is selected; the details of the dataset are shared in Appendix F.1. The search space and search strategy follow sec. 5.1 with one differentiating point: we use convolutional layers instead of fully connected layers in Equation (2).

We select DARTS on a Convolutional Neural Network with $L = 10$ and $m = 1024$, while we repeat the experiment for 5 times. After training, the probability of these activation functions and skip connections in each layer are reported in Figure 7(a) and 7(b), respectively. Compared with the Figure 1, the activation function search exhibits similar characteristics with the results of the fully connected network. Namely: (1) ReLU and LeakyReLU have the highest probability to be selected, (2) the difference of probability between different activation functions in the first layer is the largest. But for skip layer search, CNN exhibits the opposite results with fully connected network, that is, almost all of the skip connections have a probability of being selected less than $50\%$.

Based on the above results, our theory can still explain some of the phenomena observed in CNNs, e.g., activation functions search. Nevertheless, our theory on skip connections search on CNNs mismatches with experimental demonstration in practice to some extent, which motivates us to conduct a refined analysis on CNNs for NAS.

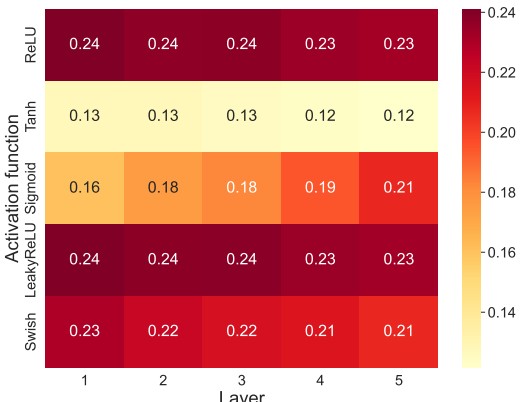

Figure 8: Architecture search results using $\beta$-DARTS on activation functions indicated by the probability of $\sigma$. We notice that for each layer, ReLU and LeakyReLU are selected with the higher probability.

### F.8  $\beta$-DARTS experiment on MLP

In this section, we use an improved DARTS-based algorithm, $\beta$-DARTS [Ye et al., 2022], for doing the activation function search. Our experiments are performed on a 5-layers MLP and the experimental results are presented in Figure 8. Compared with the results of DARTS in Figure 1, the experimental results of $\beta$-DARTS indicate that the probability difference between different activation functions is smaller, which may verify that DART is more easily to overfit . This is also the advantage mentioned in the $\beta$-DARTS paper.

## G  Societal impact

This is a theoretical work that derived generalization bounds for the architectures obtained by NAS. As such, we do not expect our work to have negative societal bias, as we do not focus on obtaining state-of-the-art results in a particular task. On the contrary, our work can have various benefits for the community:

- We provide the first generalization bounds for the class of NAS architectures, which is expected to have a positive impact on the understanding and the application of such architectures.
- As we illustrate in sec. 5, we can use the minimum eigenvalue as a promising metric to guide NAS. This can lead to further investigation on techniques for efficient evaluation of NAS by avoiding solving the intensive bi-level optimization of NAS explicitly.

Nevertheless, we encourage researchers to further investigate the impact of different architectures and their inductive biases on the society.