# OpenReview forum: "Generalization Properties of NAS under Activation and Skip Connection Search"
_NeurIPS.cc/2022/Conference — NeurIPS 2022 Accept_

### Official Review · Reviewer_LWxJ · 2022-07-10

**Rating:** 6
**Confidence:** 3
**Soundness:** 3 good
**Presentation:** 3 good
**Contribution:** 3 good

**Summary:**

This paper presents theoretical analyses on Neural Architecture Search using the recent theory of NTK. Especially, it has provided the lower and upper bounds of the minimum eigenvalue of the NTK matrix as well as the generalization bound induced by the NTK matrix for the architectures in their pre-defined search space. Finally, based on these theoretical analyses, this paper develops a training-free NAS algorithm, namely Eigen-NAS. Eigen-NAS has shown its competitive performance in training-free NAS, which in turn also supports the theoretical analyses in this paper.

**Questions:**

1. This paper focuses on the theoretical study of NAS with only skip connection and activation function search mainly due to the non-trivial analysis of other general structures. Interestingly, both [R2] and Shu et al. [2020] have empirically shown that topology will be more important than diverse operations in NAS, and NAS with only topology search can also provide competitive performance compared with standard NAS. I suppose that these results may also serve as a good motivation to support why the theoretical study on only skip connection and activation function will still be attractive in the literature.
2. In line 98, to my best knowledge, Du et al. [2019] and Nguyen [2021] are studying the convergence of GD instead of SGD.
3. In the problem setting section of this paper, an explicit representation of the classification task (i.e., the binary classification task) and the possible value of $y$ (i.e., $y \in \\{-1,1\\}$) will be more helpful for the readers to understand this section. Similarly, the “cross-entropy loss” in line 111 can more specific, e.g., the cross-entropy loss with a sigmoid function to represent the probability of $y=1$.
4. In line 134, I suppose it should be “we conduct the skip connection search and activation function search independently”. Besides, I am curious about why skip connection search and activation function search can not be conducted jointly.
5. The lemma 1 and therefore the results based on Lemma 1 actually do not match the initialization applied in this paper (i.e., the Gaussian initialization with the variance of $1/m$ instead of $2/m$ in Tirer et al. [2021], Huang et al. [2020], Belfer et al. [2021].).
6. The claim about depth $L$ in line 208 may not be true because whether the minimum eigenvalue will increase or not with larger $L$ highly depends on the activation functions as shown in Figure 3,4. Moreover, I am curious about why the minimum eigenvalue of Tanh does not increase with an increasing $L$ especially when it achieves larger $\beta_1$ and $\beta_2$ and hence a larger upper bound of the minimum eigenvalue than ReLU and Sigmoid?
7. One of the comments about Table 2 is that the search cost can also be included to verify the improved efficiency of Eigen-NAS over KNAS. Another one is that the recent training-free NAS paper [R1] should also be included in the comparison or discussion. Because [R1] exploits the trace norm of the NTK matrix to indicate the performance of different architectures while this paper exploits the Frobious norm in its Eigen-NAS algorithm (line 324). They are similar.
8. An empirical study of how the number of skip connections layers (i.e., the topology) will affect the performance of neural architectures will be more interesting and related to the NAS area, which can also further verify the impact of the skip connection on the minimum eigenvalue of NTK (Theorem 1, 2) and also the generalization performance of candidate architecture (Theorem 3).

[R1] NASI: Label-and Data-agnostic Neural Architecture Search at Initialization

[R2] Exploring randomly wired neural networks for image recognition

**Limitations:**

This paper has provided a valuable discussion about limitations and future works.

**Strengths And Weaknesses:**

*Strengths*

1. This paper has provided non-trivial theoretical analyses on NAS using the recent theory of NTK, which may inspire more theoretical studies on NAS in the NAS area.
2. The Eigen-NAS inspired by their theoretical results shows competitive performance on different NAS benchmarks.

*Weaknesses*

1. This paper does not really study the convergence or optimization of NAS as what has been claimed in its title or abstract. Instead, it is mainly focusing on the generalization property of its restricted search space (i.e., a small search space compared with the standard NAS search space).
2. The motivation for why the minimum eigenvalue of NTK needs to be studied and how it is explicitly related to the generalization bound (i.e., Theorem 3) have not been well clarified. As a result, the study of the minimum eigenvalue of NTK in this paper may be less related. Using the minimum eigenvalue of NTK to bound the generalization performance of DNN in Theorem 3 will make the theoretical results in this paper more coherent.
3. This paper does not provide a clear or explicit interpretation of how the theoretical results in this paper can help us understand NAS (shown in the title). Instead, from my own perspective, this paper has mainly developed a generalization bound for NAS and then proposed a training-free NAS algorithm (i.e., Eigen-NAS) inspired by their theoretical results. And such a training-free NAS algorithm is similar to the one in [R1] using the trace norm of the NTK matrix.

---

> ### Author Response · Authors · 2022-08-02
> **Response to the reviewer LWxJ (2 out of 2)**
>
> Q8: One of the comments about Table 2 is that the search cost can also be included to verify the improved efficiency of Eigen-NAS over KNAS. Another one is that the recent training-free NAS paper [R1] should also be included in the comparison or discussion. Because [R1] exploits the trace norm of the NTK matrix to indicate the performance of different architectures while this paper exploits the Frobenius norm in its Eigen-NAS algorithm (line 324). They are similar.
>
> A8: We are thankful to the reviewer for their suggestion. We have provided the running times of three different train-free algorithms in Table 7, Appendix F.5. We will provide more experimental results about other NAS algorithms in the final version. And according to your suggestion, we added the experimental results of a new train-free algorithm [1] in Table 2, sec. 5.2. Our algorithm achieves better results on two of three datasets as follows:
>
> |Algorithm|CIFAR-10(\%)|CIFAR-100(\%)|ImageNet-16(\%)|
> |----------------|-------------------------------|-----------------------------|-------------------|
> |Eigen-NAS ($k=5$)|$93.43\pm0.08$|$69.92\pm1.82$|$45.53\pm0.06$|
> |Eigen-NAS ($k=20$)| $93.46\pm0.01$|$\mathbf{71.42}\pm0.63$|$\mathbf{45.54}\pm0.04$|
> |NASI(T)|$93.08\pm0.24$|$69.51\pm0.59$|$40.87\pm0.85$|
> |NASI(4T)|$\mathbf{93.55}\pm0.10$|$71.20\pm0.14$|$44.84\pm1.41$|
>
> ### References
>
> [1] Shu, Yao, Cai Shaofeng, Dai Zhongxiang, Ooi Beng, and Low Bryan Kian Hsiang. "NASI: Label- and Data-agnostic Neural Architecture Search at Initialization." ICLR (2022).
>
> [2] Du Simon S., Zhai Xiyu, Poczos Barnabas and Singh Aarti. "Gradient Descent Provably Optimizes Over-parameterized Neural Networks." ICLR (2019).
>
> [3] Allen-Zhu, Zeyuan, Li Yuanzhi, and Song Zhao. "A Convergence Theory for Deep Learning via Over-Parameterization." NeurIPS (2019).
>
> [4] Arora Sanjeev, Du Simon S., Hu Wei, Li Zhiyuan, and Wang Ruosong. "Fine-Grained Analysis of Optimization and Generalization for Overparameterized Two-Layer Neural Networks." ICML (2019).
>
> [5] Montanari Andrea, Zhong Yiqiao. "The Interpolation Phase Transition in Neural Networks: Memorization and Generalization under Lazy Training." arXiv:2007.12826.

---

> > ### Author Response · Authors · 2022-08-06
> > **Any remaining questions from reviewer LWxJ?**
> >
> > Dear reviewer LWxJ,
> >
> > We welcome your feedback and thoughtful questions. We agree with the reviewer on the need for a large search space and the significance of topology. However, we believe our work makes non-trivial steps towards expanding the search space by a) including popular activation functions and b) including the topology search through skip layers for the first time. As we identify in the experiments in Figure 1, Figure 7 along with the newly added Figure 8 (https://imgur.com/a/6L8eB3A), the theory seems to be verified in a number of empirical experiments.
> >
> > Therefore, we would like to check whether there are any additional questions we could answer. We appreciate the time and feedback of the reviewer since it enables us to improve the quality of our work and the clarity of our contributions.

---

> > > ### Comment · Reviewer_LWxJ · 2022-08-07
> > > **Thank the authors for their detailed response!**
> > >
> > > Dear authors,
> > >
> > > I appreciate that you have provided detailed explanations to my questions. Most of my questions have been addressed. But my concerns regarding the weaknesses of this paper have not been explicitly answered. I have read your general response, but it can not directly address my concerns. I hope the authors can give more explanations on this part.

---

> > > > ### Author Response · Authors · 2022-08-08
> > > > **Response to the reviewer LWxJ on the weakness and updates**
> > > >
> > > > Dear reviewer LWxJ,
> > > >
> > > > We are thankful for your feedback and thoughtful questions.
> > > > Based on your suggestions, we have renamed the title and polished the abstract/introduction/conclusion: now our work centers around NAS under a certain search space.
> > > >
> > > > Q1: This paper does not really study the convergence or optimization of NAS as what has been claimed in its title or abstract.
> > > >
> > > > A1: Thank you for following up on this point. We agree with the reviewer that our work focuses on the generalization properties of NAS under a certain search space.
> > > >
> > > > We adopted this title as our work provides the upper bounds on the generalization error of the network in a standard search space: activation function and skip-layer search in the residual network. We think our NTK analysis framework on mixed activations and skip-connection setup could be a possible way of understanding NAS, even under the somewhat limited search space. That is, our analysis can explain the existence of certain components. For instance, the recent work of [1] illustrates how skip-connections are a critical component across different NAS approaches (both bi-level and train-free). According to our analysis, networks with skip connections exhibit a lower generalization error.
> > > >
> > > > Even though we agree that our work does not attempt to explain every detail of NAS, we truly believe that *our (theoretical) outcomes can provide further insights into further developments of NAS*. To this end, to align with the feedback we have received, we have renamed our title to “**Generalization Properties of NAS under Activation and Skip Connection Search**” and polished our abstract/introduction/conclusion to reflect this new approach.
> > > >
> > > > We welcome the feedback of the reviewer on the revised approach.
> > > > ___
> > > >
> > > > Q2: Motivation for studying minimum eigenvalue and its relation to the generalization.
> > > >
> > > > A2: Building on top of our previous explanation, which can be found here [https://openreview.net/forum?id=aQySSrCbBul&noteId=GLQQcbDa93L], we provide an illustrative example next.  Considering the square loss $\Phi(\theta) = \frac{1}{2}\sum_{i=1}^{n} \left \| f(x_i) - y_i \right \|^2$, then a simple calculation shows that
> > > >
> > > >
> > > >  $\Phi(\theta) \leq \frac{[\nabla \Phi(\theta)]^2}{2\lambda_{\min}(K)}$
> > > >
> > > > . Thus if the minimum eigenvalue of NTK is strictly greater than 0, then minimizing the gradient on the LHS will drive the loss to zero. The larger the minimum eigenvalue, the smaller the loss.
> > > >
> > > > Regarding the relationship between minimum eigenvalue and generalization, according to the courant minimax principle [2], we have $\frac{1}{\lambda_{\min}(K)} =\lambda_{\max}(K^{-1}) = \max \frac{y^{\top}(K^{-1})y}{y^{\top}y}$, that is $y^{\top}(K^{-1})y \leq \frac{y^{\top}y}{\lambda_{\min}(K)}$. Combining this with Theorem 3 in sec. 4.5, a larger minimum eigenvalue corresponds to a tighter bound on the generalization error. In addition, a larger minimum eigenvalue means that the NTK of finite-width networks converges faster with width, that is, Theorem 3 has a looser requirement on the width m.
> > > >
> > > > In practice, our experiments show that some NAS algorithms such as DARTS and $\beta$-DARTS [3] have the characteristics of tending to search the architectures with a larger minimum eigenvalue of NTK. Besides, our train-free method based on the minimum eigenvalue achieved great results on some standard NAS benchmarks.
> > > >
> > > > Best regards,
> > > >
> > > > Authors
> > > > ___
> > > >
> > > > [1] Wan, Xingchen, Ru Binxin, Esperança Pedro M, and Li  Zhenguo "On Redundancy and Diversity in Cell-based Neural Architecture Search." ICLR (2022).
> > > >
> > > > [2] Courant, Richard, and Hilbert David, Method of Mathematical Physics, Vol. I, Wiley-Interscience (1989).
> > > >
> > > > [3] Ye, Peng, Li Baopu, Li Yikang, Chen Tao, Fan Jiayuan, and Ouyang Wanli "β -DARTS: Beta-Decay Regularization for Differentiable Architecture Search." CVPR (2022).

---

> > > > > ### Comment · Reviewer_LWxJ · 2022-08-08
> > > > > **Thanks to address my major concerns!**
> > > > >
> > > > > Dear authors,
> > > > >
> > > > > I am glad to know that my major concerns have been addressed at this time. I sincerely hope that your answer A2 can be placed before your studying on the minimum eigenvalue in the main paper so as to provide a good motivation for why studying the minimum eigenvalue is important and needed to understand NAS. I believe this can make the paper more coherent. Overall, I decided to increase my score to 6.

---

> > > > > > ### Author Response · Authors · 2022-08-09
> > > > > > **Response to the reviewer LWxJ on the requested changes**
> > > > > >
> > > > > > Dear reviewer LWxJ,
> > > > > >
> > > > > > We are thankful to the reviewer for the new feedback.
> > > > > >
> > > > > > According to your suggestion, we include a discussion of the motivation for studying the minimum eigenvalues of the NTK. We add a footnote in the introduction about this motivation, while we modify Theorem 3 to make it directly relate to minimum eigenvalues in the new version of the paper.
> > > > > >
> > > > > > Best,
> > > > > >
> > > > > > Authors

---

> ### Author Response · Authors · 2022-08-02
> **Response to the reviewer LWxJ (1 out of 2)**
>
> We are thankful to the reviewer LWxJ for their feedback. We respond below to their core remarks:
>
> Q1: Motivation for minimum eigenvalue and relationship to the generalization bound.
>
> A1: The minimum eigenvalue of NTK has been used in previous work to demonstrate the global convergence of gradient descent, such as two-layer networks [2], and deep networks with polynomially wide [3]. Besides, the minimum eigenvalue of NTK is also used to prove generalization bounds [4] and memory ability [5]. These works show that understanding the minimum eigenvalues of NTK is a significant problem. This is why our work focuses on the minimum eigenvalue of NTK.
>
> ____
>
> Q2: Topology will be more important than diverse operations in NAS? An empirical study of how the number of skip connections layers (i.e., the topology) will affect the performance of neural architectures will be more interesting and related to the NAS area, which can also further verify the impact of the skip connection on the minimum eigenvalue of NTK (Theorem 1, 2) and also the generalization performance of candidate architecture (Theorem 3).
>
> A2: We agree that neural topology is interesting, which is why we consider the skip layer in our theoretical framework. Even though it is non-trivial to theoretically analyze the impact of topologies with higher degrees of freedom on network performance, we want to consider this as part of our future work.
>
> Experimentally, our experimental results of skip layer search on MLP and CNN are shown in Figure. 1, sec. 5.1 and Figure. 7, Appendix. F.7. In addition, for MLPs with different skip layers, the simulations of the minimum eigenvalue of the NTK matrices are shown in the Figures. 3, 4, 5, Appendix. F. 4.
> ____
>
> Q3: In line 98, to my best knowledge, Du et al. [2019] and Nguyen [2021] are studying the convergence of GD instead of SGD.
>
> A3: We are thankful for the reviewer's reminder, we have revised the content of the paper here.
>
> ____
>
> Q4: In the problem setting section of this paper, an explicit representation of the classification task (i.e., the binary classification task) and the possible value of y (i.e.,y∈{−1,1}) will be more helpful for the readers to understand this section. Similarly, the “cross-entropy loss” in line 111 can be more specific, e.g., the cross-entropy loss with a sigmoid function to represent the probability of y=1.
>
> A4: In fact, the problem of our paper is not limited to binary classification tasks, so using binary classification tasks will mislead readers. But we follow the reviewer's suggestion and polish this part to make the meaning clear.
>
> ____
> Q5: In line 134, I suppose it should be “we conduct the skip connection search and activation function search independently”. Besides, I am curious about why skip connection search and activation function search can not be conducted jointly.
>
> A5:
> In our problem, the activation function search is exclusive as only one activation function must exist between two adjacent layers; while the skip layer can exist or not. If we jointly search for activation functions and skip connections, this results in a competitive search, where sometimes a skip layer is selected but activation functions are missing. In such a case, we cannot ensure that the search runs on the space for our theory. Therefore, for simplicity, we conduct the search process independently.
>
> ____
> Q6: The lemma 1 and therefore the results based on Lemma 1 actually do not match the initialization applied in this paper (i.e., the Gaussian initialization with the variance of 1/m instead of 2/m in Tirer et al. [2021], Huang et al. [2020], Belfer et al. [2021].).
>
> A6: Exactly. This discrepancy has already been considered in our proof. In fact, the difference in constants in initialization does not affect our neural network training in the NTK regime.
> ____
> Q7: The claim about depth L in line 208 may not be true because whether the minimum eigenvalue will increase or not with larger L highly depends on the activation functions as shown in Figure 3,4. Moreover, I am curious about why the minimum eigenvalue of Tanh does not increase with an increasing L especially when it achieves larger β1 and β2 and hence a larger upper bound of the minimum eigenvalue than ReLU and Sigmoid?
>
> A7: Our claim in line 208 is based on information obtained from Theorem 1. The bound in Theorem 1 will become larger when it has a skip layer. And when the skip layer exists, the lower bound will become larger as the depth increases.
> Compared with the upper bound of the minimum eigenvalue, the lower bound is often more concern by researchers. Although the values of β1 and β2 of the activation function Tanh are large, the value of β3 is small, and β3 plays a major role in the lower bound of the minimum eigenvalue of NTK, so the lower bound of the minimum eigenvalue of tanh is small in the experiment.

---

### Official Review · Reviewer_Nckt · 2022-07-11

**Rating:** 5
**Confidence:** 4
**Soundness:** 2 fair
**Presentation:** 3 good
**Contribution:** 3 good

**Summary:**

This paper provides the theoretical analysis of a fully-connected neural network with mixed activation functions in both infinite and finite width schemes. Furthermore, the authors provide analysis on both lower and upper bounds of minimum eigenvalues of NTK and generalization error bounds in SGD. Finally, the authors provide empirical experiments to support their theoretical claims on NAS problems.

**Questions:**

1. Why does the theoretical analysis of mixed activation neural networks help the understanding of the NAS problem? NAS algorithms such as DARTS or random search WS have two phases: search and evaluation. The authors show just an analysis of the evaluation phase in the paper, which corresponds to the final architecture. In other words, the paper seems to study NTK and generalization bounds on a particular family of the neural network, which has mixed activation functions.
2. Continuing from the Q1, how do we know architecture found from DARTS or random search RS are optimal architecture as stated in the line 147?
3. In lines 295-296, the author mentions that DARTS results coincide with the theoretical results. However, a long line of NAS research has shown DARTS's problems ([3, 4, 5, 6, 7]), especially the overfitting. Will other latest algorithms such as $\beta$-DARTS or DRNAS find the same results as DARTS?
4. In Table 2, Eigen-NAS on CIFAR-10 with $k=20$ seems to converge to sub-optimal architecture with a very small standard deviation of 0.01. Since the optimal architecture achieves $94.37\%$, can we conclude that Eigen-NAS only can find sub-optimal architecture? From 15,625 architecture candidates in NAS-Bench-201, what is the ranking of the architecture Eigen-NAS found?
5. Can you also verify the numerical results on DARTS Search Space with Eigen-NAS?

**Limitations:**

The author provides a thorough theoretical analysis of the minimum eigenvalue on NTK and generalization bounds on a neural network with mixed activation functions. However, the analysis is only for a particular family of neural networks (with mixed activation functions and selective skip-connections). Furthermore, the neural network analysis is on the final architecture, assuming that existing NAS algorithms, such as DARTS,  random search WS, or Eigen-NAS, can find the optimal architecture from their search phase.

Overall, I see the authors' theoretical analysis is not closely correlated to understanding the actual NAS problem. Furthermore, the empirical performance is far lower than existing state-of-the-art in NAS literature. Therefore, I give a reject.

1. Cao, Yuan, and Quanquan Gu. "Generalization bounds of stochastic gradient descent for wide and deep neural networks." Advances in neural information processing systems 32 (2019).
2. Nguyen, Quynh, Marco Mondelli, and Guido F. Montufar. "Tight bounds on the smallest eigenvalue of the neural tangent kernel for deep relu networks." International Conference on Machine Learning. PMLR, 2021.
3. Ye, Peng, et al. "b-DARTS: Beta-Decay Regularization for Differentiable Architecture Search." Proceedings of the IEEE/CVF Conference on Computer Vision and Pattern Recognition. 2022.
4. Chen, Xiangning, et al. "Drnas: Dirichlet neural architecture search." arXiv preprint arXiv:2006.10355 (2020).
5. Liang, Hanwen, et al. "Darts+: Improved differentiable architecture search with early stopping." arXiv preprint arXiv:1909.06035 (2019).
6. Dong, Xuanyi, and Yi Yang. "Nas-bench-201: Extending the scope of reproducible neural architecture search." arXiv preprint arXiv:2001.00326 (2020).
7. Yang, Antoine, Pedro M. Esperança, and Fabio M. Carlucci. "NAS evaluation is frustratingly hard." arXiv preprint arXiv:1912.12522 (2019).

**Strengths And Weaknesses:**

Strengths
1. Building upon Cao et al. and Nguyen et al.'s proof framework, the author provides the theoretical analysis of the minimal eigenvalue of NTK and generalization error on mixed activation neural networks.

Weaknesses
1. A considerable gap exists between theoretical analysis and the actual NAS problem. Their theoretical results only provide the minimum eigenvalue of NTK and generalization error of a fully-connected neural network with mixed activation functions. I will elaborate on this in more detail in the Question section.
2. NAS-Bench-201 results in Table 2 are not compelling enough. Latest methods such as $\beta$-DARTS [3] or DrNAS [4] find the optimal architecture in NAS-Bench-201.
3. General NAS problems contain a larger degree of freedom in choosing the components of a neural network, not only just activation functions and skip connections. Similar to the first point, there is a considerable discrepancy between the analysis and practice.

---

> ### Author Response · Authors · 2022-08-02
> **Response to the reviewer Nckt (2 out of 2)**
>
> ### References
>
> [1] Ye, Peng, Li Baopu, Li Yikang, Chen Tao, Fan Jiayuan, and Ouyang Wanli "β -DARTS: Beta-Decay Regularization for Differentiable Architecture Search." CVPR (2022).
>
> [2] Oymak, Samet, Li Mingchen, and Soltanolkotabi Mahdi. "Generalization Guarantees for Neural Architecture Search with Train-Validation Split." ICML2021.
>
> [3] Mellor, Joseph, Turner Jack, Storkey Amos, and Crowley Elliot J. "Neural architecture search without training." ICML (2021).
>
> [4] Xu, Jingjing, Zhao Liang, Lin Junyang, Gao Rundong, Sun Xu, and Yang Hongxia. "KNAS: Green Neural Architecture Search." ICML (2021).

---

> > ### Comment · Reviewer_Nckt · 2022-08-06
> > **Thank you for the rebuttal.**
> >
> > Thank you to the authors for their rebuttal in answering my questions. After the rebuttal, I agree that their NTK analysis on mixed activations and skip-connection setup can be the direction to understanding the neural architecture search field. However, I am still not convinced since this theoretical approach is limited to a subset of the NAS scenario: leveraging NTK to find the architecture topology. To my understanding, the upper/lower bound of the minimum eigenvalue of NTK (Theorem 1) determines the generalization bound (Theorem 3). Therefore, their theoretical framework supports the NTK-based search method but not the general NAS approach. I believe the authors are overstating their contributions.
> >
> > In conclusion, I raise my rating from reject to borderline accept.

---

> > > ### Author Response · Authors · 2022-08-08
> > > **Thankful for your response**
> > >
> > > Dear reviewer Nckt,
> > >
> > > We are thankful for your feedback for improving our manuscript.
> > >
> > > We agree with the reviewer that our theoretical framework supports the NTK-based search method rather than the explanation of any NAS algorithm proposed so far.
> > >
> > > To that end, according to your suggestions, we have renamed our title to “**Generalization Properties of NAS under Activation and Skip Connection Search**”, restated our story, and polished our abstract/introduction/conclusion to reflect this new perspective.
> > >
> > > Best regards,
> > >
> > > Authors

---

> ### Author Response · Authors · 2022-08-02
> **Response to the reviewer Nckt (1 out of 2)**
>
> We are thankful to the reviewer Nckt for their feedback. We respond below to their core remarks:
>
> Q1: How do we know architecture found from DARTS or random search RS are optimal architecture as stated in the line 147?
>
> A1: Here the “optimal” terminology indicates choosing the optimal architecture under our metrics (i.e. the minimum eigenvalue of NTK matrix) rather than the architecture achieving the best performance. We have polished this sentence to make the meaning clear.
>
> ____
>
> Q2: Will other latest algorithms such as β-DARTS or DRNAS find the same results as DARTS?
>
> A2: We have conducted our experiments in 5-layers MLP using β-DARTS. Compared with the results of DARTS, the experimental results (added in Appendix F.8 of the paper and displayed on this [link](https://imgur.com/a/6L8eB3A)) of β-DARTS indicate that the probability difference between different activation functions is smaller, which may verify that DARTS can easily overfit (as identified by the reviewer).
> ____
>
> Q3: In Table 2, Eigen-NAS on CIFAR-10 with k=20  seems to converge to suboptimal architecture with a very small standard deviation of 0.01. Since the optimal architecture achieves 94.37, can we conclude that Eigen-NAS only can find sub-optimal architecture? From 15,625 architecture candidates in NAS-Bench-201, what is the ranking of the architecture Eigen-NAS found?
>
> A3: Our proposed Eigen-NAS is a train-free algorithm that directly selects untrained models after initialization. The selected neural networks for each experiment are not exactly the same due to the different initialization. That means convergence is not suitable for describing such train-free algorithms. According to the results provided in Table 1 of [1], it is a common phenomenon that the results of the NAS algorithm have a small variance. In fact, the scatter plot and correlations are reported in Appendix F.5. Compared with [1] and [2], the absolute value of the Kendall correlation coefficient of Eigen-NAS is larger, which translates to a stronger correlation with respect to the accuracy.
>
> ____
> Q4: Can you also verify the numerical results on DARTS Search Space with Eigen-NAS?
>
> A4: We have already conducted results on NAS-Bench-201 in Table.2, sec. 5.2 that DARTS are also compared in the Table. The experimental results show that our Eigen-NAS algorithm achieves at least 4% improvement on all three datasets compared to DARTS.
>
> ____
> Q5: The neural network analysis is on the final architecture, assuming that existing NAS algorithms can find the optimal architecture from their search phase.
>
> A5: We do NOT assume that the final architecture has been found, and then analyze it. This is why we don't have a fixed architecture. Instead, we consider a search space containing a variety of different activation functions and skip layers. The reviewer is correct that we do not study the selection algorithm, but according to our theoretical results, we propose a train-free algorithm, called Eigen-NAS, which indicates that the minimum eigenvalue is a promising metric to guide NAS (without training).
>
>
> ____
> Q6: The empirical performance is far lower than existing state-of-the-art in NAS literature.
>
> A6: We respectfully disagree with the reviewer.
>
> We need to emphasize that the target of this work is not to pursue state-of-the-art performance but to provide an initial, non-trivial step to the theoretical understanding of NAS. We establish an answer to two fundamental questions in NAS from the perspective of theory (as also emphasized in the [general response](https://openreview.net/forum?id=aQySSrCbBul&noteId=bUbnIQ5GmIR).
>
> * How does different architectures/components search affect the final performance?
> * How to guide the searching procedure of NAS based on theory?
>
> Regarding the second question, our theory motivates us to design a *train-free* algorithm, which achieves a promising performance as empirically supported by NAS-bench-101 and NAS-bench-201 with high computational efficiency (orders of magnitude faster than DARTS methods).
>
> If the reviewer’s claim was right, theoretical analyses of NAS [2] or designing *train-free* algorithms [3, 4] could not be fairly treated.
>
> ____
>
> We appreciate the remarks of the reviewer and have made our best effort to address their concerns in the empirical validation.  Besides, our framework contains non-trivial theoretical derivations, e.g. as highlighted in the [general response](https://openreview.net/forum?id=aQySSrCbBul&noteId=bUbnIQ5GmIR),  which have not appeared in the literature previously. If the reviewer is aware of any such theoretical developments in the literature or has any technical comments on our proof, then we are more than happy to discuss those.

---

### Official Review · Reviewer_Yjm8 · 2022-07-18

**Rating:** 6
**Confidence:** 1
**Soundness:** 3 good
**Presentation:** 3 good
**Contribution:** 2 fair

**Summary:**

This paper extends the Neural Tangent Kernel limit to different (and mixed) activations functions, as well as a varying proportion of skip-connections. Then this paper is able to bound the minimum eigenvalue of the neural tangent kernel matrix, and to link this value to the generalization capability of the model. This allows to guide Neural Architecture search without training.

The authors validate their methods numerically by first showing that the derived bounds match empirical results, and that using their bounds to guide NAS improves NAS performances.



**Questions:**

Can this framework be used to guide NAS through choices more meaningful than choosing between a ReLu and a Sigmoid?

Could the authors investigate how reliable is their criterion to guide NAS? Maybe something like how often their criterion would rank two models in the correct order?

**Limitations:**

The authors clearly state that a big limitation of their work is that it only works for FC networks, and not more common CNN or Transformers.

**Strengths And Weaknesses:**

# Originality

While I am not very familiar with the field, this is to my knowledge the first time that an NTK approximation was derived for various and mixed activations functions.

# Quality

The paper contributions seems sounds, though the empirical validation could be more convincing: how well does the criterion rank different architectures?

# Clarity

The paper is clearly written. The title and the abstract, while quite clear, appear to be a little misleading: I don't think this paper helps understanding NAS (for instance NAS convergence), rather than providing a criterion which can help guide NAS.

# Significance

While the empirical results don't show a huge improvement, being able to guide NAS across different activation functions and skip-connections proportions seems useful. The result would be much more significant if it applied to CNN and Transformers, but this is an interesting progress.
I wish the authors had tested their approach on more challenging tasks, and with more competitive activation functions.

---

> ### Author Response · Authors · 2022-08-02
> **Response to the reviewer Yjm8**
>
> We are thankful to the reviewer Yjm8 for their feedback. We respond below to their core remarks, along with the [general response](https://openreview.net/forum?id=aQySSrCbBul&noteId=bUbnIQ5GmIR):
>
> Q1: How well does the criterion rank different architectures? How often would the criterion rank two models in the correct order?
>
> A1: The scatter plot and correlations have been reported in Appendix F.5. Compared with [1] and [2], the absolute value of Kendall correlation coefficient of **Eigen-NAS** is larger, which translates to a **stronger correlation with respect to the accuracy**.
>
> ____
>
> Q2: Experiments on more challenging tasks.
>
> A2: Our experiments are mainly divided into two parts:
>
> * The verification of the theorem. The experiments are based on the search space defined by the theory, using the DARTS algorithm search for activation function/skip layers on MLP (Figure.1 in sec. 5.1) and CNN (Figure.7 in Appendix. F.7).
>
> *  Use Eigen-NAS to guide NAS on the standard NAS benchmark. It is a challenging task that the search space here contains all possible network architectures in the benchmark.
>
> The task selected in this work is the image classification, where most of the NAS papers conduct their benchmarking. Therefore, we do conduct the experiments that are both challenging in terms of the task and we use competitive benchmarks.
>
> ____
>
> Q3: Can this framework be used to guide NAS through choices more meaningful than choosing between a ReLu and a Sigmoid?  (more competitive activation functions).
>
> A3: Yes, in practice we validate our framework beyond the theoretical analysis. In particular, we use our framework with standard NAS benchmarks, i.e. NAS-Bench-101 and NAS-Bench-201, which are popular benchmarks used by practitioners. Those benchmarks have a larger search space than this considered in the theory and our framework still works well.
>
>
>
> ### References
>
> [1] Mellor, Joseph, Turner Jack, Storkey Amos, and Crowley Elliot J. "Neural architecture search without training." ICML (2021).
>
> [2] Xu, Jingjing, Zhao Liang, Lin Junyang, Gao Rundong, Sun Xu, and Yang Hongxia. "KNAS: Green Neural Architecture Search." ICML (2021).

---

> > ### Comment · Reviewer_Yjm8 · 2022-08-08
> > **Thank you**
> >
> > Thank you to the authors for their answers to my review!
> >
> > While I still think that the title and the abstract are a little misleading and the paper orientation could be improved, I raise my rating to 6.

---

> > > ### Author Response · Authors · 2022-08-08
> > > **Question on the title**
> > >
> > > Dear reviewer Yjm8,
> > >
> > > We are thankful for your response.
> > >
> > > Just a minor clarification: We have updated the title to “**Generalization Properties of NAS under Activation and Skip Connection Search**”, restated our story, and polished our abstract/introduction/conclusion to reflect this new perspective.
> > >
> > > I guess this happened concurrently with our revisions, so we are not sure if this is what the reviewer meant.
> > >
> > > We are happy to discuss further the updates on the title, but we believe it reflects more accurately our findings and theoretical contributions.
> > >
> > > Best regards,
> > >
> > > Authors

---

> > > > ### Comment · Reviewer_Yjm8 · 2022-08-09
> > > > **Title**
> > > >
> > > > Thank you for your comment, my answer indeed happened concurrently with your change.
> > > >
> > > > Though I find the new title / abstract much better, I decided to keep my rating to 6 after some hesitations, due to limited significance.

---

### Official Review · Reviewer_h7Bq · 2022-07-18

**Rating:** 6
**Confidence:** 4
**Soundness:** 3 good
**Presentation:** 3 good
**Contribution:** 2 fair

**Summary:**

This paper provides an NTK analysis for a class of residual networks with a mixture of activation functions at different layers. The authors derive bounds on the minimum eigenvalue for both finite and infinite width settings. They also give a generalization bound for this class of network architecture.

**Questions:**

What is the data used in the simulation of minimum eigenvalues of NTK for different activations and architectures?

**Strengths And Weaknesses:**

The main contribution of this work, in my opinion, is a non-trivial derivation of lower and upper bounds on the minimum and for residual networks with a mixture of activation functions. However, analysis and proof techniques are mainly taken from the well-established NTK literature, such as Huang et al., 2020, Oymak and Soltanolkotabi, 2020… Similar generalization bounds are established by Cao and Gu, 2019, and Arora et al., 2019a.

There is an interesting connection to NAS that the authors observe empirically where NAS always picks up RELU and/or Leaky RELU in the optimal search. That coincides with the fact that RELU and LeakyRELU imply larger minimum eigenvalue in the NTK sense. However, deep neural networks in practice do not generally operate in the NTK regime.

In short, I see this work purely as an NTK analysis rather than an “unifying framework” for NAS. I don’t see how the work studies “optimization and generalization of NAS” or how the derived results guide NAS.

In terms of the presentation, the paper is well-written and easy to follow. One thing I would suggest is to instantiate Theorem 1 with a few special cases, such as the standard 2-layer networks with ReLU activation. Illustrating the Hermite coefficients of such cases would help the readers to compare with existing results, e.g., Oymak and Soltanolkotabi, 2020.

---

> ### Author Response · Authors · 2022-08-02
> **Response to the reviewer h7Bq**
>
> We are thankful to the reviewer h7Bq for their feedback. We respond below to their core remarks:
>
> Q1: Analysis and proof techniques are mainly taken from the well-established NTK literature.
>
> A1: We respectfully disagree. Their analyses **are not directly applicable** to our setting for the following two reasons:
>
>  * Results of [3] are only valid to ReLU due to its homogeneity.
>  * The probability of concentration inequality can be negative in Theorem 3.2 of [1].
>
> To tackle these technical challenges, we develop the following techniques:
> a) to handle the non-homogeneous property of Tanh, Sigmoid, and Swish, we develop a new integral estimation approach for minimal eigenvalue estimation. To connect the minimum eigenvalues of NTK and generalization errors, we use the Lipschitz continuity to avoid the special property of ReLU.
> b) to tackle negative probability, we introduce a new technique from [2] to replace Gershgorin circle theorem for minimum eigenvalue estimation, which avoids concentration inequalities with negative probability in some certain cases.
>
> Besides, our proof also allows for residual connections.
>
> Additionally, our analysis framework under general activations and residual connections with experimental validations on three compelling benchmarks might be of interest in its own right in the deep learning theory community.
>
> ____
>
> Q2: Deep neural networks in practice do not generally operate in the NTK regime.
>
> A2:
> In practice, under He or LeCun initialization, ResNet-18 still performs like a linear function training style to some extent but achieves with more than $92\%$ accuracy on CIFAR10.
> We provided the experiments to see the ratio w.r.t. the weights changing before/after training, $\| \mathbf{W}_t - \mathbf{W}_0 \| / \|  \mathbf{W}_0 \|$ vs. epochs, see the figure on [this link](https://imgur.com/a/brGWPMn).
>
> This is similar to the target of NTK that works in a lazy training regime, and accordingly NTK can still describe the optimization and generalization properties of neural networks to some extent, e.g., training in the early stage [3],  convergence [4, 5], generalization [6] and spectral bias [7].
> In practice, NTK-style algorithms (e.g., CNTK [8]) are able to achieve comparable performance on several typical image datasets.
>
> Accordingly, using NTK as a tool to study the theoretical properties of NAS is a good starting point. We would appreciate it if the reviewer can provide another possible way for analysis.
>
> ____
> Q3: Instantiate Theorem 1 with a few special cases to compare with existing results.
>
> A3: The previous work of [9] computes the Hermite coefficients of different activation functions. For example, the first three coefficients of Sigmoid are 0.5, 0.206621 and 0. However these coefficients ranging from (0,1) pale in importance for studying the effect of different activations functions on NAS.
>
> Theorem 3 which gives the generalization error bound for this special search space in our paper is a more appropriate result for comparison with previous work [10].
> * Same point: the convergence rate of the main theorem (Theorem 3) in [10] is the same as the Theorem 3 in our paper, which is $1/\sqrt{n}$.
> * Difference: Theorem 3 in our paper gives the generalization bound for five commonly used different activation functions with different NTK, analyzes their similarities and differences, and is suitable for multi-layer neural networks. However, in Theorem 3 of [10], the specific results of different activation functions are not given, and put harsh conditions for the activation functions (for example, the most commonly used activation function ReLU does not satisfy the bounded second derivative). And the theorem is only based on a 2-layer neural network.
>
> ____
> Q4: What is the data used in the simulation of minimum eigenvalues of NTK for different activations and architectures?
> A4: We initially sampled a few Gaussian data points with zero-mean and unit-variance, and then normalized the data to 1.
>
> ### References
>
> [1] Nguyen, Quynh, Mondelli Marco, and Montufar Guido. "Tight Bounds on the Smallest Eigenvalue of the Neural Tangent Kernel for Deep ReLU Networks." ICML (2021).
>
> [2] Yaskov Pavel. "Lower bounds on the smallest eigenvalue of a sample covariance matrix." Electron. Commun. Probab. 19: 1-10 (2014). DOI: 10.1214/ECP.v19-3807.
>
> [3] Hu, Wei, Xiao Lechao, Adlam Ben and  Pennington Jeffrey. "The Surprising Simplicity of the Early-Time Learning Dynamics of Neural Networks." NeurIPS (2020).
>
> [4] Allen-Zhu, Zeyuan, Li Yuanzhi, and Song Zhao. "A Convergence Theory for Deep Learning via Over-Parameterization." NeurIPS (2019).
>
> [5] Du, Simon, Lee Jason, Li Haochuan, Wang Liwei, and Zhai Xiyu. "Gradient Descent Finds Global Minima of Deep Neural Networks." ICML (2019).
>
> [6] Cao, Yuan, and Gu Quanquan. "Generalization Bounds of Stochastic Gradient Descent for Wide and Deep Neural Networks." ICML (2019).

---

> > ### Author Response · Authors · 2022-08-02
> > **Additional references**
> >
> > [7] Cao, Yuan, Fang Zhiying, Wu Yue, Zhou Ding-Xuan, and Gu Quanquan. "Towards Understanding the Spectral Bias of Deep Learning." IJCAI (2021).
> >
> > [8] Arora, Sanjeev, Du Simon S., Hu Wei, Li Zhiyuan, Salakhutdinovk Ruslan, and Wang Ruosong. "On Exact Computation with an Infinitely Wide Neural Net." NeurIPS (2019).
> >
> > [9] Zhang, Gege, Li Gangwei, Shen Ningwei, and Zhang Weidong. "The Expressivity and Training of Deep Neural Networks: toward the Edge of Chaos?" Neurocomputing (2020).
> >
> > [10] Oymak, Samet, Li Mingchen, and Soltanolkotabi Mahdi. "Generalization Guarantees for Neural Architecture Search with Train-Validation Split." ICML2021.

---

> > > ### Author Response · Authors · 2022-08-06
> > > **Any remaining questions from reviewer h7Bq?**
> > >
> > > Dear reviewer h7Bq,
> > >
> > > We are thankful for your constructive feedback. We agree with the reviewer that the NTK analysis does not characterize completely all the phenomena observed in the actual training of deep neural networks. However, it has been widely reported that it provides a good starting point for understanding the success of deep neural networks [1-6].
> > >
> > > In this work, we conduct *a non-trivial theoretical analysis*, which requires us to overcome a few limitations as we mentioned in Q1 of our [response](https://openreview.net/forum?id=aQySSrCbBul&noteId=3mn8BZmEwNk). This analysis aims to become the starting point *for a principled understanding of NAS*. We provide a number of *new experiments in the rebuttal*, based on the suggestions of the reviewers to further strengthen the connection with our analysis.
> > >
> > > Since the discussion period is closing soon, we would like to check if the reviewer has been covered by our responses and the improvements in our manuscript.
> > >
> > > [1] Hu, Wei, Xiao Lechao, Adlam Ben, and Pennington Jeffrey. "The Surprising Simplicity of the Early-Time Learning Dynamics of Neural Networks." NeurIPS (2020).
> > >
> > > [2] Allen-Zhu, Zeyuan, Li Yuanzhi, and Song Zhao. "A Convergence Theory for Deep Learning via Over-Parameterization." NeurIPS (2019).
> > >
> > > [3] Du, Simon, Lee Jason, Li Haochuan, Wang Liwei, and Zhai Xiyu. "Gradient Descent Finds Global Minima of Deep Neural Networks." ICML (2019).
> > >
> > > [4] Cao, Yuan, and Gu Quanquan. "Generalization Bounds of Stochastic Gradient Descent for Wide and Deep Neural Networks." ICML (2019).
> > >
> > > [5] Cao, Yuan, Fang Zhiying, Wu Yue, Zhou Ding-Xuan, and Gu Quanquan. "Towards Understanding the Spectral Bias of Deep Learning." IJCAI (2021).
> > >
> > > [6] Arora, Sanjeev, Du Simon S., Hu Wei, Li Zhiyuan, Salakhutdinovk Ruslan, and Wang Ruosong. "On Exact Computation with an Infinitely Wide Neural Net." NeurIPS (2019).

---

> > > > ### Author Response · Authors · 2022-08-09
> > > > **Minor update to reviewer h7Bq for the title change**
> > > >
> > > > Dear reviewer h7Bq,
> > > >
> > > > We are thankful for your feedback and thoughtful questions.
> > > >
> > > > Since the rebuttal window is closing, we want to make sure that the reviewer is informed about the change of the title we have done. The updated title is “Generalization Properties of NAS under Activation and Skip Connection Search”. We believe this reflects better our contributions.
> > > >
> > > > Best regards,
> > > >
> > > > Authors

---

### Author Response · Authors · 2022-08-02
**General response**

We are grateful to the reviewers for their constructive and detailed feedback. Before responding to all questions and remarks specifically, we would like to emphasize the improvements made in our manuscript in the course of this revision:



Q1: Why do we use understanding NAS in the title?

A1:
Understanding NAS includes two parts:

* The first one is to interpret the mechanism of NAS, e.g., how the different activation functions and skip layers affect the final performance and generalization of NAS.

* The second one is to guide NAS, e.g., how does our theoretical results indicate the direction to ensure searching of a performant and efficient architecture? We follow the theoretical setting of [1] that studies bilevel optimization for shallow activation search to provide generalization guarantees for neural architecture search.

Therefore, we have provided theoretical guarantees in terms of optimization and generalization of neural networks via activation functions search and residual connection search in Theorem 3, sec. 4.5.

More importantly, our results demonstrate that the generalization error bound of the searched architecture trained by SGD mainly depends on the NTK matrix and its minimal eigenvalue, which are determined by the searched activation functions and skip connections. Consequently, the relationship between the generalization error bound and the searched architecture are combined together via the minimal eigenvalue of NTK.

According to our theoretical results on this relationship, the minimal eigenvalue of NTK in turn motivates us to design a train-free algorithm Eigen-NAS. Our experimental evidence on Eigen-NAS indicates that the minimum eigenvalue is a promising metric to guide NAS (without training) as suggested by our theory.

____

Q2: Why we only consider MLP and skip layers, the difficulty of considering more complex structures like CNN.

A2:  It is non-trivial to extend our proof framework to cover more general structures in NAS such as CNN due to the tensors that emerge.
To be specific, it requires the element-recursive form of the NTK matrices to be transformed into a global-recursive form (similar to Lemma 1 in our paper), then analyze its minimum eigenvalue.
 Besides, the contraction operation of tensors, the locality and boundary effects of convolutional layers in CNNs make the analysis intractable. We believe it is a topic in its own right, which has been discussed/studied in sec. 6.

____

We emphasize once more that **our goal here is to conduct a theoretical analysis**. We do not claim to have state-of-the-art results, however we simply provide an empirical validation of our theoretical framework.

### References

[1] Oymak, Samet, Li Mingchen, and Soltanolkotabi Mahdi. "Generalization Guarantees for Neural Architecture Search with Train-Validation Split." ICML (2021).

[2] Du, Simon, Lee Jason, Li Haochuan, Wang Liwei, and Zhai Xiyu. "Gradient Descent Finds Global Minima of Deep Neural Networks." ICML (2019).

[3] Cao, Yuan, and Gu Quanquan. "Generalization Bounds of Stochastic Gradient Descent for Wide and Deep Neural Networks." ICML (2019).

---

### Author Response · Authors · 2022-08-08
**Updates during rebuttal**

We are grateful to the reviewers for their constructive and detailed feedback. During the discussion phase, we have made the following improvements to this work:

* We update the title to “**Generalization Properties of NAS under Activation and Skip Connection Search**” in the revised version. This makes the title more refined to our proofs and provides a clear idea of the search space in the theoretical proofs.
* We have revised the text (e.g. abstract, introduction) to reflect the comments of the reviewers and to constrain appropriately the search space.
* Experiments: We compared our Eigen-NAS algorithm with another train-free method NASI [1], reported in Tab. 2; and conducted our architecture search using the latest algorithm $\beta$-DARTS [2] in Appendix F.8.

[1] Shu, Yao, Cai Shaofeng, Dai Zhongxiang, Ooi Beng, and Low Bryan Kian Hsiang. "NASI: Label- and Data-agnostic Neural Architecture Search at Initialization." ICLR (2022).

[2] Ye, Peng, Li Baopu, Li Yikang, Chen Tao, Fan Jiayuan, and Ouyang Wanli "β -DARTS: Beta-Decay Regularization for Differentiable Architecture Search." CVPR (2022).

---

### Meta-Review · Area_Chair_Dh64 · 2022-08-26

**Recommendation:** Accept
**Confidence:** Certain

**Metareview:**

This work relates NAS to the conditioning of a DNN through the NTK framework. This work is well supported theoretically and empirically, and making this connexion is surprising. Given the potential interest for the NAS community, I recommend accepting this paper.

**Award:**

No

---

### Decision · Program_Chairs · 2022-09-14

Accept